# Uncoupled and Convergent Learning in Two-Player Zero-Sum Markov Games with Bandit Feedback

**Yang Cai**
Yale University
yang.cai@yale.edu

**Haipeng Luo**
University of Southern California
haipengl@usc.edu

**Chen-Yu Wei**
University of Virginia
chenyu.wei@virginia.edu

**Weiqiang Zheng**
Yale University
weiqiang.zheng@yale.edu

## Abstract

We revisit the problem of learning in two-player zero-sum Markov games, focusing on developing an algorithm that is *uncoupled*, *convergent*, and *rational*, with non-asymptotic convergence rates to Nash equilibrium. We start from the case of stateless matrix game with bandit feedback as a warm-up, showing an $\mathcal{O}(t^{-\frac{1}{8}})$ last-iterate convergence rate. To the best of our knowledge, this is the first result that obtains finite last-iterate convergence rate given access to only bandit feedback. We extend our result to the case of irreducible Markov games, providing a last-iterate convergence rate of $\mathcal{O}(t^{-\frac{1}{9+\varepsilon}})$ for any $\varepsilon > 0$. Finally, we study Markov games without any assumptions on the dynamics, and show a *path convergence* rate, a new notion of convergence we define, of $\mathcal{O}(t^{-\frac{1}{10}})$. Our algorithm removes the coordination and prior knowledge requirement of [WLZL21a], which pursued the same goals as us for irreducible Markov games. Our algorithm is related to [CMZ21, CWC21] and also builds on the entropy regularization technique. However, we remove their requirement of communications on the entropy values, making our algorithm entirely uncoupled.

## 1 Introduction

In multi-agent learning, a central question is how to design algorithms so that agents can *independently* learn (i.e., with little coordination overhead) how to interact with each other. Additionally, it is desirable to maximally reuse existing single-agent learning algorithms, so that the multi-agent system can be built in a modular way. Motivated by this question, *decentralized* multi-agent learning emerges with the goal to design decentralized systems, in which no central controller governs the policies of the agents, and each agent learns based on only their local information – just like in a single-agent algorithm. In recent years, we have witnessed significant success of this new decentralized learning paradigm. For example, *self-play*, where each agent independently deploys the same single-agent algorithm to play against each other without further direct supervision, plays a crucial role in the training of AlphaGo [SSS+17] and AI for Stratego [PDVH+22].

Despite the recent success, many important questions remain open in decentralized multi-agent learning. Indeed, unless the decentralized algorithm is carefully designed, self-play often falls short of attaining certain sought-after global characteristics, such as convergence to the global optimum or stability as seen in, for example, [MPP18, BP18].

In this work, we revisit the problem of learning in two-player zero-sum Markov games, which has received extensive attention recently. Our goal is to design a decentralized algorithm that resembles

standard single-agent reinforcement learning (RL) algorithms, but with an additional crucial assurance, that is, *guaranteed convergence* when both players deploy the algorithm. The simultaneous pursuit of independence and convergence has been advocated widely [BV01, AY16, WLZL21a, SZL$^+$21], while the results are still not entirely satisfactory. In particular, all of these results rely on assumptions on the dynamics of the Markov game. Our paper takes the first step to remove such assumptions.

More specifically, our goal is to design algorithms that simultaneously satisfy the following three properties (the definitions are adapted from [BV01, DDK11]):

- **Uncoupled**: Each player $i$'s action is generated by a standalone procedure $\mathcal{P}_i$ which, in every round, only receives the current state and player $i$'s own reward as feedback (in particular, it has no knowledge about the actions or policies used by the opponent). There is no communication or shared randomness between the players.
- **Convergent**: The policy pair of the two players converges to a Nash equilibrium.
- **Rational**: If $\mathcal{P}_i$ competes with an opponent who uses a policy sequence that converges to a stationary one, then $\mathcal{P}_i$ converges to the best response of this stationary policy.

The uncoupledness and rationality property capture the independence of the algorithm, while the convergence property provides a desirable global guarantee. Interestingly, as argued in [WLZL21a], if an algorithm is uncoupled and convergent, then it is also rational, so we only need to ensure that the algorithm is uncoupled and convergent. Regarding the notion of convergence, the standard definition above only allows *last-iterate* convergence. Considering the difficulty of achieving such convergence, in the related work review (Section 2) and in the design of our algorithm for general Markov games (Section 6), we also consider weaker notions of convergence, including the *best-iterate* convergence, which only requires that the Cesaro mean of the duality gap is convergent, and the *path* convergence, which only requires the convergence of the Cesaro mean of the duality gap *assuming minimax/maximin policies are followed in future steps*. The precise definitions of these convergence notions are given at the end of Section 3.

## 1.1 Our Contributions

The main results in this work are as follows (see also Table 1 for comparisons with prior works):

- As a warm-up, for the special case of matrix games with bandit feedback, we develop an uncoupled algorithm with a last-iterate convergence rate of $\mathcal{O}(t^{-\frac{1}{8}})$ under self-play (Section 4). To the best of our knowledge, this is the first algorithm with provable last-iterate convergence rate in the setting.
- Generalizing the ideas from matrix games, we further develop an uncoupled algorithm for irreducible Markov games with a last-iterate convergence rate of $\mathcal{O}(t^{-\frac{1}{9+\varepsilon}})$ for any $\varepsilon > 0$ under self-play (Section 5).
- Finally, for general Markov games without additional assumptions, we develop an uncoupled algorithm with a path convergence rate of $\mathcal{O}(t^{-\frac{1}{10}})$ under self-play (Section 6).

Our algorithms leverage recent advances on using entropy to regularize the policy updates [CWC21, CMZ21] and the Nash-V-styled value updates [BJY20]. On the one hand, compared to [CWC21, CMZ21], our algorithm has the following advantages: 1) it does not require the two players to exchange their entropy information, which allows our algorithm to be fully uncoupled; 2) it does not require the players to have coordinated policy updates, 3) it naturally extends to general Markov games without any assumptions on the dynamics (e.g., irreducibility). On the other hand, our algorithm inherits appealing properties of Nash-V [BJY20], but additionally guarantees path convergence during execution.

## 2 Related Work

The study of two-player zero-sum Markov games originated from [Sha53], with many other works further developing algorithms and establishing convergence properties [HK66, PAI69, VDW78, FT91]. However, these works primarily focused on *solving* the game with full knowledge of its parameters (i.e., payoff function and transition kernel). The problem of *learning* in zero-sum games was first formalized by [Lit94]. Designing a *provably* uncoupled, rational, and convergent algorithm

Table 1: (Sample-based) Learning algorithms for finding NE in two-player zero-sum games. Our results are shaded. A halfcheck "✓̸" in the convergent column means that the policy convergence is proven only for one player (typically this is a result of asymmetric updates). (L) and (B) stand for last-iterate convergence and best-iterate convergence, respectively. (P) stands for path convergence, a weaker convergence notion we introduce (see Section 3, 6.1).
*: While [WLZL21a] also proposes an uncoupled and convergent algorithm for irreducible Markov games, their algorithm requires coordinated updates and some prior knowledge of the game, while ours does not. See Section 2.1 for a more detailed discussion.

| Setting | Algorithm | Uncoupled? | Converegent? |
|---|---|---|---|
| Matrix Game | EXP3 vs. EXP3 | ✓ | ✗ |
| | Algorithm 1 | ✓ | ✓(L) |
| Markov game + assumptions on dynamics | [DFG20] | ✓ | ✓̸(B) |
| | [ZTLD22, AVHC22] | ✓ | ✓̸(L) |
| | [SZL+21] | ✓ | ✗ |
| | [CMZ21] | ✗ | ✓(L) |
| | [WLZL21a] | ✓* | ✓(L) |
| | Algorithm 2 | ✓ | ✓(L) |
| Markov Game | [WHL17, JJJN21, HLWY22] [JLY22, XZS+22] | ✗ | ✓̸(B) |
| | [BJ20, XCWY20] [LYBJ21, CZG22] | ✗ | ✓(B) |
| | [BJY20, JLWY21] | ✓ | ✗ |
| | Algorithm 3 | ✓ | ✓(P) |

is challenging, with many attempts [SL99, BV01, HW03, CS07, AY16, SPO22] falling short in one aspect or another, often lacking either uncoupledness or convergence. Moreover, these works only establish asymptotic convergence without providing a concrete convergence rate.

## 2.1 Non-asymptotic convergence guarantees

Recently, a large body of works on learning two-player zero-sum Markov games use regret minimization techniques to establish *non-asymptotic* guarantees. They focus on fast computation under full information of payoff and transitions [CWC21, CCDX23, ZLW+22, SLY23, YM23], though many of their algorithms are decentralized and can be viewed as the first step towards the learning setting.

With rationality and uncoupledness satisfied, [DFG20] established one-sided policy convergence for players using independent policy gradient with asymmetric learning rates. Such an asymmetric update rule is also adopted by [ZTLD22, AVHC22] to establish one-sided policy convergence guarantees. When using a symmetric update rule, [SZL+21] developed a decentralized-Q learning algorithm. However, the convergence is only shown for the $V$-function maintained by the players instead of the policies being used, so the policies may still cycle and are not provably convergent in our definition. [ELS+23] studied regret minimization in general-sum Markov games and provided an algorithm with sublinear regret under self-play and *average-iterate* convergence rates to equibria, while our work focuses on last-iterate convergence rates to Nash equilibria.

To our knowledge, [WLZL21a] first provided an uncoupled, rational, and convergent algorithm with non-asymptotic convergence guarantee, albeit only for irreducible Markov game. They achieved this via *optimistic gradient descent/ascent*. Despite satisfying all our criteria, their algorithm still has unnatural coordination between the players and a requirement on some prior knowledge of the game such as the maximum revisiting time of the Markov game. Our algorithm removes all these extra requirements. A follow-up work by [CMZ21] improved the rate of [WLZL21a] using entropy regularization; however, this requires their players to inform the opponent about the entropy of their own policy, making the algorithm coupled again. We show that such an exchange of information is unnecessary under entropy regularization.

## 2.2 Further handling exploration

The algorithms introduced above all require full information or some assumption on the dynamics of the Markov game. To handle exploration, some works design coupled learning algorithms

which guarantee that the player's long-term payoff is at least the minimax value [BT02, WHL17, XCWY20, HLWY22, JLY22, JJJN21, XZS$^+$22]. Interestingly, as shown in [WHL17, HLWY22, JLY22, XZS$^+$22], if the player is paired with an optimistic best-response opponent (instead of using the same algorithm), the first player's strategy can converge to the minimax policy. [XCWY20, BJ20, LYBJ21, CZG22] developed another coupled learning framework to handle exploration, but with symmetric updates on both players. In each round, the players need to jointly solve a general-sum equilibrium problem due to the different exploration bonus added by each player. Hence, the execution of these algorithms is more similar to the Nash-Q algorithm by [HW03].

So far, exploration has been handled through coupled approaches that are also not rational. To our knowledge, the first uncoupled and rational algorithm that handles exploration is the Nash-V algorithm by [BJY20]. Nash-V can output a nearly-minimax policy through weighted averaging [JLWY21]; however, it is not provably convergent during execution. A major remaining open problem is whether one can design a natural algorithm that is provably rational, uncoupled, and convergent with exploration capability. Our work provides the first progress towards this goal.

### 2.3 Other works on last-iterate convergence

Uncoupled Learning dynamics in normal-form games with provable last-iterate convergence rate receives extensive attention recently. Most of the works assume that the players receive gradient feedback, and convergence results under bandit feedback remain sparse. Linear convergence is shown for strongly monotone games or bilinear games under gradient feedback [Tse95, LS19, MOP20, WLZL21b] and sublinear rates are proven for strongly monotone games with bandit feedback [BLM18, HIMM19, LZBZ21, TK22, DFR22, HH23]. Convergence rate to strict Nash equilibrium is analyzed by [GVGM21]. For monotone games that includes two-player zero-sum games as a special case, the last-iterate convergence rate of no-regret learning under gradient feedback has been shown recently [GPD20, COZ22, GTG22, CZ23]. With bandit feedback, [MPS20] showed an impossibility result that certain algorithms with optimal $\mathcal{O}(\sqrt{T})$ regret do not converge in last-iterate. To the best of our knowledge, there is no natural uncoupled learning dynamics with provable last-iterate convergence rate in two-player zero-sum games with bandit feedback.

## 3 Preliminaries

**Basic Notations** Throughout the paper, we assume for simplicity that the action set for the two players are the same, denoted by $\mathcal{A}$ with cardinality $A = |\mathcal{A}|$.[1] We usually call player 1 the $x$-player and player 2 the $y$-player. The set of mixed strategies over an action set $\mathcal{A}$ is denoted as $\Delta_{\mathcal{A}} := \{x : \sum_{a \in \mathcal{A}} x_a = 1; 0 \leq x_a \leq 1, \forall a \in \mathcal{A}\}$. To simplify notation, we denote by $z = (x, y)$ the concatenated strategy of the players. We use $\phi$ as the entropy function such that $\phi(x) = -\sum_{a \in \mathcal{A}} x_a \ln x_a$, and KL as the Kullback–Leibler (KL) divergence such that $\mathrm{KL}(x, x') = \sum_{a \in \mathcal{A}} x_a \ln \frac{x_a}{x'_a}$. The all-one vector is denoted by $\mathbf{1} = (1, 1, \cdots, 1)$.

**Matrix Games** In a two-player zero-sum matrix game with a loss matrix $G \in [0, 1]^{A \times A}$, when the $x$-player chooses action $a$ and the $y$-player chooses action $b$, the $x$-player suffers loss $G_{a,b}$ and the $y$-player suffers loss $-G_{a,b}$. A pair of mixed strategy $(x^\star, y^\star)$ is a *Nash equilibrium* for $G$ if for any strategy profile $(x, y) \in \Delta_{\mathcal{A}} \times \Delta_{\mathcal{A}}$, it holds that $(x^\star)^\top G y \leq (x^\star)^\top G y^\star \leq x^\top G y^\star$. Similarly, $(x^\star, y^\star)$ is a Nash equilibrium for a two-player zero-sum game with a general convex-concave loss function $f(x, y) : \Delta_{\mathcal{A}} \times \Delta_{\mathcal{A}} \to \mathbb{R}$ if for all $(x, y) \in \Delta_{\mathcal{A}} \times \Delta_{\mathcal{A}}$, $f(x^\star, y) \leq f(x^\star, y^\star) \leq f(x, y^\star)$. The celebrated minimax theorem [vN28] guarantees the existence of Nash equilibria in two-player zero-sum games. For a pair of strategy $(x, y)$, we use *duality gap* defined as $\mathrm{GAP}(G, x, y) \triangleq \max_{y'} x^\top G y' - \min_{x'} x'^\top G y$ to measure its proximity to Nash equilibria.

**Markov Games** A generalization of matrix games, which models dynamically changing environment, is *Markov games*. We consider infinite-horizon discounted two-player zero-sum Markov games, denoted by a tuple $(\mathcal{S}, \mathcal{A}, (G^s)_{s \in \mathcal{S}}, (P^s)_{s \in \mathcal{S}}, \gamma)$ where (1) $\mathcal{S}$ is a finite state space; (2) $\mathcal{A}$ is a finite action space for both players; (3) Player 1 suffers loss $G^s_{a,b} \in [0, 1]$ (respectively player 2 suffers

---

[1]We make this assumption only to simplify notations; our proofs can be easily extended to the case where the action sets of the two players are different.

loss $-G^s_{a,b}$) when player 1 chooses action $a$ and player 2 chooses action $b$ at state $s$; (4) $P$ is the transition function such that $P^s_{a,b}(s')$ is the probability of transiting to state $s'$ when player 1 plays $a$ and player 2 plays $b$ at state $s$; (5) $\gamma \in [\frac{1}{2}, 1)$ is a discount factor.

A stationary policy for player 1 is a mapping $\mathcal{S} \to \Delta_{\mathcal{A}}$ that specifies player 1's strategy $x^s \in \Delta_{\mathcal{A}}$ at each state $s \in \mathcal{S}$. We denote $x = (x^s)_{s \in \mathcal{S}}$. Similar notations apply to player 2. We denote $z^s = (x^s, y^s)$ as the concatenated strategy for the players and $z = (x, y)$. The value function $V^s_{x,y}$ denotes the expected loss of player 1 (or the expected payoff of player 2) given a pair of stationary policy $(x, y)$ and initial state $s$:

$$V^s_{x,y} = \mathbb{E}\left[\sum_{t=1}^{\infty} \gamma^{t-1} G^{s_t}_{a_t, b_t} | s_1 = s, a_t \sim x^{s_t}, b_t \sim y^{s_t}, s_{t+1} \sim P^{s_t}_{a_t, b_t}(\cdot), \forall t \geq 1\right].$$

The *minimax game value* on state $s$ is defined as $V^s_\star = \min_x \max_y V^s_{x,y} = \max_y \min_x V^s_{x,y}$. We call a pair of policy $(x_\star, y_\star)$ a *Nash equilibrium* if it attains minimax game value of a state $s$ (such policy pair necessarily attains the minimax game value over all states). The *duality gap* of $(x, y)$ is $\max_s (\max_{y'} V^s_{x,y'} - \min_{x'} V^s_{x',y})$. The $Q$-function on state $s$ under policy pair $(x, y)$ is defined via $Q^s_{x,y}(a, b) = G^s_{a,b} + \gamma \cdot \mathbb{E}_{s' \sim P^s_{a,b}(\cdot)}[V^{s'}_{x,y}]$, which can be rewritten as a matrix $Q^s_{x,y}$ such that $V^s_{x,y} = x^s Q^s_{x,y} y^s$. We denote $Q^s_\star = Q^s_{x_\star, y_\star}$ the $Q$-function under a Nash equilibrium $(x_\star, y_\star)$. It is known that $Q^s_\star$ is unique for any $s$ even when multiple equilibria exist.

**Uncoupled Learning with Bandit Feedback** We assume the following uncoupled interaction protocol: at each round $t = 1, \ldots, T$, the players both observe the current state $s_t$, and then, with the policy $x_t$ and $y_t$ in mind, they independently choose actions $a_t \sim x_t^{s_t}$ and $b_t \sim y_t^{s_t}$, respectively. Both of them then observe $\sigma_t \in [0, 1]$ with $\mathbb{E}[\sigma_t] = G^{s_t}_{a_t, b_t}$, and proceed to the next state $s_{t+1} \sim P^{s_t}_{a_t, b_t}(\cdot)$. Importantly, they do not observe each other's action.

**Notions of Convergence** For Markov games with the irreducible assumption (Assumption 1), given players' history of play $(s_t, x_t, y_t)_{t \in [T]}$, the *best-iterate* convergence rate is measured by the average duality gap $\frac{1}{T} \sum_{t=1}^T \max_{s,x,y} (V^s_{x_t,y} - V^s_{x,y_t})$, while the stronger *last-iterate* convergence rate is measured by $\max_{s,x,y} (V^s_{x_T,y} - V^s_{x,y_T})$, i.e., the duality gap of $(x_T, y_T)$. For general Markov games, we propose the *path* convergence rate, which is measured by the average duality gap at the visited states with respect to the optimal $Q$-function: $\frac{1}{T} \sum_{t=1}^T \max_{x,y} (x_t^{s_t \top} Q^{s_t}_\star y^{s_t} - x^{s_t \top} Q^{s_t}_\star y_t^{s_t})$. We remark that the path convergence guarantee is weaker than the counterpart of the other two notions of convergence in general Markov games, but still provides meaningful implications (see detailed discussion in Section 6.1 and Appendix F).

# 4 Matrix Games

In this section, we consider two-player zero-sum matrix games. We propose Algorithm 1 for decentralized learning of Nash equilibria. We only present the algorithm for the $x$-player as the algorithm for the $y$-player is symmetric.

---

**Algorithm 1** Matrix Game with Bandit Feedback

---

1: **Define:** $\eta_t = t^{-k_\eta}$, $\beta_t = t^{-k_\beta}$, $\epsilon_t = t^{-k_\epsilon}$ where $k_\eta = \frac{5}{8}$, $k_\beta = \frac{3}{8}$, $k_\epsilon = \frac{1}{8}$.
$\Omega_t = \left\{ x \in \Delta_{\mathcal{A}} : x_a \geq \frac{1}{At^2}, \forall a \in \mathcal{A} \right\}$.
2: **Initialization::** $x_1 = \frac{1}{A}\mathbf{1}$.
3: **for** $t = 1, 2, \ldots$ **do**
4:     Sample $a_t \sim x_t$, and receive $\sigma_t \in [0, 1]$ with $\mathbb{E}[\sigma_t] = G_{a_t, b_t}$.
5:     Compute $g_t$ where $g_{t,a} = \frac{\mathbf{1}[a_t=a]\sigma_t}{x_{t,a} + \beta_t} + \epsilon_t \ln x_{t,a}, \forall a \in \mathcal{A}$.
6:     Update $x_{t+1} \leftarrow \operatorname{argmin}_{x \in \Omega_{t+1}} \left\{ x^\top g_t + \frac{1}{\eta_t} \mathrm{KL}(x, x_t) \right\}$.
7: **end for**

---

The algorithm is similar to the EXP3-IX algorithm by [Neu15] that achieves a high-probability regret bound for adversarial multi-armed bandits, but with several modifications. First (and most

importantly), in addition to the standard loss estimators used in [Neu15], we add another negative term $\epsilon_t \ln x_{t,a}$ to the loss estimator of action $a$ (see Line 5). This is equivalent to the entropy regularization approach in, e.g., [CWC21, CMZ21], since the gradient of the negative entropy $-\phi(x_t)$ is $(\ln x_{t,a} + 1)_{a \in \mathcal{A}}$ and the constant 1 takes no effect in Line 6. Like [CWC21, CMZ21], the entropy regularization drives last-iterate convergence; however, while their results require full-information feedback, our result holds in the bandit feedback setting. The second difference is that instead of choosing the players' strategies in the full probability simplex $\Delta_{\mathcal{A}}$, our algorithm chooses from $\Omega_t$, a subset of $\Delta_{\mathcal{A}}$ where every coordinate is lower bounded by $\frac{1}{At^2}$. The third is the choices of the learning rate $\eta_t$, clipping factor $\beta_t$, and the amount of regularization $\epsilon_t$. The main result of this section is the following last-iterate convergence rate of Algorithm 1.

**Theorem 1** (Last-Iterate Convergence Rate). *Algorithm 1 guarantees with probability at least $1 - \mathcal{O}(\delta)$, for any $t \geq 1$,*

$$
\max_{x,y \in \Delta_A} \left( x_t^\top G y - x^\top G y_t \right) = \mathcal{O}\left( \sqrt{A} \ln^{3/2}(At/\delta) t^{-\frac{1}{8}} \right).
$$

Algorithm 1 also guarantees $\mathcal{O}(t^{-\frac{1}{8}})$ regret even when the other player is adversarial. If we only target at an *expected* bound instead of a high-probability bound, the last-iterate convergence rate can be improved to $\mathcal{O}(\sqrt{A} \ln^{3/2}(At) t^{-\frac{1}{6}})$. The details are provided in Appendix C.

## 4.1 Analysis Overview

We define a regularized zero-sum game with loss function $f_t(x,y) = x^\top G y - \epsilon_t \phi(x) + \epsilon_t \phi(y)$ over domain $\Omega_t \times \Omega_t$, and denote by $z_t^\star = (x_t^\star, y_t^\star)$ its unique Nash equilibrium since $f_t$ is strongly convex-strongly concave. The regularized game is a slight perturbation of the original matrix game $G$ over a smaller domain $\Omega_t \times \Omega_t$, and we prove that $z_t^\star$ is an $\mathcal{O}(\epsilon_t)$-approximate Nash equilibrium of the original matrix game $G$ (Lemma 9). Therefore, it suffices to bound $\mathrm{KL}(z_t^\star, z_t)$ since the duality gap of $z_t$ is at most $\mathcal{O}(\sqrt{\mathrm{KL}(z_t^\star, z_t)} + \epsilon_t)$.

**Step 1: Single-Step Analysis**  We start with a single-step analysis of Algorithm 1, which shows:

$$
\mathrm{KL}(z_{t+1}^\star, z_{t+1}) \leq (1 - \eta_t \epsilon_t) \mathrm{KL}(z_t^\star, z_t) + \underbrace{20\eta_t^2 A \ln^2(At) + 2\eta_t^2 A \lambda_t}_{\text{instability penalty}} + \underbrace{\eta_t \xi_t + \eta_t \zeta_t}_{\text{estimation error}} + v_t
$$

where we define $v_t = \mathrm{KL}(z_{t+1}^\star, z_{t+1}) - \mathrm{KL}(z_t^\star, z_{t+1})$ (see Appendix B for definitions of $\lambda_t, \xi_t, \zeta_t$) The instability penalty comes from some local-norm of the gradient estimator $g_t$. The estimation error comes from the bias between the gradient estimator $g_t$ and the real gradient $G y_t$. We pay the last term $v_t$ since the Nash equilibrium $z_t^*$ of the regularized game $f_t$ is changing over time.

**Step 2: Strategy Convergence to NE of the Regularized Game**  Expanding the above recursion up to $t_0$, we get

$$
\mathrm{KL}(z_{t+1}^\star, z_{t+1}) \leq \mathcal{O}\Big( \underbrace{\sum_{i=1}^t w_t^i \eta_i^2}_{\textbf{term}_1} + 2A \underbrace{\sum_{i=1}^t w_t^i \eta_i^2 \lambda_i}_{\textbf{term}_2} + \underbrace{\sum_{i=1}^t w_t^i \eta_i \xi_i}_{\textbf{term}_3} + \underbrace{\sum_{i=1}^t w_t^i \eta_i \zeta_i}_{\textbf{term}_4} + \underbrace{\sum_{i=1}^t w_t^i v_i}_{\textbf{term}_5} \Big), \quad (1)
$$

where $w_t^i \triangleq \prod_{j=i+1}^t (1 - \eta_j \epsilon_j)$. To upper bound **term₁**-**term₄**, we apply careful sequence analysis (Appendix A.1) and properties of the EXP3-IX algorithm with changing step size (Appendix A.2). The analysis of **term₅** uses Lemma 13, which states $v_t = \mathrm{KL}(z_{t+1}^\star, z_{t+1}) - \mathrm{KL}(z_t^\star, z_{t+1}) \leq \mathcal{O}(\ln(At) \|z_{t+1}^\star - z_t^\star\|_1) = \mathcal{O}(\frac{\ln^2(At)}{t})$ and is slightly involved as $\Omega_t$ and $\epsilon_t$ are both changing. With these steps, we conclude that with probability at least $1 - \mathcal{O}(\delta)$, $\mathrm{KL}(z_t^\star, z_t) = \mathcal{O}\left( A \ln^3(At/\delta) t^{-\frac{1}{4}} \right)$.

## 5 Irreducible Markov Games

We now extend our results on matrix games to two-player zero-sum Markov games. Similarly to many previous works, our first result makes the assumption that the Markov game is *irreducible* with bounded travel time between any pair of states. The assumption is formally stated below:

**Assumption 1** (Irreducible Game). *We assume that under any pair of stationary policies of the two players, and any pair of states $s, s'$, the expected time to reach $s'$ from $s$ is upper bounded by $L$.*

We propose Algorithm 2 for uncoupled learning in irreducible two-player zero-sum games, which is closely related to the Nash-V algorithm by [BJY20], but with additional entropy regularization. It can also be seen as players using Algorithm 1 on each state $s$ to update the policies $(x_t^s, y_t^s)$ whenever state $s$ is visited, but with $\sigma_t + \gamma V_t^{s_{t+1}}$ as the observed loss to construct loss estimators. Importantly, $V_1^s, V_2^s, \ldots$ is a slowly changing sequence of value estimations that ensures stable policy updates [BJY20, WLZL21a, SZL$^+$21]. Note that in Algorithm 2, the updates of $V_t^s$ only use players' local information (Line 8).

---

**Algorithm 2** Irreducible Markov Game

---

1: **Define:** $\eta_t = (1-\gamma)t^{-k_\eta}$, $\beta_t = t^{-k_\beta}$, $\epsilon_t = \frac{1}{1-\gamma}t^{-k_\epsilon}$, $\alpha_t = t^{-k_\alpha}$ with $k_\alpha, k_\epsilon, k_\beta, k_\eta \in (0,1)$,
   $\Omega_t = \left\{ x \in \Delta_\mathcal{A} : x_a \geq \frac{1}{At^2}, \forall a \in \mathcal{A} \right\}$.
2: **Initialization:** $x_1^s \leftarrow \frac{1}{A}\mathbf{1}$, $n_1^s \leftarrow 0$, $V_1^s \leftarrow \frac{1}{2(1-\gamma)}$, $\forall s$.
3: **for** $t = 1, 2, \ldots,$ **do**
4:    $\tau = n_{t+1}^{s_t} \leftarrow n_t^{s_t} + 1$ (the number of visits to state $s_t$ up to time $t$).
5:    Draw $a_t \sim x_t^{s_t}$, observe $\sigma_t \in [0,1]$ with $\mathbb{E}[\sigma_t] = G_{a_t, b_t}^{s_t}$, and observe $s_{t+1} \sim P_{a_t, b_t}^{s_t}(\cdot)$.
6:    Compute $g_t$ where $g_{t,a} = \frac{\mathbf{1}[a_t = a](\sigma_t + \gamma V_t^{s_{t+1}})}{x_{t,a}^{s_t} + \beta_\tau} + \epsilon_\tau \ln x_{t,a}^{s_t}$, $\forall a \in \mathcal{A}$.
7:    Update $x_{t+1}^{s_t} \leftarrow \operatorname{argmin}_{x \in \Omega_{\tau+1}} \left\{ x^\top g_t + \frac{1}{\eta_\tau}\mathrm{KL}(x, x_t^{s_t}) \right\}$.
8:    Update $V_{t+1}^{s_t} \leftarrow (1-\alpha_\tau)V_t^{s_t} + \alpha_\tau (\sigma_t + \gamma V_t^{s_{t+1}})$.
9:    For all $s \neq s_t$, $x_{t+1}^s \leftarrow x_t^s$, $n_{t+1}^s \leftarrow n_t^s$, $V_{t+1}^s \leftarrow V_t^s$.
10: **end for**

---

**Comparison to Previous Works** Although Algorithm 2 shares similarity with previous works that also use entropy regularization, we believe that both the design and the analysis of our algorithm are novel and non-trivial. To the best of our knowledge, all previous entropy regularized two-player zero-sum Markov game algorithms are coupled (e.g., [CWC21, CMZ21, CCDX23]), while ours is the first that achieves uncoupledness under entropy regularization. We further discuss this by comparing our algorithm to those in [CCDX23], highlighting the new technical challenges we encounter.

The entropy-regularized OMWU algorithm in [CCDX23] is tailored to the full-information setting. Moreover, in the value function update step both players need to know the entropy value of the other player's policy, which is unnatural. Indeed, the authors explicitly present the removal of this information sharing as an open question. We answer this open question affirmatively by giving a fully decentralized algorithm for zero-sum Markov games with provable last-iterate convergence rates. In Algorithm 2 (Line 8), the update of the value function $V$ is simple and does not require any entropy information: $V_{t+1}^{s_t} \leftarrow (1-\alpha_\tau)V_t^{s_t} + \alpha_\tau (\sigma_t + \gamma V_t^{s_{t+1}})$. This modification results in a discrepancy between the policy update and the value update. While the policy now incorporates a regularization term, the value function does not. Such a mismatch is unprecedented in earlier studies and necessitates a non-trivial approach to resolve. Additionally, Algorithm 2 operates on bandit feedback instead of full-information feedback, presenting further technical challenges.

Algorithm 2 also offers improvement over the uncoupled algorithm of [WLZL21a]. The algorithm of [WLZL21a] requires coordinated policy update where the players interact with each other using the current policy for several iterations to get an approximately accurate gradient (the number of iterations required depends on $L$ as defined in Assumption 1), and then simultaneously update the policy pair on all states. We do not require such unnatural coordination between the players or prior knowledge on $L$.

Our main result is the following theorem on the last-iterate convergence rate of Algorithm 2.

**Theorem 2** (Last-Iterate Convergence Rate). *For any $\varepsilon, \delta > 0$, Algorithm 2 with $k_\alpha = \frac{9}{9+\varepsilon}$, $k_\epsilon = \frac{1}{9+\varepsilon}$, $k_\beta = \frac{3}{9+\varepsilon}$, and $k_\eta = \frac{5}{9+\varepsilon}$ guarantees, with probability at least $1 - \mathcal{O}(\delta)$, for any time $t \geq 1$,*

$$\max_{s,x,y} \left( V_{x_t, y}^s - V_{x, y_t}^s \right) \leq \mathcal{O}\left( \frac{AL^{2+1/\varepsilon}\ln^{4+1/\varepsilon}(SAt/\delta)\ln^{1/\varepsilon}(t/(1-\gamma))}{(1-\gamma)^{2+1/\varepsilon}} \cdot t^{-\frac{1}{9+\varepsilon}} \right).$$

## 5.1 Analysis Overview

We introduce some notations for simplicity. We denote by $\mathbb{E}_{s'\sim P^s}[V_t^{s'}]$ the $A \times A$ matrix such that $(\mathbb{E}_{s'\sim P^s}[V_t^{s'}])_{a,b} = \mathbb{E}_{s'\sim P^s_{a,b}}[V_t^{s'}]$. Let $t_\tau(s)$ be the $\tau$-th time the players visit state $s$, and define $\hat{x}_\tau^s = x_{t_\tau(s)}^s$ and $\hat{y}_\tau^s = y_{t_\tau(s)}^s$. Then, define the regularized game for each state $s$ via the loss function $f_\tau^s(x,y) = x^\top (G^s + \gamma \mathbb{E}_{s'\sim P^s}[V_{t_\tau(s)}^{s'}])y - \epsilon_\tau \phi(x) + \epsilon_\tau \phi(y)$. Furthermore, let $\hat{z}_{\tau\star}^s = (\hat{x}_{\tau\star}^s, \hat{y}_{\tau\star}^s)$ be the equilibrium of $f_\tau^s(x,y)$ over $\Omega_\tau \times \Omega_\tau$. In the following analysis, we fix some $t \geq 1$.

**Step 1: Policy Convergence to NE of Regularized Game**  Using similar techniques to Step 1 and Step 2 in the analysis of Algorithm 1, we can upper bound $\text{KL}(\hat{z}_{\tau+1\star}^s, \hat{z}_{\tau+1}^s)$ like Eq. (1) with similar subsequent analysis for **term**$_1$-**term**$_4$. The analysis for **term**$_5$ where $v_i^s = \text{KL}(\hat{z}_{i+1\star}^s, \hat{z}_{i+1}^s) - \text{KL}(\hat{z}_{i\star}^s, \hat{z}_{i+1}^s)$ is more challenging compared to the matrix game case since here $V_{t_i(s)}^s$ is changing between two visits to state $s$. To handle this term, we leverage the following facts for any $s'$: (1) the irreducibility assumption ensures that $t_{i+1}(s) - t_i(s) \leq \mathcal{O}(L\ln(St/\delta))$ thus the number of updates of the value function at state $s'$ is bounded; (2) until time $t_i(s) \geq i$, state $s'$ has been visited at least $\Omega(\frac{i}{L\ln(St/\delta)})$ times thus each change of the value function between $t_i(s)$ and $t_{i+1}(s)$ is at most $\mathcal{O}((\frac{i}{L\ln(St/\delta)})^{-k_\alpha})$. With these arguments, we can bound **term**$_5$ by $\mathcal{O}\left(\ln^4(SAt/\delta)L\tau^{-k_\alpha+k_\eta+2k_\epsilon}\right)$. Overall, we have the following policy convergence of NE of the regularized game (Lemma 17): $\text{KL}(\hat{z}_{\tau\star}^s, \hat{z}_\tau^s) \leq \mathcal{O}\left(A\ln^4(SAt/\delta)L\tau^{-k_\sharp}\right)$, where $k_\sharp = \min\{k_\beta - k_\epsilon, k_\eta - k_\beta, k_\alpha - k_\eta - 2k_\epsilon\}$.

**Step 2: Value Convergence**  Unlike matrix games, policy convergence to NE of the regularized game is not enough for convergence in duality gap. We also need to bound $|V_t^s - V_\star^s|$ since the regularized game is defined using $V_t^s$, the value function maintained by the algorithm, instead of the minimax game value $V_\star^s$. We use the following weighted regret quantities as a proxy: $\text{Reg}_\tau^s \triangleq \max_{x,y}\left(\sum_{i=1}^\tau \alpha_\tau^i\left(f_i^s(\hat{x}_i^s, \hat{y}_i^s) - f_i^s(x^s, \hat{y}_i^s)\right), \sum_{i=1}^\tau \alpha_\tau^i\left(f_i^s(\hat{x}_i^s, y_i^s) - f_i^s(\hat{x}_i^s, \hat{y}_i^s)\right)\right)$, where $\alpha_\tau^i = \alpha_i \prod_{j=i+1}^\tau(1-\alpha_j)$. We can upper bound the weighted regret $\text{Reg}_\tau^s$ using a similar analysis as in Step 1 (Lemma 19). We then show a contraction for $|V_{t_\tau(s)}^s - V_\star^s|$ with the weighted regret quantities: $|V_{t_\tau(s)}^s - V_\star^s| \leq \gamma \sum_{i=1}^\tau \alpha_\tau^i \max_{s'} |V_{t_i(s)}^{s'} - V_\star^{s'}| + \tilde{\mathcal{O}}(\epsilon_\tau + \text{Reg}_\tau^s)$. This leads to the following convergence of $V_t^s$ (Lemma 20): $|V_t^s - V_\star^s| \leq \tilde{\mathcal{O}}(t^{-k_*})$, where $k_* = \min\{k_\eta, k_\beta, k_\alpha - k_\beta, k_\epsilon\}$.

**Obtaining Last-Iterate Convergence Rate**  Fix any $t$ and let $\tau$ be the number of visits to $s$ before time $t$. So far we have shown (1) policy convergence of $\text{KL}(\hat{z}_{\tau\star}^s, \hat{z}_\tau^s)$ in the regularized game; (2) and value convergence of $|V_t^s - V_\star^s|$. Using the fact that the regularized game is at most $\mathcal{O}(\epsilon_\tau + |V_t^s - V_\star^s|)$ away from the minimax game martrix $Q^\star$ and appropriate choices of parameters proves Theorem 2.

## 6 General Markov Games

In this section, we consider general two-player zero-sum Markov games without Assumption 1. We propose Algorithm 3, an uncoupled learning algorithm that handles exploration and has path convergence rate. Compared to Algorithm 2, the update of value function in Algorithm 3 uses a bonus term $\mathsf{bns}_\tau$ based on the optimism principle to handle exploration.

Theorem 3 below implies that we can achieve $\frac{1}{t}\sum_{\tau=1}^t \max_{x,y}(x_\tau^{s_\tau\top} Q_\star^{s_\tau} y^{s_\tau} - x^{s_\tau\top} Q_\star^{s_\tau} y_\tau^{s_\tau}) = \mathcal{O}(t^{-\frac{1}{10}})$ path convergence rate if we use the doubling trick to tune down $u$ at a rate of $t^{-\frac{1}{10}}$.

**Theorem 3.** *For any $u \in \left[0, \frac{1}{1-\gamma}\right]$ and $T \geq 1$, there exists a proper choice of parameters $\epsilon, \beta, \eta$ such that Algorithm 3 guarantees with probability at least $1 - \mathcal{O}(\delta)$,*

$$\sum_{t=1}^T \mathbf{1}\left[\max_{x,y}\left(x_t^{s_t\top} Q_\star^{s_t} y^{s_t} - x^{s_t\top} Q_\star^{s_t} y_t^{s_t}\right) > u\right] \leq \mathcal{O}\left(\frac{S^2 A^3 \ln^{20}(SAT/\delta)}{u^9(1-\gamma)^{16}}\right). \qquad (2)$$

---

**Algorithm 3** General Markov Game

---

1: **Input:** $\eta \leq \beta \leq \epsilon$ and $T$.
2: **Define:** $\Omega = \left\{ x \in \Delta_{\mathcal{A}} : x_a \geq \frac{1}{AT}, \forall a \in \mathcal{A} \right\}$, $\alpha_\tau = \frac{H+1}{H+\tau}$, where $H = \frac{\ln(T)}{1-\gamma}$.
   $\mathsf{bns}_\tau = \kappa A \ln^2(SAT/\delta)(\beta + \eta^{-1}\alpha_\tau)/(1-\gamma)^2$ for a sufficiently large absolute constant $\kappa > 0$
3: **Initialization:** $\underline{V}_1^s, n_1^s \leftarrow 0, x_1^s \leftarrow \frac{1}{A}\mathbf{1}, \forall s$.
4: **for** $t = 1, 2, \ldots,$ **do**
5:     $\tau = n_{t+1}^{s_t} \leftarrow n_t^{s_t} + 1$.
6:     Sample $a_t \sim x_t^{s_t}$, observe $\sigma_t \in [0, 1]$ with $\mathbb{E}[\sigma_t] = G_{a_t,b_t}^{s_t}$, and observe $s_{t+1} \sim P_{a_t,b_t}^{s_t}(\cdot)$.
7:     Compute $g_t$ where $g_{t,a} = \frac{\mathbf{1}[a_t=a]\left(\sigma_t + \gamma \underline{V}_t^{s_{t+1}}\right)}{x_{t,a}^{s_t} + \beta} + \epsilon \ln x_{t,a}^{s_t}, \forall a \in \mathcal{A}$.
8:     Update $x_{t+1}^{s_t} \leftarrow \mathrm{argmin}_{x \in \Omega} \left\{ x^\top g_t + \frac{1}{\eta}\mathrm{KL}(x, x_t^{s_t}) \right\}$.
9:     Update $\underset{\sim}{V}_{t+1}^{s_t} \leftarrow (1-\alpha_\tau)\underline{V}_t^{s_t} + \alpha_\tau\left(\sigma_t + \gamma\underline{V}_t^{s_{t+1}} - \mathsf{bns}_\tau\right)$ and $\underline{V}_{t+1}^{s_t} \leftarrow \max\left\{ \underset{\sim}{V}_{t+1}^{s_t}, 0 \right\}$.
10:    For all $s \neq s_t$, $x_{t+1}^s \leftarrow x_t^s$, $\underline{V}_{t+1}^s \leftarrow \underline{V}_t^s$, $\underset{\sim}{V}_{t+1}^s \leftarrow \underset{\sim}{V}_t^s$, $n_{t+1}^s \leftarrow n_t^s$.
11: **end for**

---

## 6.1 Path Convergence

Path convergence has multiple meaningful game-theoretic implications. By definition, It implies that frequent visits to a state bring players' policies closer to equilibrium, leading to both players using near-equilibrium policies for all but $o(T)$ number of steps over time.

Path convergence also implies that both players have no regret compared to the game value $V_\star^s$, which has been considered and motivated in previous works such as [BT02, TWYS20]. To see this, we apply the results to the *episodic* setting, where in every step, with probability $1 - \gamma$, the state is redrawn from $s \sim \rho$ for some initial distribution $\rho$. If the learning dynamics enjoys path convergence, then $\mathbb{E}[\sum_{t=1}^T x_t^{s_t^\top} G^{s_t} y_t^{s_t}] = (1-\gamma)\mathbb{E}_{s\sim\rho}[V_\star^s]T \pm o(T)$. Hence the one-step average reward is $(1 - \gamma)\mathbb{E}_{s\sim\rho}[V_\star^s]$ and both players have no regret compared to the game value. A more important implication of path convergence is that it guarantees stability of players' policies, while cycling behaviour is inevitable for any FTRL-type algorithms even in zero-sum matrix games [MPP18, BP18]. We defer the proof and more discussion of path convergence to Appendix F.

Finally, we remark that our algorithm is built upon Nash V-learning [BJY20], so it inherits properties of Nash V-learning, e.g., one can still output near-equilibrium policies through policy averaging [JLWY21], or having no regret compared to the game value when competing with an arbitrary opponent [TWYS20]. We demonstrate extra benefits brought by entropy regularization regarding the stability of the dynamics.

## 6.2 Analysis Overview of Theorem 3

For general Markov games, it no longer holds that every state $s$ is visited often, and thus the analysis is much more challenging. We first define two regularized games based on $\underline{V}_t^s$ and the corresponding quantity $\overline{V}_t^s$ for the $y$-player. Define $t_\tau(s), \hat{x}_\tau^s, \hat{y}_\tau^s$ the same way as in the previous section. Then define $\underline{f}_\tau^s(x,y) \triangleq x^\top(G^s + \gamma\mathbb{E}_{s'\sim P^s}[\underline{V}_{t_\tau(s)}^{s'}])y - \epsilon\phi(x) + \epsilon\phi(y)$, $\overline{f}_\tau^s(x,y) \triangleq x^\top(G^s + \gamma\mathbb{E}_{s'\sim P^s}[\overline{V}_{t_\tau(s)}^{s'}])y - \epsilon\phi(x) + \epsilon\phi(y)$ and denote $J_t = \max_{x,y}(x_t^{s_t^\top}(G^{s_t} + \gamma\mathbb{E}_{s'\sim P^{s_t}}[\overline{V}_t^{s'}])y^{s_t} - x_t^{s_t^\top}(G^s + \gamma\mathbb{E}_{s'\sim P^{s_t}}[\underline{V}_t^{s'}])y_t^{s_t})$. We first bound the "path duality gap" as follows

$$\max_{x,y}\left( x_t^{s_t^\top} Q_\star^{s_t} y^s - x^{s_t^\top} Q_\star^{s_t} y_t^s \right) \leq J_t + \mathcal{O}\left( \max_{s'}\left( V_\star^{s'} - \overline{V}_t^{s'}, \underline{V}_t^{s'} - V_\star^{s'} \right) \right). \tag{3}$$

**Value Convergence: Bounding $\underline{V}_t^s - V_\star^s$ and $V_\star^s - \overline{V}_t^s$** This step is similar to Step 2 in the analysis of Algorithm 2. We first show an upper bound of the weighted regret (Lemma 23): $\sum_{i=1}^\tau \alpha_\tau^i(\underline{f}_i^s(\hat{x}_i^s, \hat{y}_i^s) - \underline{f}_i^s(x^s, \hat{y}_i^s)) \leq \frac{1}{2}\mathsf{bns}_\tau$, where $\alpha_\tau^i = \alpha_i\prod_{j=i+1}^\tau(1 - \alpha_j)$. Note that the value function $\underline{V}_t^s$ is updated using $\sigma_t + \gamma\underline{V}_t^{s_t+1} - \mathsf{bns}_\tau$. Thus when relating $|\underline{V}_t^s - V_\star^s|$ to the regret, the regret term and the bonus term cancel out and we get $\underline{V}_t^s \leq V_\star^s + \mathcal{O}(\frac{\epsilon\ln(AT)}{1-\gamma})$ (Lemma 26). The analysis for $V_\star^s - \overline{V}_t^s$ is symmetric. By proper choice of $\epsilon$, both terms are bounded by $\frac{1}{8}u$. Combining

the above with Eq. (3), we can upper bound the left-hand side of the desired inequality Eq. (2) by $\sum_{t=1}^{T} \mathbf{1}\left[J_t \geq \frac{3}{4}u\right]$, which is further upper bounded in Eq. (29) by

$$\sum_s \sum_{\tau=1}^{n_{T+1}(s)} \mathbf{1}\left[\max_y \overline{f}_\tau^s(\hat{x}_\tau^s, y^s) - \overline{f}_\tau^s(\hat{z}_\tau^s) \geq \frac{u}{8}\right] + \sum_s \sum_{\tau=1}^{n_{T+1}(s)} \mathbf{1}\left[\underline{f}_\tau^s(\hat{z}_\tau^s) - \min_x \underline{f}_\tau^s(x^s, \hat{y}_\tau^s) \geq \frac{u}{8}\right]$$

$$+ \sum_{t=1}^{T} \mathbf{1}\left[x_t^{s_t\top}\left(\gamma \mathbb{E}_{s'\sim P^{s_t}}\left[\overline{V}_t^{s'} - \underline{V}_t^{s'}\right]\right) y_t^{s_t} \geq \frac{u}{4}\right]. \tag{4}$$

**Policy Convergence to NE of Regularized Games**  To bound the first two terms, we show convergence of the policy $(\hat{x}_\tau^s, \hat{y}_\tau^s)$ to Nash equilibria of both games $\underline{f}_\tau^s$ and $\overline{f}_\tau^s$. To this end, fix any $p \in [0,1]$, we define $f_\tau^s = p\underline{f}_\tau^s + (1-p)\overline{f}_\tau^s$ and let $\hat{z}_{\tau\star}^s = (\hat{x}_{\tau\star}^s, \hat{y}_{\tau\star}^s)$ be the equilibrium of $f_\tau^s(x, y)$. The analysis is similar to previous algorithms where we first conduct single-step analysis (Lemma 22) and then carefully bound the weighted recursive terms. We show in Lemma 27 that for any $0 < \epsilon' \leq 1$: $\sum_s \sum_{\tau=1}^{n_{T+1}(s)} \mathbf{1}\left[\mathrm{KL}(\hat{z}_{\tau\star}^s, \hat{z}_\tau^s) \geq \epsilon'\right] \leq \mathcal{O}(\frac{S^2 A \ln^5(SAT/\delta)}{\eta^2 \epsilon'(1-\gamma)^3})$. This proves policy convergence: the number of iterations where the policy is far away from Nash equilibria of the regularized games is bounded, which can then be translated to upper bounds on the first two terms.

**Value Convergence: Bounding $|\overline{V}_t^s - \underline{V}_t^s|$**  It remains to bound the last term in Eq. (4). Define $c_t = \mathbf{1}[x_t^{s_t}(\mathbb{E}_{s'\sim P^{s_t}}[\overline{V}_t^{s'} - \underline{V}_t^{s'}])y_t^{s_t} \geq \tilde{\epsilon}]$ where $\tilde{\epsilon} = \frac{u}{4}$. Then we only need to bound $C \triangleq \sum_{t=1}^{T} c_t$. We use the weighted sum $P_T \triangleq \sum_{t=1}^{T} c_t x_t^{s_t}(\mathbb{E}_{s'\sim P^{s_t}}[\overline{V}_t^{s'} - \underline{V}_t^{s'}])y_t^{s_t}$ as a proxy. On the one hand, $P_T \geq C\tilde{\epsilon}$. On the other hand, in Lemma 25, by recursively tracking the update of the value function and carefully choosing $\eta$ and $\beta$, we upper bound $P_T$ by $\leq \frac{C\tilde{\epsilon}}{2} + \mathcal{O}(\frac{AS \ln^4(AST/\delta)}{\eta(1-\gamma)^3})$. Combining the upper and lower bound of $P_T$ gives $C \leq \mathcal{O}(\frac{AS \ln^4(AST/\delta)}{\eta u(1-\gamma)^3})$ (Corollary 2). Plugging appropriate choices of $\epsilon$, $\eta$, and $\beta$ in the above bounds proves Theorem 3 (see Appendix E).

## 7 Conclusion and Future Directions

In this work, we study decentralized learning in two-player zero-sum Markov games with bandit feedback. We propose the first uncoupled and convergent algorithms with non-asymptotic last-iterate convergence rates for matrix games and irreducible Markov games, respectively. We also introduce a novel notion of path convergence and provide algorithm with path convergence in Markov games without any assumption on the dynamics. Previous results either focus on average-iterate convergence or require stronger feedback/coordination or lack non-asymptotic convergence rates. Our results contribute to the theoretical understanding of the practical success of regularization and last-iterate convergence in multi-agent reinforcement learning.

Settling the optimal last-iterate convergence rate that is achievable by uncoupled learning dynamics is an important open question. The following directions are promising towards closing the gap between current upper bounds $\mathcal{O}(T^{-1/8})$ and $O(T^{-1/(9+\varepsilon)})$ and lower bound $\Omega(T^{-\frac{1}{2}})$, The impossibility result by [MPS20] demonstrates that certain algorithms with $\mathcal{O}(\sqrt{T})$ regret diverge in last-iterate. Their result indicates that the current $\Omega(\frac{1}{\sqrt{T}})$ lower bound on convergence rate may not be tight. On the other hand, our algorithms provides insights and useful templates to potential improvements on the upper bound. For instance, instead of using EXP3-IX update, adapting optimistic policy update or other accelerated first-order methods to the bandit feedback setting is an interesting future direction.

**Acknowledgement**  We thank Chanwoo Park and Kaiqing Zhang for pointing out a mistake in our previous proof. We also thank the anonymous reviewers for their constructive feedback. HL is supported by NSF Award IIS-1943607 and a Google Research Scholar Award.

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

## A   Auxiliary Lemmas

### A.1   Sequence Properties

**Lemma 1.** *Let* $0 < h < 1$, $0 \le k \le 2$, *and let* $t \ge \left( \frac{24}{1-h} \ln \frac{12}{1-h} \right)^{\frac{1}{1-h}}$. *Then*

$$\sum_{i=1}^{t} \left( i^{-k} \prod_{j=i+1}^{t} (1 - j^{-h}) \right) \le 9 \ln(t) t^{-k+h}.$$

*Proof.* Define

$$s \triangleq \left\lceil (k+1) t^h \ln t \right\rceil$$

We first show that $s \le \frac{t}{2}$. Suppose not, then we have

$$(k+1) t^h \ln t > \frac{t}{2} - 1 \ge \frac{t}{4} \qquad\qquad \text{(because } t \ge 12 > 4\text{)}$$

and thus $t^{1-h} < 4(k+1) \ln t \le 12 \ln t$. However, by the condition for $t$ and Lemma 3, it holds that $t^{1-h} \ge 12 \ln t$, which leads to contradiction.

Then the sum can be decomposed as

$$\sum_{i=1}^{t-s} i^{-k} \prod_{j=i+1}^{t} (1 - j^{-h}) + \sum_{i=t-s+1}^{t} i^{-k} \prod_{j=i+1}^{t} (1 - j^{-h})$$

$$\leq t \times (1 - t^{-h})^s + s \, (t - s + 1)^{-k}$$

$$\leq t \times (e^{-t^{-h}})^s + s \times \left(\frac{t}{2}\right)^{-k}$$

$$\leq t \times e^{-(k+1)\ln t} + s \times 2^k \times t^{-k}$$

$$\leq t^{-k} + \left((k+1)t^h \ln t + 1\right) \times 2^k \times t^{-k}$$

$$\leq 9 \ln(t) t^{-k+h}.$$

$\square$

**Lemma 2.** *Let* $0 < h < 1$, $0 \leq k \leq 2$, *and let* $t \geq \left(\frac{24}{1-h} \ln \frac{12}{1-h}\right)^{\frac{1}{1-h}}$. *Then*

$$\max_{1 \leq i \leq t} \left(i^{-k} \prod_{j=i+1}^{t} (1 - j^{-h})\right) \leq 4t^{-k}.$$

*Proof.*

$$\max_{\frac{t}{2} \leq i \leq t} \left(i^{-k} \prod_{j=i+1}^{t} (1 - j^{-h})\right) \leq \left(\frac{t}{2}\right)^{-k} \leq 2^2 t^{-k} = 4t^{-k}$$

$$\max_{1 \leq i \leq \frac{t}{2}} \left(i^{-k} \prod_{j=i+1}^{t} (1 - j^{-h})\right) \leq \left(1 - t^{-h}\right)^{\frac{t}{2}} \leq \left(\exp\left(-t^{-h}\right)\right)^{\frac{t}{2}} = \exp\left(-\frac{1}{2} t^{1-h}\right)$$

$$\overset{(a)}{\leq} \exp\left(-\frac{1}{2} \times 12 \ln t\right) = \frac{1}{t^6} \leq t^{-k}.$$

where in $(a)$ we use Lemma 3. Combining the two inequalities finishes the proof. $\square$

**Lemma 3.** *Let* $0 < h < 1$ *and* $t \geq \left(\frac{24}{1-h} \ln \frac{12}{1-h}\right)^{\frac{1}{1-h}}$. *Then* $t^{1-h} \geq 12 \ln t$.

*Proof.* By the condition, we have

$$t^{1-h} \geq 2 \times \frac{12}{1-h} \ln \frac{12}{1-h}.$$

Applying Lemma 12, we get

$$t^{1-h} \geq \frac{12}{1-h} \ln(t^{1-h}) = 12 \ln t.$$

$\square$

**Lemma 4** (Lemma A.1 of [SSBD14]). *Let* $a > 0$. *Then* $x \geq 2a \ln(a) \Rightarrow x \geq a \ln(x)$.

**Lemma 5** (Freedman's Inequality). *Let* $\mathcal{F}_0 \subset \mathcal{F}_1 \subset \cdots \subset \mathcal{F}_n$ *be a filtration, and* $X_1, \ldots, X_n$ *be real random variables such that* $X_i$ *is* $\mathcal{F}_i$-measurable, $\mathbb{E}[X_i|\mathcal{F}_{i-1}] = 0$, $|X_i| \leq b$, *and* $\sum_{i=1}^{n} \mathbb{E}[X_i^2|\mathcal{F}_{i-1}] \leq V$ *for some fixed* $b > 0$ *and* $V > 0$. *Then with probability at least* $1 - \delta$,

$$\sum_{i=1}^{n} X_i \leq 2\sqrt{V \log(1/\delta)} + b \log(1/\delta).$$

## A.2 Properties Related to EXP3-IX

In Lemma 6 and Lemma 7, we assume that $\mathcal{F}_0 \subset \mathcal{F}_1 \subset \mathcal{F}_2 \subset \cdots$ is a filtration, and assume that $x_i$, $\ell_i$ are $\mathcal{F}_{i-1}$-measurable, where $x_i \in \Delta_A$, $\ell_i \in [0,1]^A$. Besides, $a_i \in [A]$ and $\sigma_i$ are $\mathcal{F}_i$-measurable with $\mathbb{E}[a_i = a | \mathcal{F}_{i-1}] = x_{i,a}$ and $\mathbb{E}[\sigma_i | \mathcal{F}_{i-1}] = \ell_i$. Define $\widehat{\ell}_{i,a} = \frac{\sigma_{i,a} \mathbf{1}[a_i = a]}{x_{i,a} + \beta_i}$ where $\beta_i$ is non-increasing.

**Lemma 6** (Lemma 20 of [BJY20]). *Let $c_1, c_2, \ldots, c_t$ be fixed positive numbers. Then with probability at least $1 - \delta$,*

$$\sum_{i=1}^{t} c_i \left\langle x_i, \ell_i - \widehat{\ell}_i \right\rangle = \mathcal{O}\left( A \sum_{i=1}^{t} \beta_i c_i + \sqrt{\ln(A/\delta) \sum_{i=1}^{t} c_i^2} \right).$$

**Lemma 7** (Adapted from Lemma 18 of [BJY20]). *Let $c_1, c_2, \ldots, c_t$ be fixed positive numbers. Then for any sequence $x_1^\star, \ldots, x_t^\star \in \Delta_A$ such that $x_i^\star$ is $\mathcal{F}_{i-1}$-measurable, with probability at least $1 - \delta$,*

$$\sum_{i=1}^{t} c_i \left\langle x_i^\star, \widehat{\ell}_i - \ell_i \right\rangle = \mathcal{O}\left( \max_{i \le t} \frac{c_i \ln(1/\delta)}{\beta_t} \right).$$

*Proof.* Lemma 18 of [BJY20] states that for any sequence of coefficients $w_1, w_2, \cdots, w_t$ such that $w_i \in [0, 2\beta_i]^A$ is $\mathcal{F}_{i-1}$-measurable, we have with probability $1 - \delta$,

$$\sum_{i=1}^{t} c_i \left\langle w_i, \widehat{\ell}_i - \ell_i \right\rangle \le \max_{i \le t} c_i \log(1/\delta).$$

Since $x_i^\star \in \Delta_A$ and $\beta_i$ is decreasing, we know $2\beta_t \cdot x_i^\star \in [0, 2\beta_i]$. Thus we can apply Lemma 18 of [BJY20] and get with probability $1 - \delta$,

$$\sum_{i=1}^{t} c_i \left\langle x_i^\star, \widehat{\ell}_i - \ell_i \right\rangle = \sum_{i=1}^{t} \frac{c_i}{2\beta_t} \left\langle 2\beta_t \cdot x_i^\star, \widehat{\ell}_i - \ell_i \right\rangle \le \max_{i \le t} \frac{c_i}{\beta_t} \log(1/\delta).$$

$\square$

**Lemma 8** (Lemma 21 of [BJY20]). *Let $c_1, c_2, \ldots, c_t$ be fixed positive numbers. Then with probability at least $1 - \delta$, for all $x^\star \in \Delta_A$,*

$$\sum_{i=1}^{t} c_i \left\langle x^\star, \widehat{\ell}_i - \ell_i \right\rangle = \mathcal{O}\left( \max_{i \le t} \frac{c_i \ln(A/\delta)}{\beta_t} \right).$$

**Lemma 9.** *Let $(x_1, y_1)$ and $(x_2, y_2)$ be equilibria of $f_1(\cdot, \cdot)$ in the domain $\mathcal{Z}_1$ and $f_2(\cdot, \cdot)$ in the domain $\mathcal{Z}_2$ respectively. Suppose that $\mathcal{Z}_1 \subseteq \mathcal{Z}_2$, and that $\sup_{(x,y) \in \mathcal{Z}_1} |f_1(x,y) - f_2(x,y)| \le \epsilon$. Then for any $(x,y) \in \mathcal{Z}_2$,*

$$f_2(x_1, y) - f_2(x, y_1) \le 2\epsilon + 2d \sup_{(\tilde{x}, \tilde{y}) \in \mathcal{Z}_2} \|\nabla f_2(\tilde{x}, \tilde{y})\|_\infty$$

*where $d = \max_{z \in \mathcal{Z}_2} \min_{z' \in \mathcal{Z}_1} \|z - z'\|_1$*

*Proof.* Since $(x_1, y_1)$ is an equilibrium of $f_1$, we have for any $(x', y') \in \mathcal{Z}_1$,

$$f_1(x_1, y') - f_1(x', y_1) \le 0,$$

which implies

$$f_2(x_1, y') - f_2(x', y_1) \le 2\epsilon.$$

For any $(x, y) \in \mathcal{Z}_2$, we can find $(x', y') \in \mathcal{Z}_1$ such that $\|(x, y) - (x', y')\|_1 \le d$. Therefore, for any $(x, y) \in \mathcal{Z}_2$,

$$
\begin{aligned}
&f_2(x_1, y) - f_2(x, y_1) \\
&\le f_2(x_1, y') - f_2(x', y_1) + \|x - x'\|_1 \|\nabla_{\mathsf{x}} f_2(x, y)\|_\infty + \|y - y'\|_1 \|\nabla_{\mathsf{y}} f_2(x, y)\|_\infty \\
&\le 2\epsilon + 2d \sup_{(\tilde{x}, \tilde{y}) \in \mathcal{Z}_2} \|\nabla f_2(\tilde{x}, \tilde{y})\|_\infty.
\end{aligned}
$$

$\square$

## A.3 Markov Games

**Lemma 10** ([WLZL21a]). *For any policy pair $x, y$, the duality gap on a two player zero-sum game can be related to duality gap on individual states:*

$$\max_{s,x',y'} \left( V_{x,y'}^s - V_{x',y}^s \right) \leq \frac{2}{1-\gamma} \max_{s,x',y'} (x^s Q_\star^s y'^s - x'^s Q_\star^s y^s).$$

## A.4 Online Mirror Descent

**Lemma 11.** *Let*

$$x' = \underset{\tilde{x} \in \Omega}{\operatorname{argmin}} \left\{ \sum_{a \in \mathcal{A}} \tilde{x}_a \left( \ell_a + \epsilon_a \ln x_a \right) + \frac{1}{\eta} KL(\tilde{x}, x) \right\}$$

*for some convex set $\Omega \subseteq \Delta_{\mathcal{A}}$, $\ell \in [0, \infty)^A$, and $\epsilon \in [0, \frac{1}{\eta}]^A$. Then*

$$(x - u)^\top (\ell + \epsilon \ln x) \leq \frac{KL(u, x) - KL(u, x')}{\eta} + \eta \sum_{a \in \mathcal{A}} x_a (\ell_a)^2 + \eta \sum_{a \in \mathcal{A}} \epsilon_a^2 \ln^2 x_a.$$

*for any $u \in \Omega$, where $\epsilon \ln x$ denotes the vector $(\epsilon_a \ln x_a)_{a \in \mathcal{A}}$.*

*Proof.* By the standard analysis of online mirror descent, we have for any $u \in \Omega$

$$(x - u)^\top (\ell + \epsilon \ln x) \leq \frac{KL(u, x) - KL(u, x')}{\eta} + (x - x')^\top (\ell + \epsilon \ln x) - \frac{1}{\eta} KL(x', x).$$

Below, we abuse the notation by defining $KL(\tilde{x}, x) = \sum_a (\tilde{x}_a \ln \frac{\tilde{x}_a}{x_a} - \tilde{x}_a + x_a)$ without restricting $\tilde{x}$ to be a probability vector. Then following the analysis in the proof of Lemma 1 of [CLW21], we have

$$(x - x')^\top (\ell + \epsilon \ln x) - \frac{1}{\eta} KL(x', x)$$

$$\leq \max_{y \in \mathbb{R}_+^A} \left\{ (x - y)^\top (\ell + \epsilon \ln x) - \frac{1}{\eta} KL(y, x) \right\}$$

$$= \frac{1}{\eta} \sum_a x_a \left( \eta(\ell_a + \epsilon_a \ln x_a) - 1 + e^{-\eta(\ell_a + \epsilon_a \ln x_a)} \right)$$

$$\leq \frac{1}{\eta} \sum_a x_a \left( \eta \epsilon_a \ln x_a + \eta^2 \ell_a^2 - e^{-\eta \ell_a} + e^{-\eta \ell_a} x_a^{-\eta \epsilon_a} \right) \qquad (z - 1 \leq z^2 - e^{-z} \text{ for } z \geq 0)$$

$$= \eta \sum_a x_a \ell_a^2 + \frac{1}{\eta} \sum_a \left( \eta \epsilon_a x_a \ln x_a + e^{-\eta \ell_a} \left( x_a^{1-\eta \epsilon_a} - x_a \right) \right)$$

$$\leq \eta \sum_a x_a \ell_a^2 + \frac{1}{\eta} \sum_a \left( \eta \epsilon_a x_a \ln x_a + \left( x_a^{1-\eta \epsilon_a} - x_a \right) \right) \qquad (\eta \ell_a \geq 0 \text{ and } x_a^{1-\eta \epsilon_a} - x_a \geq 0)$$

$$\leq \eta \sum_a x_a \ell_a^2 + \frac{1}{\eta} \sum_a \left( \eta \epsilon_a x_a \ln x_a - \eta \epsilon_a x^{1-\eta \epsilon_a} \ln x_a \right) \qquad \text{(by Lemma 12)}$$

$$\leq \eta \sum_a x_a \ell_a^2 + \frac{1}{\eta} \sum_a (\eta \epsilon_a \ln x_a)^2 x_a^{1-\eta \epsilon_a} \qquad \text{(by Lemma 12)}$$

$$\leq \eta \sum_a x_a \ell_a^2 + \frac{1}{\eta} \sum_a (\eta \epsilon_a \ln x_a)^2. \qquad (\eta \epsilon_a \leq 1 \text{ and } x_a \in (0, 1))$$

$\square$

**Lemma 12.** *For $x \in (0, 1)$ and $y > 0$, we have $x^{1-y} - x \leq -yx^{1-y} \ln x$.*

*Proof.*

$$x^{1-y} - x = e^{(\ln \frac{1}{x})(y-1)} - e^{(\ln \frac{1}{x})(-1)}$$

$$= y \left( \ln \frac{1}{x} \right) e^{(\ln \frac{1}{x})\tilde{y}} \qquad \text{(for some } \tilde{y} \in [-1, y-1])$$

$$\leq y \left( \ln \frac{1}{x} \right) e^{(\ln \frac{1}{x})(-1+y)}$$

$$= -y(\ln x)x^{1-y}$$

where the second equality is by the mean value theorem. $\qquad \square$

## B   Last-Iterate Convergence Rate of Algorithm 1

**Proof of Theorem 1.**   The proof is divided into three parts. In Part I, we establish a descent inequality for $\text{KL}(z_t^\star, z_t)$. In Part II, we give an upper bound $\text{KL}(z_t^\star, z_t)$ by recursively applying the descent inequality. Finally in Part III, we show last-iterate convergence rate on the duality gap of $z_t = (x_t, y_t)$. In the proof, we assume without loss of generality that $t \geq t_0 = (\frac{24}{1-k_\eta-k_\epsilon} \ln(\frac{12}{1-k_\eta-k_\epsilon}))^{\frac{1}{1-k_\eta-k_\epsilon}} = (96 \ln(48))^4$ since the theorem holds trivially for constant $t$.

**Part I.**

$$f_t(x_t, y_t) - f_t(x_t^\star, y_t)$$

$$= (x_t - x_t^\star)^\top G y_t + \epsilon_t \left( \sum_a x_{t,a} \ln x_{t,a} - \sum_a x_{t,a}^\star \ln x_{t,a}^\star \right)$$

$$= (x_t - x_t^\star)^\top G y_t + \epsilon_t \left( \sum_a (x_{t,a} - x_{t,a}^\star) \ln x_{t,a} \right) - \underbrace{\epsilon_t \sum_a x_{t,a}^\star \left( \ln x_{t,a}^\star - \ln x_{t,a} \right)}_{=\text{KL}(x_t^\star, x_t)}$$

$$= (x_t - x_t^\star)^\top g_t - \epsilon_t \text{KL}(x_t^\star, x_t) + \underbrace{\sum_a x_{t,a} \left( (Gy_t)_a - \frac{\mathbf{1}[a_t = a]\sigma_t}{x_{t,a} + \beta_t} \right)}_{\triangleq \underline{\xi}_t} + \underbrace{\sum_a x_{t,a}^\star \left( \frac{\mathbf{1}[a_t = a]\sigma_t}{x_{t,a} + \beta_t} - (Gy_t)_a \right)}_{\triangleq \underline{\zeta}_t}$$

$$\text{(by the definition of } g_t)$$

$$\leq \frac{\text{KL}(x_t^\star, x_t) - \text{KL}(x_t^\star, x_{t+1})}{\eta_t} + \eta_t \sum_a x_{t,a} \left( \frac{\mathbf{1}[a_t = a]}{x_{t,a} + \beta_t} \right)^2 + \eta_t \sum_a \epsilon_t^2 \ln^2(x_{t,a}) - \epsilon_t \text{KL}(x_t^\star, x_t) + \underline{\xi}_t + \underline{\zeta}_t$$

$$\text{(by Lemma 11)}$$

$$\leq \frac{(1 - \eta_t \epsilon_t)\text{KL}(x_t^\star, x_t) - \text{KL}(x_t^\star, x_{t+1})}{\eta_t} + 2\eta_t \sum_a \left( \frac{\mathbf{1}[a_t = a]}{x_{t,a} + \beta_t} + \epsilon_t^2 \ln^2(x_{t,a}) \right) + \underline{\xi}_t + \underline{\zeta}_t$$

$$\leq \frac{(1 - \eta_t \epsilon_t)\text{KL}(x_t^\star, x_t) - \text{KL}(x_t^\star, x_{t+1})}{\eta_t}$$

$$\quad + 2\eta_t A \times \underbrace{\frac{1}{A} \sum_a \left( \frac{\mathbf{1}[a_t = a]}{x_{t,a} + \beta_t} - 1 \right)}_{\triangleq \underline{\lambda}_t} + 2\eta_t A + 2\eta_t A \epsilon_t^2 \ln^2 \left( At^2 \right) + \underline{\xi}_t + \underline{\zeta}_t$$

$$\leq \frac{(1 - \eta_t \epsilon_t)\text{KL}(x_t^\star, x_t) - \text{KL}(x_t^\star, x_{t+1})}{\eta_t} + 10\eta_t A \ln^2 \left( At \right) + 2\eta_t A \underline{\lambda}_t + \underline{\xi}_t + \underline{\zeta}_t. \qquad (5)$$

Rearranging the above inequality, we get

$$\text{KL}(x_{t+1}^\star, x_{t+1})$$

$$\leq (1 - \eta_t \epsilon_t)\text{KL}(x_t^\star, x_t) + \eta_t(f_t(x_t^\star, y_t) - f_t(x_t, y_t)) + 10\eta_t^2 A \ln^2 \left( At \right) + 2\eta_t^2 A \underline{\lambda}_t + \eta_t \underline{\xi}_t + \eta_t \underline{\zeta}_t + \underline{v}_t,$$

where $\underline{v}_t \triangleq \mathrm{KL}(x_{t+1}^\star, x_{t+1}) - \mathrm{KL}(x_t^\star, x_{t+1})$. Similarly, since the algorithm for the $y$-player is symmetric, we have the following:

$\mathrm{KL}(y_{t+1}^\star, y_{t+1})$
$\leq (1 - \eta_t \epsilon_t)\mathrm{KL}(y_t^\star, y_t) + \eta_t(f_t(x_t, y_t) - f_t(x_t, y_t^\star)) + 10\eta_t^2 A \ln^2(At) + 2\eta_t^2 A \overline{\lambda}_t + \eta_t \overline{\xi}_t + \eta_t \overline{\zeta}_t + \overline{v}_t$

where

$$\overline{\lambda}_t \triangleq \frac{1}{A} \sum_b \left( \frac{\mathbf{1}[b_t = b]}{y_{t,b} + \beta_t} - 1 \right)$$

$$\overline{\xi}_t \triangleq \sum_b y_{t,b} \left( \left( -(G^\top x_t)_b + 1 \right) - \frac{\mathbf{1}[b_t = b](-\sigma_t + 1)}{y_{t,b} + \beta_t} \right)$$

$$\overline{\zeta}_t \triangleq \sum_b y_{t,b}^\star \left( \frac{\mathbf{1}[b_t = b](-\sigma_t + 1)}{y_{t,b} + \beta_t} - \left( -(G^\top x_t)_b + 1 \right) \right)$$

$$\overline{v}_t \triangleq \mathrm{KL}(y_{t+1}^\star, y_{t+1}) - \mathrm{KL}(y_t^\star, y_{t+1}).$$

Adding the two inequalities above up and using the fact that $f_t(x_t^\star, y_t) - f_t(x_t, y_t^\star) \leq 0$, we get

$$\mathrm{KL}(z_{t+1}^\star, z_{t+1}) \leq (1 - \eta_t \epsilon_t)\mathrm{KL}(z_t^\star, z_t) + 20\eta_t^2 A \ln^2(At) + 2\eta_t^2 A \lambda_t + \eta_t \xi_t + \eta_t \zeta_t + v_t, \quad (6)$$

where $\Box \triangleq \underline{\Box} + \overline{\Box}$ for $\Box = \lambda_t, \xi_t, \zeta_t, v_t$.

**Part II.** Expanding the recursion in Eq. (6), and using the fact that $1 - \eta_1 \epsilon_1 = 0$, we get

$$\mathrm{KL}(z_{t+1}^\star, z_{t+1}) \leq \underbrace{20A \ln^2(At) \sum_{i=1}^t w_t^i \eta_i^2}_{\textbf{term}_1} + \underbrace{2A \sum_{i=1}^t w_t^i \eta_i^2 \lambda_i}_{\textbf{term}_2} + \underbrace{\sum_{i=1}^t w_t^i \eta_i \xi_i}_{\textbf{term}_3} + \underbrace{\sum_{i=1}^t w_t^i \eta_i \zeta_i}_{\textbf{term}_4} + \underbrace{\sum_{i=1}^t w_t^i v_i}_{\textbf{term}_5}$$

where $w_t^i \triangleq \prod_{j=i+1}^t (1 - \eta_j \epsilon_j)$. We can bound each term as follows.

By Lemma 1 and the fact that that $t \geq t_0$, we have

$$\textbf{term}_1 \leq \mathcal{O}\left( A \ln^2(At) \ln(t) t^{-2k_\eta + (k_\eta + k_\epsilon)} \right) = \mathcal{O}\left( A \ln^3(At) t^{-k_\eta + k_\epsilon} \right) = \mathcal{O}\left( A \ln^3(At) t^{-\frac{1}{2}} \right).$$

Using Lemma 7 with $x^\star = \frac{1}{A}\mathbf{1}$, $\ell_i = \mathbf{1}$ for all $i$, and $c_i = w_t^i \eta_i^2$, we have with probability $1 - \frac{\delta}{t^2}$,

$$\textbf{term}_2 = \mathcal{O}\left( \frac{A \ln(At/\delta) \max_{i \leq t} c_i}{\beta_t} \right) \overset{(a)}{=} \mathcal{O}\left( A \ln(At/\delta) t^{k_\beta} \times t^{-2k_\eta} \right) = \mathcal{O}\left( A \ln(At/\delta) t^{-\frac{1}{2}} \right)$$

where in $(a)$ we use Lemma 2 with the fact that $t \geq t_0$.
Using Lemma 6 with $c_i = w_t^i \eta_i$, we have with probability at least $1 - \frac{\delta}{t^2}$,

$$\textbf{term}_3 \leq \mathcal{O}\left( A \sum_{i=1}^t \beta_i c_i + \sqrt{\ln(At/\delta) \sum_{i=1}^t c_i^2} \right)$$

$$= \mathcal{O}\left( A \sum_{i=1}^t \left[ i^{-k_\beta - k_\eta} \prod_{j=i+1}^t \left( 1 - j^{-k_\eta - k_\epsilon} \right) \right] + \sqrt{\ln(At/\delta) \sum_{i=1}^t \left[ i^{-2k_\eta} \prod_{j=i+1}^t \left( 1 - j^{-k_\eta - k_\epsilon} \right) \right]} \right)$$

$$= \mathcal{O}\left( A \ln(t) t^{-k_\beta + k_\epsilon} + t^{-\frac{1}{2}k_\eta + \frac{1}{2}k_\epsilon} \log(At/\delta) \right) \qquad \text{(by Lemma 1 and } t \geq t_0\text{)}$$

$$= \mathcal{O}\left( A \log(At/\delta) t^{-\frac{1}{4}} \right).$$

Using Lemma 7 with $c_i = w_t^i \eta_i$, we get with probability at least $1 - \frac{\delta}{t^2}$,

$$\textbf{term}_4 = \mathcal{O}\left( \frac{\ln(At/\delta) \max_{i \leq t} c_i}{\beta_t} \right) \overset{(a)}{\leq} \mathcal{O}\left( \ln(At/\delta) t^{-k_\eta + k_\beta} \right) = \mathcal{O}\left( \ln(At/\delta) t^{-\frac{1}{4}} \right)$$

where $(a)$ is by Lemma 2 and $t \geq t_0$.

By Lemma 13 and Lemma 1,

$$\mathbf{term}_5 = \mathcal{O}\left(\ln^2(At)\sum_{i=1}^{t} w_t^i t^{-1}\right) = \mathcal{O}\left(\ln^3(At)t^{-1+k_\eta+k_\epsilon}\right) = \mathcal{O}\left(\ln^3(At)t^{-\frac{1}{4}}\right).$$

Combining all terms above, we get that with probability at least $1 - \frac{3\delta}{t^2}$,

$$\mathrm{KL}(z_{t+1}^\star, z_{t+1}) = \mathcal{O}\left(A\ln^3(At/\delta)t^{-\frac{1}{4}}\right). \tag{7}$$

Using an union bound over $t$, we see that Eq. (7) holds for all $t \geq t_0$ with probability at least $1 - \mathcal{O}(\delta)$.

**Part III.**  Using Lemma 9 with $f_t(x, y)$ and $x^\top Gy$ with domains $\Omega_t \times \Omega_t$ and $\Delta_A \times \Delta_A$, we get that for any $(x, y) \in \Delta_A \times \Delta_A$,

$$x_t^{\star\top} Gy - x^\top Gy_t^\star \leq \mathcal{O}\left(\epsilon_t \ln(A) + \frac{1}{t}\right) = \mathcal{O}\left(\ln(A)t^{-k_\epsilon}\right) = \mathcal{O}\left(\ln(A)t^{-\frac{1}{8}}\right).$$

Further using Eq. (7), we get that with probability at least $1 - 3\delta$, for any $t$ and any $(x, y) \in \Delta_A \times \Delta_A$,

$$x_t^\top Gy - x^\top Gy_t \leq \mathcal{O}\left(\ln(A)t^{-\frac{1}{8}} + \|z_t - z_t^\star\|_1\right) \overset{(a)}{=} \mathcal{O}\left(\ln(A)t^{-\frac{1}{8}} + \sqrt{\mathrm{KL}(z_t^\star, z_t)}\right)$$

$$= \mathcal{O}\left(\sqrt{A}\ln^{3/2}(At/\delta)t^{-\frac{1}{8}}\right)$$

where $(a)$ is by Pinsker's inequality. This completes the proof of Theorem 1. ∎

**Lemma 13.** $|v_t| = \mathcal{O}\left(\ln^2(At)t^{-1}\right)$.

*Proof.*

$$|v_t| = \left|\mathrm{KL}(z_{t+1}^\star, z_{t+1}) - \mathrm{KL}(z_t^\star, z_{t+1})\right|$$

$$\leq \mathcal{O}\left(\ln(At)\|z_{t+1}^\star - z_t^\star\|_1\right) \qquad \text{(by Lemma 14)}$$

$$= \mathcal{O}\left(\ln^2(At)t^{-1}\right). \qquad \text{(by Lemma 15)}$$

□

**Lemma 14.** *Let* $x, x_1, x_2 \in \Omega_t$. *Then*

$$|KL(x_1, x) - KL(x_2, x)| \leq \mathcal{O}\left(\ln(At)\|x_1 - x_2\|_1\right).$$

*Proof.*

$$\mathrm{KL}(x_1, x) - \mathrm{KL}(x_2, x)$$

$$= \sum_a \left(x_{1,a} \ln \frac{x_{1,a}}{x_a} - x_{2,a} \ln \frac{x_{2,a}}{x_a}\right)$$

$$= \sum_a (x_{1,a} - x_{2,a}) \ln \frac{x_{1,a}}{x_a} + \sum_a x_{2,a}\left(\ln \frac{x_{1,a}}{x_a} - \ln \frac{x_{2,a}}{x_a}\right)$$

$$\leq \mathcal{O}\left(\ln(At)\|x_1 - x_2\|_1\right) - \mathrm{KL}(x_2, x_1)$$

$$\leq \mathcal{O}\left(\ln(At)\|x_1 - x_2\|_1\right).$$

Similarly, $\mathrm{KL}(x_2, x) - \mathrm{KL}(x_1, x) \leq \mathcal{O}\left(\ln(At)\|x_1 - x_2\|_1\right).$ □

**Lemma 15.** $\|z_t^\star - z_{t+1}^\star\|_1 = \mathcal{O}\left(\frac{\ln(At)}{t}\right).$

*Proof.* Notice that the feasible sets for the two time steps are different. Let $(x'_{t+1}, y'_{t+1})$ be such that $x'_{t+1} = \frac{p_{t+1}}{A}\mathbf{1} + (1 - p_{t+1})x^\star_{t+1}$ and $y'_{t+1} = \frac{p_{t+1}}{A}\mathbf{1} + (1 - p_{t+1})y^\star_{t+1}$ where $p_{t+1} = \min\{1, 2t^{-3}\}$. Since $(x^\star_{t+1}, y^\star_{t+1}) \in \Omega_{t+1} \times \Omega_{t+1}$, we have that for any $a$, $x'_{t+1,a} \geq \frac{p_{t+1}}{A} + (1 - p_{t+1})\frac{1}{A(t+1)^2} \geq \frac{1}{At^2}$. Hence, $(x'_{t+1}, y'_{t+1}) \in \Omega_t \times \Omega_t$.

Because $(x^\star_{t+1}, y^\star_{t+1})$ is the equilibrium of $f_{t+1}$ in $\Omega_{t+1} \times \Omega_{t+1}$, we have that for any $(x, y) \in \Omega_{t+1} \times \Omega_{t+1}$,

$$
\begin{aligned}
& f_{t+1}(x, y^\star_{t+1}) - f_{t+1}(x^\star_{t+1}, y) \\
&= f_{t+1}(x, y^\star_{t+1}) - f_{t+1}(x^\star_{t+1}, y^\star_{t+1}) + f_{t+1}(x^\star_{t+1}, y^\star_{t+1}) - f_{t+1}(x^\star_{t+1}, y) \\
&\geq \epsilon_{t+1}\mathrm{KL}(x, x^\star_{t+1}) + \epsilon_{t+1}\mathrm{KL}(y, y^\star_{t+1}) \\
&\geq \frac{1}{2}\epsilon_{t+1}\left(\|x - x^\star_{t+1}\|_1^2 + \|y - y^\star_{t+1}\|_1^2\right) && \text{(Pinsker's inequality)} \\
&\geq \frac{1}{4}\epsilon_{t+1}\|z - z^\star_{t+1}\|_1^2.
\end{aligned}
$$

where the first inequality is due to the following calculation:

$$
\begin{aligned}
\epsilon_{t+1}\mathrm{KL}(x, x^\star_{t+1}) &= f_{t+1}(x, y^\star_{t+1}) - f_{t+1}(x^\star_{t+1}, y^\star_{t+1}) - \nabla_x f_{t+1}(x^\star_{t+1}, y^\star_{t+1})^\top(x - x^\star_{t+1}) \\
&\leq f_{t+1}(x, y^\star_{t+1}) - f_{t+1}(x^\star_{t+1}, y^\star_{t+1})
\end{aligned}
$$

where we use $\nabla_x f_{t+1}(x^\star_{t+1}, y^\star_{t+1})^\top(x - x^\star_{t+1}) \geq 0$ since $x^\star_{t+1}$ is the minimizer of $f_{t+1}(\cdot, y^\star_{t+1})$ in $\Omega_{t+1}$. Specially, we have

$$
f_{t+1}(x^\star_t, y^\star_{t+1}) - f_{t+1}(x^\star_{t+1}, y^\star_t) \geq \frac{1}{4}\epsilon_{t+1}\|z^\star_t - z^\star_{t+1}\|_1^2. \tag{8}
$$

Similarly, because $(x^\star_t, y^\star_t)$ is the equilibrium of $f_t$ in $\Omega_t \times \Omega_t$, we have

$$
f_t(x'_{t+1}, y^\star_t) - f_t(x^\star_t, y'_{t+1}) \geq \frac{1}{4}\epsilon_t\|z'_{t+1} - z^\star_t\|_1^2,
$$

which implies

$$
\begin{aligned}
& f_t(x^\star_{t+1}, y^\star_t) - f_t(x^\star_t, y^\star_{t+1}) \\
&= f_t(x'_{t+1}, y^\star_t) - f_t(x^\star_t, y'_{t+1}) + f_t(x^\star_{t+1}, y^\star_t) - f_t(x'_{t+1}, y^\star_t) + f_t(x^\star_t, y'_{t+1}) - f_t(x^\star_t, y^\star_{t+1}) \\
&\geq \frac{1}{4}\epsilon_t\|z'_{t+1} - z^\star_t\|_1^2 - \sup_{x \in \Omega_{t+1}}\|\nabla_x f_t(x, y^\star_t)\|_\infty\|x'_{t+1} - x^\star_{t+1}\|_1 - \sup_{y \in \Omega_{t+1}}\|\nabla_y f_t(x^\star_t, y)\|_\infty\|y'_{t+1} - y^\star_{t+1}\|_1 \\
&\geq \frac{1}{8}\epsilon_t\|z^\star_{t+1} - z^\star_t\|_1^2 - \frac{1}{4}\epsilon_t\|z'_{t+1} - z^\star_{t+1}\|_1^2 - \mathcal{O}\left(\ln(At) \times \frac{1}{t^3}\right) \\
&\geq \frac{1}{8}\epsilon_t\|z^\star_{t+1} - z^\star_t\|_1^2 - \mathcal{O}\left(\frac{\ln(At)}{t^3}\right). \tag{9}
\end{aligned}
$$

In the first inequality, we use the fact that $f_t(x, y)$ is convex in $x$ and concave in $y$ and Hölder's inequality. In the second inequality, we use the triangle inequality, $\|\nabla_x f_t(x, y)\|_\infty \leq \max_a\{(Gy)_a + \ln(x_a)\} \leq \mathcal{O}(\ln(At))$, and $\|\nabla_y f_t(x, y)\|_\infty \leq \max_b\{(G^\top x)_b + \ln(y_b)\} \leq \mathcal{O}(\ln(At))$. In the second and third inequality, we use $\|z'_{t+1} - z^\star_{t+1}\|_1 = \mathcal{O}(\frac{1}{t^3})$ by the definition of $z'_{t+1}$.

Combining Eq. (8) and Eq. (9), we get

$$
\begin{aligned}
& \frac{3}{8}\epsilon_{t+1}\|z^\star_t - z^\star_{t+1}\|_1^2 \\
&\leq f_{t+1}(x^\star_t, y^\star_{t+1}) - f_t(x^\star_t, y^\star_{t+1}) - f_{t+1}(x^\star_{t+1}, y^\star_t) + f_t(x^\star_{t+1}, y^\star_t) + \mathcal{O}\left(\frac{\ln(At)}{t^3}\right) \\
&= (f_{t+1} - f_t)(x^\star_t, y^\star_{t+1}) - (f_{t+1} - f_t)(x^\star_{t+1}, y^\star_t) + \mathcal{O}\left(\frac{\ln(At)}{t^3}\right) \\
&\leq \sup_{x,y \in \Omega_{t+1} \times \Omega_{t+1}}\|\nabla f_{t+1}(x, y) - \nabla f_t(x, y)\|_\infty\|(x^\star_t, y^\star_{t+1}) - (x^\star_{t+1}, y^\star_t)\|_1 + \mathcal{O}\left(\frac{\ln(At)}{t^3}\right)
\end{aligned}
$$

$$= \sup_{x,y\in\Omega_{t+1}\times\Omega_{t+1}} \|\nabla f_{t+1}(x,y) - \nabla f_t(x,y)\|_\infty \|z_t^\star - z_{t+1}^\star\|_1 + \mathcal{O}\left(\frac{\ln(At)}{t^3}\right)$$

Solving the inequality, we get

$$\|z_t^\star - z_{t+1}^\star\|_1 \le \mathcal{O}\left(\frac{1}{\epsilon_{t+1}} \sup_{x,y\in\Omega_{t+1}\times\Omega_{t+1}} \|\nabla f_{t+1}(x,y) - \nabla f_t(x,y)\|_\infty + \frac{\ln^{1/2}(At)}{\sqrt{\epsilon_{t+1}}t^{3/2}}\right) \quad (10)$$

$$\le \mathcal{O}\left(\frac{(\epsilon_t - \epsilon_{t+1})\ln(At)}{\epsilon_{t+1}} + \frac{\ln^{1/2}(At)}{\sqrt{\epsilon_{t+1}}t^{3/2}}\right)$$

$$= \mathcal{O}\left(\frac{t^{-k_\epsilon-1}\ln(At)}{t^{-k_\epsilon}} + \frac{\ln^{1/2}(At)}{\sqrt{\epsilon_{t+1}}t^{3/2}}\right)$$

$$= \mathcal{O}\left(\frac{\ln(At)}{t}\right).$$

$\square$

## C  Improved Last-Iterate Convergence under Expectation

In this section, we analyze Algorithm 4, which is almost identical to Algorithm 1 but does not involve the parameter $\beta_t$. The choices of stepsize $\eta_t$ and amount of regularization $\epsilon_t$ are also tuned differently to obtain the best convergence rate.

---

**Algorithm 4** Matrix Game with Bandit Feedback

---
1: **Define:** $\eta_t = t^{-k_\eta}$, $\epsilon_t = t^{-k_\epsilon}$ where $k_\eta = \frac{1}{2}$, $k_\epsilon = \frac{1}{6}$.
  $\Omega_t = \left\{x \in \Delta_\mathcal{A} : x_a \ge \frac{1}{At^2}, \forall a \in \mathcal{A}\right\}$.
2: **Initialization::** $x_1 = \frac{1}{A}\mathbf{1}$.
3: **for** $t = 1, 2, \dots$ **do**
4:     Sample $a_t \sim x_t$, and receive $\sigma_t \in [0,1]$ with $\mathbb{E}\left[\sigma_t\right] = G_{a_t,b_t}$.
5:     Compute $g_t$ where $g_{t,a} = \frac{\mathbf{1}[a_t=a]\sigma_t}{x_{t,a}} + \epsilon_t \ln x_{t,a}, \forall a \in \mathcal{A}$.
6:     Update $x_{t+1} \leftarrow \operatorname{argmin}_{x\in\Omega_{t+1}} \left\{x^\top g_t + \frac{1}{\eta_t}\mathrm{KL}(x,x_t)\right\}$.
7: **end for**

---

**Theorem 4.** *Algorithm 4 guarantees* $\mathbb{E}\left[\max_{x,y\in\Delta_A}\left(x_t^\top Gy - x^\top Gy_t\right)\right] = \mathcal{O}\left(\sqrt{A}\ln^{3/2}(At)t^{-\frac{1}{6}}\right)$ *for any t.*

*Proof.* With the same analysis as in Part I of the proof of Theorem 1, we have

$$f_t(x_t, y_t) - f_t(x_t^\star, y_t)$$
$$\le \frac{(1-\eta_t\epsilon_t)\mathrm{KL}(x_t^\star, x_t) - \mathrm{KL}(x_t^\star, x_{t+1})}{\eta_t} + 10\eta_t A\ln^2(At) + 2\eta_t A\underline{\lambda}_t + \underline{\xi}_t + \underline{\zeta}_t.$$

where

$$\underline{\xi}_t \triangleq \sum_a x_{t,a}\left((Gy_t)_a - \frac{\mathbf{1}[a_t=a]\sigma_t}{x_{t,a}}\right), \qquad \zeta_t \triangleq \sum_a x_{t,a}^\star\left(\frac{\mathbf{1}[a_t=a]\sigma_t}{x_{t,a}} - (Gy_t)_a\right),$$

$$\underline{\lambda}_t \triangleq \frac{1}{A}\sum_a\left(\frac{\mathbf{1}[a_t=a]}{x_{t,a}} - 1\right).$$

Unlike in Theorem 1, here these three terms all have zero mean. Thus, following the same arguments that obtain Eq. (6) and taking expectations, we get

$$\mathbb{E}_t[\mathrm{KL}(z_{t+1}^\star, z_{t+1})] \le (1-\eta_t\epsilon_t)\mathrm{KL}(z_t^\star, z_t) + 20\eta_t^2 A\ln^2(At) + \mathbb{E}_t[v_t]$$

$$\leq (1 - \eta_t \epsilon_t)\mathrm{KL}(z_t^\star, z_t) + \mathcal{O}\left(\eta_t^2 A \ln^2(At) + \frac{\ln^2(At)}{t}\right) \quad \text{(by Lemma 13)}$$

where $v_t = \mathrm{KL}(z_{t+1}^\star, z_{t+1}) - \mathrm{KL}(z_t^\star, z_{t+1})$ and $\mathbb{E}_t[\cdot]$ is the expectation conditioned on history up to round $t$. Then following the same arguments as in Part II of the proof of Theorem 1, we get

$$\mathbb{E}[\mathrm{KL}(z_{t+1}^\star, z_{t+1})] \leq \mathcal{O}\left(A \ln^2(At) \sum_{i=1}^{t} w_t^i \eta_i^2 + \ln^2(At) \sum_{i=1}^{t} w_t^i t^{-1}\right)$$

$$\text{(define } w_t^i \triangleq \prod_{j=i+1}^{t}(1 - \eta_j \epsilon_j))$$

$$\leq \mathcal{O}\left(A \ln^3(At) t^{-k_\eta + k_\epsilon} + \ln^3(At) t^{-1 + k_\eta + k_\epsilon}\right) = \mathcal{O}\left(A \ln^3(At) t^{-\frac{1}{3}}\right).$$

Finally, following the arguments in Part III, we get

$$\mathbb{E}\left[\max_{x,y}\left(x_t^\top G y - x^\top G y_t\right)\right] \leq \mathcal{O}\left(\ln(A)t^{-k_\epsilon} + \sqrt{\mathbb{E}\left[\mathrm{KL}(z_t^\star, z_t)\right]}\right) = \mathcal{O}\left(\sqrt{A}\ln^{3/2}(At)t^{-\frac{1}{6}}\right).$$

$\square$

# D   Last-Iterate Convergence Rate of Algorithm 2

## D.1   On the Assumption of Irreducible Markov Game

**Proposition 1.** *If Assumption 1 holds, then for any $L' = 2L\log_2(S/\delta)$ consecutive steps, under any (non-stationary) policies of the two players, with probability at least $1 - \delta$, every state is visited at least once.*

*Proof.* We first show that for any pair of states $s', s''$, under any non-stationary policy pair, the expected time to reach $s''$ from $s'$ is upper bounded by $L$. For a particular pair of states $(s', s'')$, consider the following modified MDP: let the reward be $r(s, a) = \mathbf{1}[s \neq s'']$, and the transition be the same as the original MDP on all $s \neq s''$, while $P(s''|s'', a) = 1$ (i.e., making $s''$ an absorbing state). Also, let $s'$ be the initial state. By construction, the expected total reward of this MDP is the travelling time from $s'$ to $s''$. By Theorem 7.1.9 of [Put14], there exists a stationary optimal policy in this MDP. The optimal expected total value is then upper bounded by $L$ by Assumption 1. Therefore, for any (possibly sub-optimal) non-stationary policies, the travelling time from $s'$ to $s''$ must also be upper bounded by $L$.

Divide $L'$ steps into $\log_2(S/\delta)$ intervals each of length $2L$, and consider a particualr $s$. Conditioned on $s$ not visited in all intervals $1, 2, \ldots, i - 1$, the probability of still not visiting $s$ in interval $i$ is smaller than $\frac{1}{2}$ (because for any $s'$, $\Pr[T_{s' \to s} > 2L] \leq \frac{\mathbb{E}[T_{s' \to s}]}{2L} \leq \frac{L}{2L} = \frac{1}{2}$, where $T_{s' \to s}$ denotes the travelling time from $s'$ to $s$). Therefore, the probability of not visiting $s$ in all $\log_2(S/\delta)$ intervals is upper bounded by $2^{-\log_2(S/\delta)} = \frac{\delta}{S}$. Using a union bound, we conclude that with probability at least $1 - \delta$, every state is visited at least once within $L'$ steps. $\square$

**Corollary 1.** *If Assumption 1 holds, then with probability $1 - \delta$, for any $t \geq 1$, players visit every state at least once in every $6L\ln(St/\delta)$ consecutive iterations before time $t$.*

*Proof.* First, we fix time $t \geq 1$ and define $t' = 3L\ln(St^3/\delta)$. Let us consider the following time intervals: $[1, t'], [t', 2t'], \ldots, [t - t', t]$. Using Proposition 1, we known for each interval, with probability at least $1 - \frac{\delta}{t^3}$, players visit every state $s$. Using a union bound over all intervals, we have with probability at least $1 - \frac{\delta}{t^2}$, in every interval, players visit every state $s$. Since every $2t'$ consecutive iterations must contain an interval of length $L'$, we have with probability at least $1 - \frac{\delta}{t^2}$, players visit every state $s$ in every $2t'$ consecutive iterations until time $t$. Applying union bound over all $t \geq 1$ completes the proof. $\square$

According to Corollary 1, in the remaining of this section , we assume that for any $t \geq 1$, players visit every state at least once in every $6L\ln(St/\delta)$ iterations until time $t$.

## D.2 Part I. Basic Iteration Properties

**Lemma 16.** *For any $x^s \in \Omega_{\tau+1}$,*

$$f_\tau^s(\hat{x}_\tau^s, \hat{y}_\tau^s) - f_\tau^s(x^s, \hat{y}_\tau^s)$$
$$\leq \frac{(1 - \eta_\tau \epsilon_\tau) KL(x^s, \hat{x}_\tau^s) - KL(x^s, \hat{x}_{\tau+1}^s)}{\eta_\tau} + \frac{10\eta_\tau A \ln^2(A\tau)}{(1-\gamma)^2} + \frac{2\eta_\tau A}{(1-\gamma)^2}\underline{\lambda}_\tau^s + \underline{\xi}_\tau^s + \underline{\zeta}_\tau^s(x^s).$$

*(see the proof for the definitions of $\underline{\lambda}_\tau^s, \underline{\xi}_\tau^s, \underline{\zeta}_\tau^s(\cdot)$)*

*Proof.* Consider a fixed $s$ and a fixed $\tau$, and let $t = t_\tau(s)$ be the time when the players visit $s$ at the $\tau$-th time.

$$f_\tau^s(\hat{x}_\tau^s, \hat{y}_\tau^s) - f_\tau^s(x^s, \hat{y}_\tau^s)$$
$$= (\hat{x}_\tau^s - x^s)^\top \left(G^s + \gamma \mathbb{E}_{s' \sim P^s}\left[V_t^{s'}\right]\right)\hat{y}_\tau^s - \epsilon_\tau \phi(\hat{x}_\tau^s) + \epsilon_\tau \phi(x^s)$$
$$= (\hat{x}_\tau^s - x^s)^\top \left[\left(G^s + \gamma \mathbb{E}_{s' \sim P^s}\left[V_t^{s'}\right]\right)\hat{y}_\tau^s + \epsilon_\tau \ln \hat{x}_\tau^s\right] - \epsilon_\tau KL(x^s, \hat{x}_\tau^s)$$
$$= (\hat{x}_\tau^s - x^s)^\top g_t - \epsilon_\tau KL(x^s, \hat{x}_\tau^s) + \underbrace{(\hat{x}_\tau^s)^\top \left(\left(G^s + \gamma \mathbb{E}_{s' \sim P^s}\left[V_t^{s'}\right]\right)\hat{y}_\tau^s - \frac{\mathbf{1}[\hat{a}_\tau^s = a]\left(\sigma_t + \gamma V_t^{s_{t+1}}\right)}{\hat{x}_{\tau,a}^s + \beta_\tau}\right)}_{\xi_\tau^s}$$
$$+ \underbrace{(x^s)^\top \left(\frac{\mathbf{1}[\hat{a}_\tau^s = a]\left(\sigma_t + \gamma V_t^{s_{t+1}}\right)}{\hat{x}_{\tau,a}^s + \beta_\tau} - \left(G^s + \gamma \mathbb{E}_{s' \sim P^s}\left[V_t^{s'}\right]\right)\hat{y}_\tau^s\right)}_{\underline{\zeta}_\tau^s(x^s)}$$
$$\leq \frac{(1 - \eta_\tau \epsilon_\tau) KL(x^s, \hat{x}_\tau^s) - KL(x^s, \hat{x}_{\tau+1}^s)}{\eta_\tau}$$
$$+ \frac{10\eta_\tau A \ln^2(A\tau)}{(1-\gamma)^2} + \frac{2\eta_\tau A}{(1-\gamma)^2} \times \underbrace{\frac{1}{|\mathcal{A}|}\sum_a \left(\frac{\mathbf{1}[\hat{a}_\tau^s = a]}{\hat{x}_{\tau,a}^s + \beta_\tau} - 1\right)}_{\underline{\lambda}_\tau^s} + \underline{\xi}_\tau^s + \underline{\zeta}_\tau^s(x^s),$$

where we omit some calculation steps due to the similarity to Eq. (5).

$\square$

## D.3 Part II. Policy Convergence to the Nash of Regularized Game

**Lemma 17.** *With probability at least $1 - \mathcal{O}(\delta)$, for all $s \in S$, $t \geq 1$ and $\tau \geq 1$ such that $t_\tau(s) \leq t$, we have*

$$KL(\hat{z}_{\tau\star}^s, \hat{z}_\tau^s) \leq \mathcal{O}\left(A\ln^5(SAt/\delta)L^2\tau^{-k_\sharp}\right),$$

*where $k_\sharp = \min\{k_\beta - k_\epsilon, k_\eta - k_\beta, k_\alpha - k_\eta - 2k_\epsilon\}$.*

*Proof.* In this proof, we abbreviate $\underline{\zeta}_i^s(\hat{x}_{i\star}^s)$ as $\underline{\zeta}_i^s$. By Lemma 16, for all $i \leq \tau$ we have

$$KL(\hat{x}_{i\star}^s, \hat{x}_{i+1}^s) \leq (1 - \eta_i \epsilon_i)KL(\hat{x}_{i\star}^s, \hat{x}_i^s) + \eta_i \left(f_i^s(\hat{x}_{i\star}^s, \hat{y}_i^s) - f_i^s(\hat{x}_i^s, \hat{y}_i^s)\right)$$
$$+ \frac{10\eta_i^2 A \ln^2(A\tau)}{(1-\gamma)^2} + \frac{2\eta_i^2 A}{(1-\gamma)^2}\underline{\lambda}_i^s + \eta_i \underline{\xi}_i^s + \eta_i \underline{\zeta}_i^s.$$

Similarly, for all $i \leq \tau$, we have

$$KL(\hat{y}_{i\star}^s, \hat{y}_{i+1}^s) \leq (1 - \eta_i \epsilon_i)KL(\hat{y}_{i\star}^s, \hat{y}_i^s) + \eta_i \left(f_i^s(\hat{x}_i^s, \hat{y}_i^s) - f_i^s(\hat{x}_i^s, \hat{y}_{i\star}^s)\right)$$
$$+ \frac{10\eta_i^2 A \ln^2(A\tau)}{(1-\gamma)^2} + \frac{2\eta_i^2 A}{(1-\gamma)^2}\overline{\lambda}_i^s + \eta_i \overline{\xi}_i^s + \eta_i \overline{\zeta}_i^s.$$

Adding the two inequalities up, and using $f_i^s(\hat{x}_{i\star}^s, \hat{y}_i^s) - f_i^s(\hat{x}_i^s, \hat{y}_{i\star}^s) \le 0$ because $(\hat{x}_{i\star}^s, \hat{y}_{i\star}^s)$ is the equilibrium of $f_i^s$, we get for $i \le \tau$

$$\mathrm{KL}(\hat{z}_{i+1\star}^s, \hat{z}_{i+1}^s) \le (1 - \eta_i \epsilon_i) \mathrm{KL}(\hat{z}_{i\star}^s, \hat{z}_i^s) + \frac{20\eta_i^2 A \ln^2(A\tau)}{(1-\gamma)^2} + \frac{2\eta_i^2 A}{(1-\gamma)^2}\lambda_i^s + \eta_i \xi_i^s + \eta_i \zeta_i^s + v_i^s, \tag{11}$$

where $v_i^s = \mathrm{KL}(\hat{z}_{i+1\star}^s, \hat{z}_{i+1}^s) - \mathrm{KL}(\hat{z}_{i\star}^s, \hat{z}_{i+1}^s)$ and $\square^s = \underline{\square}^s + \overline{\square}^s$ for $\square = \xi_i, \zeta_i$.

Expanding Eq. (11), we get

$$\mathrm{KL}(\hat{z}_{\tau+1\star}^s, \hat{z}_{\tau+1}^s) \le \underbrace{\frac{20A\ln^2(A\tau)}{(1-\gamma)^2}\sum_{i=1}^{\tau} w_\tau^i \eta_i^2}_{\textbf{term}_1} + \underbrace{\frac{2A}{(1-\gamma)^2}\sum_{i=1}^{\tau} w_\tau^i \eta_i^2 \lambda_i^s}_{\textbf{term}_2} + \underbrace{\sum_{i=1}^{\tau} w_\tau^i \eta_i \xi_i^s}_{\textbf{term}_3} + \underbrace{\sum_{i=1}^{t} w_\tau^i \eta_i \zeta_i^s}_{\textbf{term}_4} + \underbrace{\sum_{i=1}^{\tau} w_\tau^i v_i^s}_{\textbf{term}_5}.$$

These five terms correspond to those in Eq. (6), and can be handled in the same way. For $\textbf{term}_1$ to $\textbf{term}_4$, we follow exactly the same arguments there, and bound their sum as with probability at least $1 - \mathcal{O}\left(\frac{\delta}{S\tau^2}\right)$,

$$\sum_{j=1}^{4} \textbf{term}_j = \mathcal{O}\left(A\ln^3(SA\tau/\delta)\left(\tau^{-k_\eta + k_\epsilon} + \tau^{-2k_\eta + k_\beta} + \tau^{-k_\beta + k_\epsilon} + \tau^{-\frac{1}{2}k_\eta + \frac{1}{2}k_\epsilon} + \tau^{-k_\eta + k_\beta}\right)\right).$$

To bound $\textbf{term}_5$, by Lemma 14 and Lemma 18, we have

$$|v_\tau^s| = \mathcal{O}\left(\ln(A\tau)\right) \cdot \|\hat{z}_{\tau\star}^s - \hat{z}_{\tau+1\star}^s\|_1 = \mathcal{O}\left(\ln^4(SAt/\delta)L^2 \cdot \tau^{-k_\alpha + k_\epsilon}\right).$$

Therefore, by Lemma 1,

$$\textbf{term}_5 = \sum_{i=1}^{\tau} w_\tau^i v_i^s = \mathcal{O}\left(\ln^5(SAt/\delta)L^2 \cdot \tau^{-k_\alpha + k_\eta + 2k_\epsilon}\right).$$

Combining all the terms with union bound over $s \in S$ and $\tau \ge 1$ finishes the proof. $\qquad\square$

**Lemma 18.** *For any $s$ and $\tau \ge 0$ such that $t_\tau(s) \le t$, $\|\hat{z}_{\tau\star}^s - \hat{z}_{\tau+1\star}^s\|_1 = \mathcal{O}\left(\ln^3(SAt/\delta)L^2 \cdot \tau^{-k_\alpha + k_\epsilon}\right).$*

*Proof.* The bound holds trivially when $\tau \le 2L$. Below we focus on the case with $\tau > 2L$. By exactly the same arguments as in the proof of Lemma 15, we have an inequality similar to Eq. (10):

$$\|z_{\tau\star}^s - z_{\tau+1\star}^s\|_1$$
$$= \mathcal{O}\left(\frac{1}{\epsilon_{\tau+1}}\sup_{x^s, y^s}\|\nabla f_\tau^s(x^s, y^s) - \nabla f_{\tau+1}^s(x^s, y^s)\|_\infty + \frac{\ln^{1/2}(A\tau)}{\sqrt{\epsilon_{\tau+1}}\tau^{3/2}}\right)$$
$$\le \mathcal{O}\left(\frac{1}{\epsilon_{\tau+1}}\sup_{s'}\left|V_{t_\tau(s)}^{s'} - V_{t_{\tau+1}(s)}^{s'}\right| + \frac{(\epsilon_\tau - \epsilon_{\tau+1})\ln(A\tau)}{\epsilon_{\tau+1}} + \frac{\ln^{1/2}(A\tau)}{\sqrt{\epsilon_{\tau+1}}\tau^{3/2}}\right)$$
$$\le \mathcal{O}\left(\frac{1}{\epsilon_{\tau+1}}\sup_{s'}\left|V_{t_\tau(s)}^{s'} - V_{t_{\tau+1}(s)}^{s'}\right| + \frac{\ln(A\tau)}{\tau}\right). \tag{12}$$

Since $t_\tau(s) \le t$ and we assume that every state is visited at least once in $6L\log(St/\delta)$ steps (Corollary 1), we have that for any state $s'$, $n_{t_\tau(s)}^{s'} \ge \frac{t_\tau(s)}{6L\log(St/\delta)} - 1$. Thus, whenever $V_t^{s'}$ updates between $t_\tau(s)$ and $t_{\tau+1}(s)$, the change is upper bounded by $\frac{1}{1-\gamma}(\frac{t_\tau(s)}{6L\log(St/\delta)} - 1)^{-k_\alpha}$. Besides, between $t_\tau(s)$ and $t_{\tau+1}(s)$, $V_t^{s'}$ can change at most $6L\log(St/\delta)$ times. Therefore,

$$\left|V_{t_\tau(s)}^{s'} - V_{t_{\tau+1}(s)}^{s'}\right|$$
$$\le \frac{1}{1-\gamma} \times 6L\log(St/\delta) \times \left(\frac{t_\tau(s)}{6L\log(St/\delta)} - 1\right)^{-k_\alpha} \le \frac{1}{1-\gamma} \times 6L\log(St/\delta) \times \left(\frac{\tau}{6L\log(St/\delta)} - 1\right)^{-k_\alpha}$$
$$= \mathcal{O}\left(\frac{L^2\ln^2(St/\delta)\tau^{-k_\alpha}}{1-\gamma}\right), \tag{13}$$

where the last inequality holds since $k_\alpha < 1$. Combining Eq. (12) and Eq. (13) with the fact that $\epsilon_\tau = \frac{1}{1-\gamma}\tau^{-k_\epsilon}$ finishes the proof. $\qquad\square$

### D.4 Part III. Value Convergence

For positive integers $\tau \geq i$, we define $\alpha_\tau^i = \alpha_i \prod_{j=i+1}^\tau (1 - \alpha_j)$.

**Lemma 19** (weighted regret bound). *With probability $1 - \mathcal{O}(\delta)$, for any $s$, any visitation count $\tau \geq \tau_0$, and any $x^s \in \Omega_{\tau+1}$,*

$$\sum_{i=1}^\tau \alpha_\tau^i \left( f_i^s(\hat{x}_i^s, \hat{y}_i^s) - f_i^s(x^s, \hat{y}_i^s) \right) \leq \mathcal{O}\left( \frac{A \ln^3(SA\tau/\delta)\tau^{-k'}}{1 - \gamma} \right).$$

*where $k' = \min\{k_\eta, k_\beta, k_\alpha - k_\beta\}$.*

*Proof.* We will be considering a weighted sum of the instantaneous regret bound established in Lemma 16. However, notice that for $f_i^s$, Lemma 16 only provides a regret bound with comparators in $\Omega_{i+1}$. Therefore, for a fixed $x^s \in \Omega_{\tau+1}$, we define the following auxiliary comparators for all $i = 1, \ldots, \tau$:

$$\widetilde{x}_i^s = \frac{p_i}{A}\mathbf{1} + (1 - p_i)\, x^s$$

where $p_i \triangleq \frac{(\tau+1)^2 - (i+1)^2}{(i+1)^2[(\tau+1)^2 - 1]}$. Since $x^s \in \Omega_{\tau+1}$, we have that for any $a$, $\widetilde{x}_{i,a}^s \geq \frac{p_i}{A} + \frac{1-p_i}{A(\tau+1)^2} = \frac{1}{A(i+1)^2}$, and thus $\widetilde{x}_i^s \in \Omega_{i+1}$.

Applying Lemma 16 and considering the weighted sum of the bounds, we get

$$\sum_{i=1}^\tau \alpha_\tau^i \left( f_i^s(\hat{x}_i^s, \hat{y}_i^s) - f_i^s(\widetilde{x}_i^s, \hat{y}_i^s) \right)$$

$$\leq \sum_{i=1}^\tau \alpha_\tau^i \left( \frac{(1 - \eta_i \epsilon_i)\mathrm{KL}(\widetilde{x}_i^s, \hat{x}_i^s) - \mathrm{KL}(\widetilde{x}_i^s, \hat{x}_{i+1}^s)}{\eta_i} + \frac{10\eta_i A \ln^2(A\tau)}{(1-\gamma)^2} + \frac{2\eta_i A}{(1-\gamma)^2}\lambda_i^s + \xi_i^s + \zeta_i^s(\widetilde{x}_i^s) \right)$$

$$\leq \underbrace{\sum_{i=2}^\tau \left( \frac{\alpha_\tau^i(1 - \eta_i\epsilon_i)}{\eta_i}\mathrm{KL}(\widetilde{x}_i^s, \hat{x}_i^s) - \frac{\alpha_\tau^{i-1}}{\eta_{i-1}}\mathrm{KL}(\widetilde{x}_{i-1}^s, \hat{x}_i^s) \right)}_{\mathbf{term}_0} \qquad \text{(notice that } (1 - \eta_1\epsilon_1) = 0)$$

$$+ \underbrace{\frac{10A\ln^2(A\tau)}{(1-\gamma)^2}\sum_{i=1}^\tau \alpha_\tau^i \eta_i}_{\mathbf{term}_1} + \underbrace{\frac{2A}{(1-\gamma)^2}\sum_{i=1}^\tau \alpha_\tau^i \eta_i \lambda_i^s}_{\mathbf{term}_2} + \underbrace{\sum_{i=1}^\tau \alpha_\tau^i \xi_i^s}_{\mathbf{term}_3} + \underbrace{\sum_{i=1}^\tau \alpha_\tau^i \zeta_i^s(\widetilde{x}_i^s)}_{\mathbf{term}_4}.$$

$$\mathbf{term}_0 = \sum_{i=2}^\tau \mathrm{KL}(\widetilde{x}_i^s, \hat{x}_i^s)\left( \frac{\alpha_\tau^i(1 - \eta_i\epsilon_i)}{\eta_i} - \frac{\alpha_\tau^{i-1}}{\eta_{i-1}} \right) + \sum_{i=2}^\tau \frac{\alpha_\tau^{i-1}}{\eta_{i-1}}\left( \mathrm{KL}(\widetilde{x}_{i-1}^s, \hat{x}_i^s) - \mathrm{KL}(\widetilde{x}_i^s, \hat{x}_i^s) \right)$$

$$\overset{(a)}{\leq} 0 + \mathcal{O}\left( \ln(A\tau)\sum_{i=2}^\tau \frac{\alpha_\tau^{i-1}}{\eta_{i-1}}\|\widetilde{x}_{i-1}^s - \widetilde{x}_i^s\|_1 \right)$$

$$= \mathcal{O}\left( \ln(A\tau)\sum_{i=2}^\tau \frac{\alpha_\tau^{i-1}}{\eta_{i-1}}|p_{i-1} - p_i| \right)$$

$$\leq \mathcal{O}\left( \ln(A\tau)\sum_{i=2}^\tau \frac{\alpha_\tau^{i-1}}{\eta_{i-1}}\frac{1}{(i-1)^2} \right)$$

$$= \mathcal{O}\left( \frac{\ln(A\tau)}{1 - \gamma}\tau^{k_\eta - 2} \right) \qquad \text{(by Lemma 1)}$$

where $(a)$ is by Lemma 14 and the following calculation:

$$\frac{\alpha_\tau^i(1 - \eta_i\epsilon_i)}{\eta_i} \times \frac{\eta_{i-1}}{\alpha_\tau^{i-1}} = \frac{\eta_{i-1}}{\eta_i} \times \frac{\alpha_i}{\alpha_{i-1}} \times \frac{1 - \eta_i\epsilon_i}{1 - \alpha_i} = \left( \frac{i-1}{i} \right)^{-k_\eta + k_\alpha} \times \frac{1 - i^{-k_\eta - k_\epsilon}}{1 - i^{-k_\alpha}} \leq 1 \times 1 = 1.$$

We proceed to bound other terms as follows: with probability at least $1 - \mathcal{O}\left(\frac{\delta}{S\tau^2}\right)$

$$\mathbf{term}_1 = \mathcal{O}\left(\frac{A \ln^3(A\tau)}{1-\gamma} \tau^{-k_\eta}\right), \tag{Lemma 1}$$

$$\mathbf{term}_2 = \mathcal{O}\left(\frac{A \ln(SA\tau/\delta)}{1-\gamma} \times \max_{i \leq \tau} \frac{\alpha_\tau^i \eta_i}{\beta_\tau}\right) \tag{Lemma 7}$$

$$= \mathcal{O}\left(\frac{A \ln(SA\tau/\delta)\tau^{-k_\alpha - k_\eta + k_\beta}}{1-\gamma}\right), \tag{Lemma 2}$$

$$\mathbf{term}_3 = \mathcal{O}\left(\frac{A}{1-\gamma}\sum_{i=1}^{\tau} \beta_i \alpha_\tau^i + \frac{1}{1-\gamma}\sqrt{\ln(SA\tau/\delta)\sum_{i=1}^{\tau}(\alpha_\tau^i)^2}\right) \tag{Lemma 6}$$

$$= \mathcal{O}\left(\frac{A \ln(A\tau)\tau^{-k_\beta}}{1-\gamma} + \frac{1}{1-\gamma}\sqrt{\ln(SA\tau/\delta)\sum_{i=1}^{\tau}\alpha_\tau^i \alpha_i}\right) \tag{Lemma 1}$$

$$= \mathcal{O}\left(\frac{A \ln(SA\tau/\delta)\left(\tau^{-k_\beta} + \tau^{-\frac{k_\alpha}{2}}\right)}{1-\gamma}\right), \tag{Lemma 1}$$

$$\mathbf{term}_4 = \sum_{i=1}^{\tau} \alpha_\tau^i p_i \underline{\zeta}_i^s\left(\frac{1}{A}\mathbf{1}\right) + \sum_{i=1}^{\tau} \alpha_\tau^i (1-p_i)\underline{\zeta}_i^s(x^s) \qquad \text{(by the linearity of } \underline{\zeta}_i^s(\cdot)\text{)}$$

$$= \mathcal{O}\left(\frac{\ln(SA\tau/\delta)}{1-\gamma} \max_{i \leq \tau} \frac{\alpha_\tau^i}{\beta_\tau}\right) \tag{Lemma 7}$$

$$= \mathcal{O}\left(\frac{\ln(SA\tau/\delta)\tau^{-k_\alpha + k_\beta}}{1-\gamma}\right). \tag{Lemma 2}$$

Combining all terms, we get

$$\sum_{i=1}^{\tau} \alpha_\tau^i \left(f_i^s(\hat{x}_i^s, \hat{y}_i^s) - f_i^s(\widetilde{x}_i^s, \hat{y}_i^s)\right) = \mathcal{O}\left(\frac{A \ln^3(SA\tau/\delta)\left(\tau^{-k_\eta} + \tau^{-k_\beta} + \tau^{-k_\alpha + k_\beta}\right)}{1-\gamma}\right). \tag{14}$$

Finally,

$$\sum_{i=1}^{\tau} \alpha_\tau^i \left(f_i^s(\widetilde{x}_i^s, \hat{y}_i^s) - f_i^s(x^s, \hat{y}_i^s)\right) = \mathcal{O}\left(\frac{\ln(A\tau)}{1-\gamma}\sum_{i=1}^{\tau} \alpha_\tau^i \|\widetilde{x}_i^s - x^s\|_1\right)$$

$$= \mathcal{O}\left(\frac{\ln(A\tau)}{1-\gamma}\sum_{i=1}^{\tau} \alpha_\tau^i p_i\right) = \mathcal{O}\left(\frac{\ln(A\tau)}{\tau^2(1-\gamma)}\right). \tag{15}$$

Adding up Eq. (14) and Eq. (15) and applying union bound over all $s \in \mathcal{S}$ and $\tau$ finish the proof. $\qquad\square$

**Lemma 20.** *With probability at least $1 - \mathcal{O}(\delta)$, for any state $s \in \mathcal{S}$ and time $t \geq 1$, we have*

$$|V_t^s - V_\star^s| \leq \mathcal{O}\left(\frac{A \ln(SAt/\delta)}{(1-\gamma)^2}\left(\frac{L \ln(St/\delta)}{1-\gamma} \ln \frac{t}{1-\gamma}\right)^{\frac{k_*}{1-k_\alpha}}\left(\frac{L \ln(St/\delta)}{t}\right)^{k_*}\right),$$

*where $k_* = \min\{k_\eta, k_\beta, k_\alpha - k_\beta, k_\epsilon\}$.*

*Proof.* Fix an $s$ and a visitation count $\tau$. Let $t_i$ be the time index when the players visit $s$ for the $i$-th time. Then with probability at least $1 - \frac{\delta}{S\tau^2}$,

$$V_{t_\tau}^s = \sum_{i=1}^{\tau} \alpha_\tau^i \left(\sigma_{t_i} + \gamma V_{t_i}^{s_{t_i+1}}\right)$$

$$\leq \sum_{i=1}^{\tau} \alpha_\tau^i \hat{x}_i^{s\top}\left(G^s + \gamma \mathbb{E}_{s' \sim P^s}\left[V_{t_i}^{s'}\right]\right)\hat{y}_i^s + \sum_{i=1}^{\tau} \alpha_\tau^i \left[\sigma_{t_i} + \gamma V_{t_i}^{s_{t_i+1}} - \hat{x}_i^{s\top}\left(G^s + \gamma \mathbb{E}_{s' \sim P^s}\left[V_{t_i}^{s'}\right]\right)\hat{y}_i^s\right]$$

$$= \sum_{i=1}^{\tau} \alpha_\tau^i \left( f_i^s(\hat{x}_i^s, \hat{y}_i^s) + \epsilon_i \phi(\hat{x}_i^s) - \epsilon_i \phi(\hat{y}_i^s) \right) + \mathcal{O}\left( \frac{1}{1-\gamma} \sqrt{\ln(S\tau/\delta) \sum_{i=1}^{\tau} (\alpha_\tau^i)^2} \right)$$

(Azuma's inequality)

$$\leq \sum_{i=1}^{\tau} \alpha_\tau^i f_i^s(\hat{x}_i^s, \hat{y}_i^s) + \mathcal{O}\left( \epsilon_\tau \ln(A) + \frac{\ln(SA\tau/\delta)\tau^{-\frac{k_\alpha}{2}}}{1-\gamma} \right)$$

$$\leq \min_x \sum_{i=1}^{\tau} \alpha_\tau^i f_i^s(x^s, \hat{y}_i^s) + \mathcal{O}\left( \epsilon_\tau \ln(A) + \frac{A \ln^3(SA\tau/\delta)\tau^{-k'}}{1-\gamma} \right)$$

($k'$ is defined in Lemma 19 with $k' \leq \frac{1}{2}(k_\beta + k_\alpha - k_\beta) = \frac{k_\alpha}{2}$)

$$\leq \min_{x^s} \sum_{i=1}^{\tau} \alpha_\tau^i (x^s)^\top \left( G^s + \gamma \mathbb{E}_{s' \sim P^s}\left[ V_{t_i}^{s'} \right] \right) \hat{y}_i^s + \mathcal{O}\left( \frac{A \ln^3(SA\tau/\delta)\tau^{-k_*}}{1-\gamma} \right)$$

$$\leq \min_{x^s} \sum_{i=1}^{\tau} \alpha_\tau^i (x^s)^\top \left( G^s + \gamma \mathbb{E}_{s' \sim P^s}\left[ V_\star^{s'} \right] \right) \hat{y}_i^s + \gamma \sum_{i=1}^{\tau} \alpha_\tau^i \max_{s'} \left| V_{t_i}^{s'} - V_\star^{s'} \right| + \mathcal{O}\left( \frac{A \ln^3(SA\tau/\delta)\tau^{-k_*}}{1-\gamma} \right)$$

$$\leq \min_{x^s} \max_{y^s} (x^s)^\top \left( G^s + \gamma \mathbb{E}_{s' \sim P^s}\left[ V_\star^{s'} \right] \right) y^s + \gamma \sum_{i=1}^{\tau} \alpha_\tau^i \max_{s'} \left| V_{t_i}^{s'} - V_\star^{s'} \right| + \mathcal{O}\left( \frac{A \ln^3(SA\tau/\delta)\tau^{-k_*}}{1-\gamma} \right)$$

$$\leq V_\star^s + \gamma \sum_{i=1}^{\tau} \alpha_\tau^i \max_{s'} \left| V_{t_i}^{s'} - V_\star^{s'} \right| + \mathcal{O}\left( \frac{A \ln^3(SA\tau/\delta)\tau^{-k_*}}{1-\gamma} \right).$$

Similar inequality can be also obtained through the perspective of the other player: with probability at least $1 - \frac{\delta}{S\tau^2}$

$$V_{t_\tau}^s \geq V_\star^s - \gamma \sum_{i=1}^{\tau} \alpha_\tau^i \max_{s'} \left| V_{t_i}^{s'} - V_\star^{s'} \right| - \mathcal{O}\left( \frac{A \ln^3(SA\tau/\delta)\tau^{-k_*}}{1-\gamma} \right),$$

which, combined with the previous inequality and union bound over $s \in \mathcal{S}$ and $\tau \geq 1$, gives the following relation: with probability at least $1 - \mathcal{O}(\delta)$, for any $s \in \mathcal{S}$ and $\tau \geq 1$,

$$\left| V_{t_\tau}^s - V_\star^s \right| \leq \gamma \sum_{i=1}^{\tau} \alpha_\tau^i \max_{s'} \left| V_{t_i}^{s'} - V_\star^{s'} \right| + \mathcal{O}\left( \frac{A \ln^3(SA\tau/\delta)\tau^{-k_*}}{1-\gamma} \right). \tag{16}$$

Before continuing, we first some auxiliary quantities. For a fixed $t$, define

$$u(t) = \left\lceil \left( \frac{16 \times 6L \ln(St/\delta)}{1-\gamma} \ln \frac{t}{1-\gamma} \right)^{\frac{1}{1-k_\alpha}} \right\rceil;$$

for fixed $(\tau, t)$ we further define

$$v(\tau, t) = \left\lfloor \tau - 3\tau^{k_\alpha} \ln \frac{t}{1-\gamma} \right\rfloor.$$

Now we continue to prove a bound for $|V_t^s - V_\star^s|$. Suppose that Eq. (16) can be written as

$$\left| V_{t_\tau}^s - V_\star^s \right| \leq \gamma \sum_{i=1}^{\tau} \alpha_\tau^i \max_{s'} \left| V_{t_i}^{s'} - V_\star^{s'} \right| + \frac{C_1 A \ln^3(SA\tau/\delta)\tau^{-k_*}}{1-\gamma} \tag{17}$$

for a universal constant $C_1 \geq 1$. Below we use induction to show that for all $t$,

$$\left| V_t^s - V_\star^s \right| \leq \Phi(t) \triangleq \frac{8 C_1 A \ln^3(SAt/\delta)}{(1-\gamma)^2} \left( \frac{6L \ln(St/\delta)(u(t)+1)}{t} \right)^{k_*}. \tag{18}$$

This is trivial for $t = 1$.

Suppose that Eq. (18) holds for all time $1, \ldots, t-1$ and for all $s$. Now we consider time $t$ and a fixed state $s$. We denote $L' = 6L \ln(St/\delta)$. Let $\tau = n_{t+1}^s$ and let $1 \leq t_1 < t_2 < \cdots < t_\tau \leq t$

be the time indices when the players visit state $s$. If $t \le L'(u(t) + 1)$, then Eq. (18) is trivial. If $t \ge L'(u(t) + 1)$, we have $\tau \ge \frac{t}{L'} - 1 \ge u(t)$. Therefore,

$$|V_t^s - V_\star^s|$$

$$= |V_{t_\tau}^s - V_\star^s| \qquad (t_\tau \text{ is the last time up to time } t \text{ when } V_t^s \text{ is updated})$$

$$\le \gamma \sum_{i=1}^{\tau} \alpha_\tau^i \max_{s'} \left|V_{t_i}^{s'} - V_\star^{s'}\right| + \frac{C_1 A \ln^3(SA\tau/\delta)\tau^{-k_*}}{1 - \gamma} \qquad \text{(by Eq. (17))}$$

$$\le \gamma \sum_{i=1}^{v(\tau,t)} \alpha_\tau^i \max_{s'} \left|V_{t_i}^{s'} - V_\star^{s'}\right| + \gamma \sum_{i=v(\tau,t)+1}^{\tau} \alpha_\tau^i \max_{s'} \left|V_{t_i}^{s'} - V_\star^{s'}\right| + \frac{C_1 A \ln^3(SA\tau/\delta)\tau^{-k_*}}{1 - \gamma}$$

$$\overset{(a)}{\le} \gamma\tau \times \frac{(1-\gamma)^3}{t^3} \times \frac{1}{1-\gamma} + \gamma \sum_{i=v(\tau,t)+1}^{\tau} \alpha_\tau^i \Phi(t_i) + \frac{C_1 A \ln^3(SA\tau/\delta)\tau^{-k_*}}{1 - \gamma}$$

$$\le \gamma \sum_{i=v(\tau,t)+1}^{\tau} \alpha_\tau^i \Phi(t_i) + \frac{2C_1 A \ln^3(SA\tau/\delta)\tau^{-k_*}}{1 - \gamma} \qquad \text{(induction hypothesis)}$$

$$\le \gamma \sum_{i=v(\tau,t)+1}^{\tau} \alpha_\tau^i \Phi(t) \times \left(\frac{t}{t_i}\right)^{k_*} + \frac{2C_1 A \ln^3(SA\tau/\delta)\tau^{-k_*}}{1 - \gamma}$$

$$\qquad \text{(by the definition of } \Phi \text{ and that } u(\cdot) \text{ is an increasing function)}$$

$$\overset{(b)}{\le} \gamma \left(1 + \frac{1-\gamma}{2}\right) \sum_{i=v(\tau,t)+1}^{\tau} \alpha_\tau^i \Phi(t) + \frac{2C_1 A \ln^3(SA\tau/\delta)\tau^{-k_*}}{1 - \gamma}$$

$$\le \gamma \left(1 + \frac{1-\gamma}{2}\right) \Phi(t) + \frac{2C_1 A \ln^3(SAt/\delta)\left(\frac{t}{2L'}\right)^{-k_*}}{1 - \gamma}$$

$$\qquad (t \ge \tau \ge \frac{t}{L'} - 1 \ge \frac{t}{2L'} \text{ since } t \ge L'(u(t) + 1) \ge 2L')$$

$$\le \gamma \left(1 + \frac{1-\gamma}{2}\right) \Phi(t) + \frac{1-\gamma}{2}\Phi(t)$$

$$= \Phi(t).$$

In $(a)$ we use the following property: if $\tau \ge u(t)$ and $i \le v(\tau, t)$, then

$$\alpha_\tau^i = i^{-k_\alpha} \prod_{j=i+1}^{\tau} \left(1 - j^{-k_\alpha}\right) \le \left(1 - \tau^{-k_\alpha}\right)^{\tau - i}$$

$$\le \left(1 - \tau^{-k_\alpha}\right)^{3\tau^{k_\alpha} \ln \frac{t}{1-\gamma}} \le \exp\left(-\tau^{-k_\alpha} \cdot 3\tau^{k_\alpha} \ln \frac{t}{1-\gamma}\right) = \frac{(1-\gamma)^3}{t^3}$$

In $(b)$ we use the following calculation:

$$\frac{t}{t_i} \le \frac{t_{\tau+1}}{t_i} = 1 + \frac{t_{\tau+1} - t_i}{t_i} \le 1 + \frac{L'(\tau + 1 - i)}{i}$$

$$\le 1 + L'\left(\frac{\tau + 1}{v(\tau, t)} - 1\right)$$

$$\le 1 + L'\left(\frac{\tau + 1}{\tau - 4\tau^{k_\alpha} \ln \frac{t}{1-\gamma}} - 1\right)$$

$$= 1 + L'\left(\frac{1 + \frac{1}{\tau}}{1 - 4\tau^{k_\alpha - 1} \ln \frac{t}{1-\gamma}} - 1\right)$$

$$\le 1 + L'\left(\frac{1 + \frac{1-\gamma}{16L'}}{1 - \frac{1-\gamma}{4L'}} - 1\right)$$

$$\le 1 + L'\left(\frac{1-\gamma}{2L'}\right)$$

$$= 1 + \frac{1-\gamma}{2}$$

where the first inequality is due to the fact that at time $t$, state $s$ has only been visited for $\tau$ times; the second inequality is because for any $k > j$, we have $t_j \geq j$ and $t_k - t_j \leq L'(k-j)$; the third inequality is by $i \geq v(\tau, t)$; the fourth inequality is by the definition of $v(\tau, t)$; the fifth inequality is because $4\tau^{k_\alpha - 1} \ln \frac{t}{1-\gamma} \leq \frac{1-\gamma}{4L'}$ since $\tau \geq u(t)$, and $\frac{1}{\tau} < \frac{1}{u(t)} \leq \frac{1-\gamma}{16L'}$ since $u(t) \geq \frac{16L'}{1-\gamma}$; the last inequality is because $\frac{1+\frac{1}{16}a}{1-\frac{1}{4}a} \leq 1 + \frac{1}{2}a$ for $a \in [0, 1]$.

$\square$

### D.5   Part IV. Combining

In this subsection, we combine previous lemmas to show last-iterate convergence rate of Algorithm 2 and prove Theorem 2.

**Lemma 21.** *With probability at least* $1 - \mathcal{O}(\delta)$, *for any time* $t \geq 1$,

$$\max_{s,x,y} \left( V_{x_t, y}^s - V_{x, y_t}^s \right) \leq \mathcal{O}\left( \frac{AL^{2+1/\varepsilon} \ln^{4+1/\varepsilon}(SAt/\delta) \ln^{1/\varepsilon}(t/(1-\gamma))}{(1-\gamma)^{2+1/\varepsilon}} t^{-\frac{1}{9+\varepsilon}} \right).$$

*Proof.* Using Lemma 10, we can bound the duality gap of the whole game by the duality gap on an individual state:

$$\max_{s,x,y} \left( V_{x_t, y}^s - V_{x, y_t}^s \right) \leq \frac{2}{1-\gamma} \max_{s,x,y} \left( x^s Q_\star^s y^s - x^s Q_\star^s y_t^s \right)$$

$$= \frac{2}{1-\gamma} \max_{s,x,y} \left( x_t^s \left( G^s + \gamma \mathbb{E}_{s' \sim P^s} \left[ V_\star^{s'} \right] \right) y^s - x^s \left( G^s + \gamma \mathbb{E}_{s' \sim P^s} \left[ V_\star^{s'} \right] \right) y_t^s \right)$$

$$\leq \frac{2}{1-\gamma} \max_{s,x,y} \left( x_t^s \left( G^s + \gamma \mathbb{E}_{s' \sim P^s} \left[ V_t^{s'} \right] \right) y^s - x^s \left( G^s + \gamma \mathbb{E}_{s' \sim P^s} \left[ V_t^{s'} \right] \right) y_t^s \right)$$

$$+ \frac{4\gamma}{1-\gamma} \max_s |V_t^s - V_\star^s|$$

With probability at least $1 - \mathcal{O}(\delta)$, for any $s, x^s, y^s$, and $t \geq 1$, denote $\tau$ the number of visitation to state $s$ until time $t$, then

$$x_t^s \left( G^s + \gamma \mathbb{E}_{s' \sim P^s} \left[ V_t^{s'} \right] \right) y^s - x^s \left( G^s + \gamma \mathbb{E}_{s' \sim P^s} \left[ V_t^{s'} \right] \right) y_t^s$$

$$= \hat{x}_\tau^s \left( G^s + \gamma \mathbb{E}_{s' \sim P^s} \left[ V_t^{s'} \right] \right) y^s - x^s \left( G^s + \gamma \mathbb{E}_{s' \sim P^s} \left[ V_t^{s'} \right] \right) \hat{y}_\tau^s$$

$$\leq \hat{x}_\tau^s \left( G^s + \gamma \mathbb{E}_{s' \sim P^s} \left[ V_{t_\tau(s)}^{s'} \right] \right) y^s - x^s \left( G^s + \gamma \mathbb{E}_{s' \sim P^s} \left[ V_{t_\tau(s)}^{s'} \right] \right) \hat{y}_\tau^s + 2 \max_{s'} |V_t^{s'} - V_{t_\tau(s)}^{s'}|$$

$$\leq \hat{x}_{\tau\star}^s \left( G^s + \gamma \mathbb{E}_{s' \sim P^s} \left[ V_{t_\tau(s)}^{s'} \right] \right) y^s - x^s \left( G^s + \gamma \mathbb{E}_{s' \sim P^s} \left[ V_{t_\tau(s)}^{s'} \right] \right) \hat{y}_{\tau\star}^s + 2 \max_{s'} |V_t^{s'} - V_{t_\tau(s)}^{s'}| + \mathcal{O}(\|\hat{z}_\tau^s - \hat{z}_{\tau\star}^s\|_1)$$

$$\leq 2\epsilon_\tau \ln(A) + \mathcal{O}\left( \frac{1}{\tau} \right) + 2 \max_{s'} |V_t^{s'} - V_{t_\tau(s)}^{s'}| + \mathcal{O}(\sqrt{\mathrm{KL}(\hat{z}_\tau^s, \hat{z}_{\tau\star}^s)}) \qquad \text{(Lemma 9)}$$

$$\leq \mathcal{O}\left( \frac{\ln(A)}{1-\gamma} \tau^{-k_\epsilon} \right) + \mathcal{O}\left( \frac{L \ln(St/\delta)\tau^{-k_\alpha}}{1-\gamma} \right) + \mathcal{O}\left( \sqrt{A \ln^5(SAt/\delta)L^2} \tau^{-\frac{k_\sharp}{2}} \right). \quad \text{(Lemma 17)}$$

Combing the above two inequality with Lemma 17 and Lemma 20 and the choice of parameters $k_\alpha = \frac{9}{9+\varepsilon}$, $k_\varepsilon = \frac{1}{9+\varepsilon}$, $k_\beta = \frac{3}{9+\varepsilon}$, and $k_\eta = \frac{5}{9+\varepsilon}$, we have $k_\sharp = \min\{k_\beta - k_\epsilon, k_\eta - k_\beta, k_\alpha - k_\eta - 2k_\epsilon\} = \frac{2}{9+\varepsilon}$, $k_* = \min\{k_\eta, k_\beta, k_\alpha - k_\beta, k_\epsilon\} = \frac{1}{9+\varepsilon}$, and

$$\max_{s,x,y} \left( V_{x_t, y}^s - V_{x, y_t}^s \right)$$

$$\leq \frac{2}{1-\gamma} \max_{s,x,y} \left( x_t^s \left( G^s + \gamma \mathbb{E}_{s' \sim P^s} \left[ V_t^{s'} \right] \right) y^s - x^s \left( G^s + \gamma \mathbb{E}_{s' \sim P^s} \left[ V_t^{s'} \right] \right) y_t^s \right) + \frac{4\gamma}{1-\gamma} \max_s |V_t^s - V_\star^s|$$

$$\leq \mathcal{O}\left( \frac{\sqrt{A} \ln^{5/2}(SAt/\delta)}{(1-\gamma)^2} L\tau^{-\min\{k_\epsilon, \frac{k_\sharp}{2}, k_\alpha\}} \right) + \mathcal{O}\left( \frac{A \ln^3(SAt/\delta)}{(1-\gamma)^3} \left( \frac{L \ln(St/\delta)}{1-\gamma} \ln \frac{t}{1-\gamma} \right)^{\frac{k_*}{1-k_\alpha}} \left( \frac{L \ln(St/\delta)}{t} \right)^{k_*} \right)$$

$$= \mathcal{O}\left(\frac{AL^{2+1/\varepsilon}\ln^{4+1/\varepsilon}(SAt/\delta)\ln^{1/\varepsilon}(t/(1-\gamma))}{(1-\gamma)^{3+1/\varepsilon}}\cdot t^{-\frac{1}{9+\varepsilon}}\right). \qquad (\tfrac{t}{L}\le\tau\le t)$$

$\square$

# E Convergent Analysis of Algorithm 3

## E.1 Part I. Basic Iteration Properties

**Definition 1.** *Let $t_\tau(s)$ be the $\tau$-th time the players visit state $s$. Define $\hat{x}_\tau^s = x_{t_\tau(s)}^s$, $\hat{y}_\tau^s = y_{t_\tau(s)}^s$, $\hat{a}_\tau^s = a_{t_\tau(s)}$, $\hat{b}_\tau^s = b_{t_\tau(s)}$,. Furthermore, define*

$$\underline{f}_\tau^s(x,y) \triangleq x^\top \left(G^s + \gamma\mathbb{E}_{s'\sim P^s}\left[\underline{V}_{t_\tau(s)}^{s'}\right]\right)y - \epsilon\phi(x) + \epsilon\phi(y),$$

$$\overline{f}_\tau^s(x,y) \triangleq x^\top \left(G^s + \gamma\mathbb{E}_{s'\sim P^s}\left[\overline{V}_{t_\tau(s)}^{s'}\right]\right)y - \epsilon\phi(x) + \epsilon\phi(y).$$

**Lemma 22.** *For any $x^s \in \Omega$,*

$$\underline{f}_\tau^s(\hat{x}_\tau^s, \hat{y}_\tau^s) - \underline{f}_\tau^s(x^s, \hat{y}_\tau^s)$$

$$\le \frac{(1-\eta\epsilon)KL(x^s,\hat{x}_\tau^s) - KL(x^s,\hat{x}_{\tau+1}^s)}{\eta} + \frac{10\eta A\ln^2(AT)}{(1-\gamma)^2} + \frac{2\eta A}{(1-\gamma)^2}\underline{\lambda}_\tau^s + \underline{\xi}_\tau^s + \underline{\zeta}_\tau^s(x^s).$$

*where*

$$\underline{\lambda}_\tau^s = \frac{1}{A}\sum_a\left(\frac{\mathbf{1}[\hat{a}_\tau^s=a]}{\hat{x}_{\tau,a}^s+\beta} - 1\right),$$

$$\underline{\xi}_\tau^s = (\hat{x}_\tau^s)^\top\left(\left(G^s + \gamma\mathbb{E}_{s'\sim P^s}\left[\underline{V}_t^{s'}\right]\right)\hat{y}_\tau^s - \frac{\mathbf{1}[\hat{a}_\tau^s=a]\left(\sigma_t + \gamma\underline{V}_t^{s_{t+1}}\right)}{\hat{x}_{\tau,a}^s+\beta}\right),$$

$$\underline{\zeta}_\tau^s(x^s) = (x^s)^\top\left(\frac{\mathbf{1}[\hat{a}_\tau^s=a]\left(\sigma_t + \gamma\underline{V}_t^{s_{t+1}}\right)}{\hat{x}_{\tau,a}^s+\beta} - \left(G^s + \gamma\mathbb{E}_{s'\sim P^s}\left[\underline{V}_t^{s'}\right]\right)\hat{y}_\tau^s\right).$$

*Proof.* The proof is exactly the same as that of Lemma 16. $\qquad\square$

## E.2 Part II. Value Convergence

**Lemma 23** (weighted regret bound). *There exists a large enough universal constant $\kappa$ (used in the definition of $\mathsf{bns}_\tau$) such that with probability $1 - \mathcal{O}(\delta)$, for any state $s$, visitation count $\tau$, and any $x^s \in \Omega$,*

$$\sum_{i=1}^\tau \alpha_\tau^i\left(\underline{f}_i^s(\hat{x}_i^s, \hat{y}_i^s) - \underline{f}_i^s(x^s, \hat{y}_i^s)\right) \le \frac{1}{2}\mathsf{bns}_\tau.$$

*Proof.* Fix state $s$ and visitation count $\tau \le T$. Applying Lemma 22 and considering the weighted sum of the bounds, we get

$$\sum_{i=1}^\tau \alpha_\tau^i\left(\underline{f}_i^s(\hat{x}_i^s, \hat{y}_i^s) - \underline{f}_i^s(x^s, \hat{y}_i^s)\right)$$

$$\le \sum_{i=1}^\tau \alpha_\tau^i\left(\frac{(1-\eta\epsilon)KL(x^s,\hat{x}_i^s) - KL(x^s,\hat{x}_{i+1}^s)}{\eta} + \frac{10\eta A\ln^2(AT)}{(1-\gamma)^2} + \frac{2\eta A}{(1-\gamma)^2}\underline{\lambda}_i^s + \underline{\xi}_i^s + \underline{\zeta}_i^s(x^s)\right)$$

$$\le \underbrace{\frac{\alpha_\tau^1(1-\eta\epsilon)}{\eta}KL(x^s,\hat{x}_1^s) + \sum_{i=2}^\tau\left(\frac{\alpha_\tau^i(1-\eta\epsilon)}{\eta}KL(x^s,\hat{x}_i^s) - \frac{\alpha_\tau^{i-1}}{\eta}KL(x^s,\hat{x}_i^s)\right)}_{\mathbf{term}_0}$$

$$+ \frac{10\eta A \ln^2(AT)}{(1-\gamma)^2} \underbrace{\sum_{i=1}^{\tau} \alpha_\tau^i}_{\textbf{term}_1} + \frac{2\eta A}{(1-\gamma)^2} \underbrace{\sum_{i=1}^{\tau} \alpha_\tau^i \underline{\lambda}_i^s}_{\textbf{term}_2} + \underbrace{\sum_{i=1}^{\tau} \alpha_\tau^i \underline{\xi}_i^s}_{\textbf{term}_3} + \underbrace{\sum_{i=1}^{\tau} \alpha_\tau^i \underline{\zeta}_i^s(x^s)}_{\textbf{term}_4}.$$

Since $\alpha_\tau^{i-1} = \frac{\alpha_{i-1}(1-\alpha_i)}{\alpha_i}\alpha_\tau^i \geq (1-\alpha_i)\alpha_\tau^i$,

$$\textbf{term}_0 \leq \frac{\alpha_\tau^1}{\eta}\text{KL}(x^s, \hat{x}_1^s) + \sum_{i=2}^{\tau}\text{KL}(x^s, \hat{x}_i^s)\left(\frac{\alpha_\tau^i(1-\eta\epsilon)}{\eta} - \frac{\alpha_\tau^i(1-\alpha_i)}{\eta}\right)$$

$$\leq \ln(AT)\left(\frac{\alpha_\tau^1}{\eta} + \sum_{i=2}^{\tau}\frac{\alpha_\tau^i\alpha_i}{\eta}\right) = \mathcal{O}\left(\frac{\ln(AT)\alpha_\tau}{\eta}\right).$$

We proceed to bound other terms as follows: wiht probability at least $1 - \frac{\delta}{S\tau^2}$

$$\textbf{term}_1 = \mathcal{O}\left(\frac{A\ln^2(AT)\eta}{(1-\gamma)^2}\right), \qquad\qquad (\sum_{i=1}^{\tau}\alpha_\tau^i = 1)$$

$$\textbf{term}_2 = \frac{2\eta A}{(1-\gamma)^2}\sum_{i=1}^{\tau}\alpha_\tau^i\left(\frac{1}{A}\sum_a\left(\frac{\mathbf{1}[\hat{a}_i^s = a]}{\hat{x}_{i,a}^s + \beta} - 1\right)\right)$$

$$\leq \frac{2A}{(1-\gamma)^2}\times\mathcal{O}\left(\ln(AS\tau/\delta)\max_{i\leq\tau}\frac{\alpha_\tau^i\eta}{\beta}\right) \qquad\qquad \text{(by Lemma 7)}$$

$$= \mathcal{O}\left(\frac{A\ln(AST/\delta)\alpha_\tau}{(1-\gamma)^2}\times\frac{\eta}{\beta}\right),$$

$$\textbf{term}_3 = \mathcal{O}\left(\frac{A}{1-\gamma}\sum_{i=1}^{\tau}\beta\alpha_\tau^i + \frac{1}{1-\gamma}\sqrt{\ln(AS\tau/\delta)\sum_{i=1}^{\tau}(\alpha_\tau^i)^2}\right) \qquad \text{(by Lemma 6)}$$

$$= \mathcal{O}\left(\frac{A\beta}{1-\gamma} + \frac{1}{1-\gamma}\sqrt{\ln(AS\tau/\delta)\sum_{i=1}^{\tau}\alpha_\tau^i\alpha_i}\right)$$

$$= \mathcal{O}\left(\frac{A\ln(AST/\delta)(\beta + \alpha_\tau)}{1-\gamma}\right),$$

$$\textbf{term}_4 = \mathcal{O}\left(\frac{\ln(AS\tau/\delta)}{1-\gamma}\max_{i\leq\tau}\frac{\alpha_\tau^i}{\beta}\right) = \mathcal{O}\left(\frac{\ln(AST/\delta)\alpha_\tau}{(1-\gamma)\beta}\right).$$

Combining all terms and applying a union bound over $s \in \mathcal{S}$ and $\tau$, we get with probability $1 - \mathcal{O}(\delta)$ such that for any $s \in \mathcal{S}$, visitation count $\tau$, and $x^s \in \Omega$,

$$\sum_{i=1}^{\tau}\alpha_\tau^i\left(\underline{f}_{-i}^s(\hat{x}_i^s, \hat{y}_i^s) - \underline{f}_{-i}^s(x^s, \hat{y}_i^s)\right) = \mathcal{O}\left(\frac{A\ln^2(AST/\delta)\left(\eta + \beta + (\eta^{-1} + \beta^{-1})\alpha_\tau\right)}{(1-\gamma)^2}\right)$$

$$= \mathcal{O}\left(\frac{A\ln^2(SAT/\delta)(\beta + \alpha_\tau/\eta)}{(1-\gamma)^2}\right). \qquad \text{(using } \eta \leq \beta\text{)}$$

This implies the conclusion of the lemma. $\qquad\square$

**Lemma 24.** *For all $t, s$, $\overline{V}_t^s \geq \underline{V}_t^s$.*

*Proof.* We prove it by induction on $t$. The inequality clearly holds for $t = 1$ by the initialization. Suppose that the inequality holds for $1, 2, \ldots, t-1$ and for all $s$. Now consider time $t$ and state $s$. Let $\tau = n_t^s$, and let $1 \leq t_1 < t_2 < \ldots < t_\tau < t$ be the time indices when the players visit state $s$. By the update rule,

$$\tilde{V}_t^s - \underline{V}_t^s = \sum_{i=1}^{\tau}\alpha_\tau^i\left(\gamma\overline{V}_{t_i}^{s_{t_i+1}} - \gamma\underline{V}_{t_i}^{s_{t_i+1}} + 2\mathsf{bns}_i\right) > 0$$

where the inequality is by the induction hypothesis. Therefore,

$$\overline{V}_t^s - \underline{V}_t^s = \min\left\{\tilde{V}_t^s, H\right\} - \max\left\{\underaccent{\tilde}{V}_t^s, 0\right\} > 0.$$

In the last inequality we also use the fact that $\underaccent{\tilde}{V}_t^s \leq H$ and $\tilde{V}_t^s \geq 0$. Note that by the induction hypothesis and the update rule of $\overline{V}_t^s$ and $\underline{V}_t^s$, we have $0 \leq \underline{V}_i^s < \overline{V}_i^s \leq H$ for all $s$ and $1 \leq i \leq t-1$. Thus $\underaccent{\tilde}{V}_t^s = \sum_{i=1}^\tau \alpha_\tau^i(\gamma \underline{V}_{t_i}^{s_{t_i}+1} - \mathsf{bns}_i) \leq H$ and similarly $\tilde{V}_t^s \geq 0$. $\qquad\square$

**Lemma 25.** *Let $c = (c_1, \ldots, c_T)$ be any non-negative sequence with $c_i \leq c_{\max} \forall i$ and $\sum_{t=1}^T c_t = C$. Then*

$$\sum_{t=1}^T c_t\left(\overline{V}_t^{s_{t+1}} - \underline{V}_t^{s_{t+1}}\right) \leq \mathcal{O}\left(\frac{CA\ln^3(AST/\delta)\beta}{(1-\gamma)^3} + \frac{c_{\max}AS\ln^4(AST/\delta)}{\eta(1-\gamma)^3}\right).$$

*Proof.*

$$Z_c \triangleq \sum_{t=1}^T c_t\left(\overline{V}_t^{s_{t+1}} - \underline{V}_t^{s_{t+1}}\right)$$

$$\leq \sum_{t=1}^T c_t\left(\overline{V}_{t+1}^{s_{t+1}} - \underline{V}_{t+1}^{s_{t+1}}\right) + \mathcal{O}\left(c_{\max}\sum_s\sum_{t=1}^T\left(\left|\overline{V}_{t+1}^s - \overline{V}_t^s\right| + \left|\underline{V}_{t+1}^s - \underline{V}_t^s\right|\right)\right)$$

$$\leq \sum_{t=1}^T c_{t-1}\left(\overline{V}_t^{s_t} - \underline{V}_t^{s_t}\right) + \mathcal{O}\left(c_{\max}\sum_s\sum_{i=1}^T\frac{\alpha_i}{1-\gamma}\right) \qquad \text{(shifting the indices and define } c_0 = 0)$$

$$\leq \sum_{t=1}^T c_{t-1}\left(\tilde{V}_t^{s_t} - \underaccent{\tilde}{V}_t^{s_t}\right) + \mathcal{O}\left(\frac{c_{\max}S}{1-\gamma} \times H\ln T\right) \qquad \text{(using } \alpha_i = \tfrac{H+1}{H+i})$$

$$= \sum_s\sum_{\tau=1}^{n_{T+1}(s)} c_{t_\tau(s)-1}\left(\tilde{V}_{t_\tau(s)}^s - \underaccent{\tilde}{V}_{t_\tau(s)}^s\right) + \mathcal{O}\left(\frac{c_{\max}S\ln^2 T}{(1-\gamma)^2}\right) \qquad (H = \tfrac{\ln T}{1-\gamma})$$

$$= \gamma\sum_s\sum_{\tau=1}^{n_{T+1}(s)} c_{t_\tau(s)-1}\sum_{i=1}^{\tau-1}\alpha_{\tau-1}^i\left(\overline{V}_{t_i(s)}^{s_{t_i(s)}+1} - \underline{V}_{t_i(s)}^{s_{t_i(s)}+1} + 2\mathsf{bns}_i\right) + \mathcal{O}\left(\frac{c_{\max}S\ln^2 T}{(1-\gamma)^2}\right)$$

$$\leq \gamma\sum_s\sum_{i=1}^{n_{T+1}(s)-1}\underbrace{\left(\sum_{\tau=i+1}^{n_{T+1}(s)}\alpha_{\tau-1}^i c_{t_\tau(s)-1}\right)}_{c'_{t_i(s)}}\left(\overline{V}_{t_i(s)}^{s_{t_i(s)}+1} - \underline{V}_{t_i(s)}^{s_{t_i(s)}+1}\right)$$

$$+ \mathcal{O}\left(\sum_s\sum_{\tau=2}^{n_{T+1}(s)} c_{t_\tau(s)-1}\mathsf{bns}_{\tau-1} + \frac{c_{\max}S\ln^2 T}{(1-\gamma)^2}\right)$$

$$\leq \gamma\sum_{t=1}^T c'_t\left(\overline{V}_t^{s_{t+1}} - \underline{V}_t^{s_{t+1}}\right) + \mathcal{O}\left(\sum_s\sum_{\tau=1}^{C_s/c_{\max}} c_{\max}\mathsf{bns}_\tau + \frac{c_{\max}S\ln^2 T}{(1-\gamma)^2}\right)$$

$$\hspace{8cm} (C_s \triangleq \sum_{\tau=1}^{n_{T+1}(s)} c_{t_\tau(s)-1})$$

$$\leq \gamma\sum_{t=1}^T c'_t\left(\overline{V}_t^{s_{t+1}} - \underline{V}_t^{s_{t+1}}\right)$$

$$+ \mathcal{O}\left(\sum_s\sum_{\tau=1}^{C_s/c_{\max}}\frac{c_{\max}A\ln^2(AST/\delta)(\beta + \alpha_\tau/\eta)}{(1-\gamma)^2} + \frac{c_{\max}\ln^2 T}{(1-\gamma)^2}\right)$$

$$\leq \gamma\sum_{t=1}^T c'_t\left(\overline{V}_t^{s_{t+1}} - \underline{V}_t^{s_{t+1}}\right) + \mathcal{O}\left(\frac{CA\ln^2(AST/\delta)\beta}{(1-\gamma)^2} + \frac{c_{\max}AS\ln^3(AST/\delta)}{\eta(1-\gamma)^2}\right). \qquad (19)$$

Note that $c'_t$ is another sequence with

$$c'_i \le c'_{\max} \le c_{\max} \sup_i \sum_{\tau=i}^{\infty} \alpha_\tau^i \le \left(1 + \frac{1}{H}\right) c_{\max}$$

and

$$\sum_{t=1}^{T} c'_t \le \sum_{t=1}^{T} c_t = C$$

since $\sum_{i=1}^{\tau} \alpha_\tau^i = 1$ for any $\tau \ge 1$. Thus, we can unroll the inequality Eq. (19) for $H$ times, which gives

$$Z_c \le \gamma^H \left(1 + \frac{1}{H}\right)^H \frac{c_{\max} T}{1 - \gamma} + H \times \mathcal{O}\left(\frac{CA \ln^2(AST/\delta)\beta}{(1-\gamma)^2} + \frac{c_{\max} AS \ln^3(AST/\delta)}{\eta(1-\gamma)^2}\right)$$

$$= \mathcal{O}\left(\frac{CA \ln^3(AST/\delta)\beta}{(1-\gamma)^3} + \frac{c_{\max} AS \ln^4(AST/\delta)}{\eta(1-\gamma)^3}\right)$$

where in the inequality we use that $(1 + \frac{1}{H})^H \le e$ and $\gamma^H = (1 - (1-\gamma))^H \le e^{-(1-\gamma)H} = \frac{1}{T}$. $\square$

**Corollary 2.** *There exists a universal constant $C_1 > 0$ such that for any $\tilde{\epsilon} \ge \frac{C_1 A \ln^3(AST/\delta)\beta}{(1-\gamma)^3}$, with probability at least $1 - \mathcal{O}(\delta)$,*

$$\sum_{t=1}^{T} \mathbf{1}\left[x_t^{s_t\top} \left(\mathbb{E}_{s' \sim P^{s_t}}\left[\overline{V}_t^{s'} - \underline{V}_t^{s'}\right]\right) y_t^{s_t} \ge \tilde{\epsilon}\right] \le \mathcal{O}\left(\frac{AS \ln^4(AST/\delta)}{\eta\tilde{\epsilon}(1-\gamma)^3}\right).$$

*Proof.* We apply Lemma 25 with the following definition of $c_t$:

$$c_t = \mathbf{1}\left[x_t^{s_t} \left(\mathbb{E}_{s' \sim P^{s_t}}\left[\overline{V}_t^{s'} - \underline{V}_t^{s'}\right]\right) y_t^{s_t} \ge \tilde{\epsilon}\right],$$

which gives

$$\sum_{t=1}^{T} c_t \left(\overline{V}_t^{s_{t+1}} - \underline{V}_t^{s_{t+1}}\right) \le C_2 \times \left(\frac{CA \ln^3(AST/\delta)\beta}{(1-\gamma)^3} + \frac{AS \ln^4(AST/\delta)}{\eta(1-\gamma)^3}\right) \qquad (20)$$

for some universal constant $C_2$ and $C = \sum_{t=1}^{T} c_t$. By Azuma's inequality, for some universal constant $C_3 > 0$, with probability $1 - \delta$,

$$\sum_{t=1}^{T} c_t x_t^{s_t} \left(\mathbb{E}_{s' \sim P^{s_t}}\left[\overline{V}_t^{s'} - \underline{V}_t^{s'}\right]\right) y_t^{s_t} - \sum_{t=1}^{T} c_t \left(\overline{V}_t^{s_{t+1}} - \underline{V}_t^{s_{t+1}}\right)$$

$$\le \frac{C_3}{1-\gamma} \sqrt{\ln(S/\delta) \sum_{t=1}^{T} c_t^2} = \frac{C_3}{1-\gamma} \sqrt{\ln(S/\delta)C}$$

$$\le C_3 \times \frac{C\beta}{1-\gamma} + C_3 \times \frac{\ln(S/\delta)}{\eta(1-\gamma)}. \qquad \text{(by AM-GM and that } \eta \le \beta\text{)}$$

$$(21)$$

Combining Eq. (20) and Eq. (21), we get

$$\sum_{t=1}^{T} c_t x_t^{s_t} \left(\mathbb{E}_{s' \sim P^{s_t}}\left[\overline{V}_t^{s'} - \underline{V}_t^{s'}\right]\right) y_t^{s_t}$$

$$\le (C_2 + C_3) \times \left(\frac{CA \ln^3(AST/\delta)\beta}{(1-\gamma)^3} + \frac{AS \ln^4(AST/\delta)}{\eta(1-\gamma)^3}\right).$$

By the definition of $c_t$, the left-hand side above is lower bounded by $\tilde{\epsilon}\sum_{t=1}^{T} c_t = C\tilde{\epsilon}$. Define $C_1 = 2(C_2 + C_3)$. Then by the condition on $\epsilon'$, the right-hand side above is above inequality is bounded by

$$\frac{C\tilde{\epsilon}}{2} + \frac{C_1}{2}\left(\frac{AS\ln^4(AST/\delta)}{\eta(1-\gamma)^3}\right)$$

by the condition on $\tilde{\epsilon}$. Combining the upper bound and the lower bound, we get

$$C \leq C_1 \times \left(\frac{AS\ln^4(AST/\delta)}{\eta\tilde{\epsilon}(1-\gamma)^3}\right).$$

□

**Lemma 26.** *With probability at least $1 - \mathcal{O}(\delta)$, for any $t \geq 1$,*

$$\underline{V}_t^s \leq V_\star^s + \mathcal{O}\left(\frac{\epsilon\ln(AT)}{1-\gamma}\right), \qquad \overline{V}_t^s \geq V_\star^s - \mathcal{O}\left(\frac{\epsilon\ln(AT)}{1-\gamma}\right).$$

*Proof.* Fix a $t$ and $s$, let $\tau = n_t(s)$, and let $t_i$ be the time index in which $s$ is visited the $i$-th time. With probability at least $1 - \frac{\delta}{ST}$, we have

$$\underline{V}_t^s = \sum_{i=1}^{\tau} \alpha_\tau^i \left(\sigma_{t_i} + \gamma\underline{V}_{t_i}^{s_{t_i}+1} - \mathsf{bns}_i\right)$$

$$\leq \sum_{i=1}^{\tau} \alpha_\tau^i \left(\underline{f}_{-i}^s(x_{t_i}^s, y_{t_i}^s) + \epsilon\phi(x_{t_i}^s) - \epsilon\phi(y_{t_i}^s)\right) + \sum_{i=1}^{\tau} \alpha_\tau^i \left(\sigma_{t_i} + \gamma\underline{V}_{t_i}^{s_{t_i}+1} - x_{t_i}^{s\top}(G^s + \mathbb{E}_{s'\sim P^{s_t}}[V_{t_i}^{s'}])y_{t_i}^s\right)$$

$$- \sum_{i=1}^{\tau} \alpha_\tau^i\mathsf{bns}_i$$

$$\leq \sum_{i=1}^{\tau} \alpha_\tau^i \left(\underline{f}_{-i}^s(x_{t_i}^s, y_{t_i}^s) + \epsilon\phi(x_{t_i}^s) - \epsilon\phi(y_{t_i}^s)\right) + \frac{1}{1-\gamma}\sqrt{2\sum_{i=1}^{\tau}(\alpha_\tau^i)^2\log(ST/\delta)} - \sum_{i=1}^{\tau} \alpha_\tau^i\mathsf{bns}_i$$

(by Hoeffding's inequality)

$$\leq \sum_{i=1}^{\tau} \alpha_\tau^i \left(\underline{f}_{-i}^s(x_{t_i}^s, y_{t_i}^s) + \epsilon\phi(x_{t_i}^s) - \epsilon\phi(y_{t_i}^s)\right) + \frac{1}{1-\gamma}\sqrt{2\alpha_\tau\log(ST/\delta)} - \sum_{i=1}^{\tau} \alpha_\tau^i\mathsf{bns}_i$$

$(\sum_{i=1}^{\tau}\alpha_\tau^i \leq 1)$

$$\leq \sum_{i=1}^{\tau} \alpha_\tau^i \left(\underline{f}_{-i}^s(x_{t_i}^s, y_{t_i}^s) + \epsilon\phi(x_{t_i}^s) - \epsilon\phi(y_{t_i}^s)\right) - \frac{1}{2}\mathsf{bns}_\tau$$

$(\sum_{i=1}^{\tau}\alpha_\tau^i \geq \frac{1}{2}$ and $\mathsf{bns}_\tau$ is decreasing, and $\sqrt{\alpha_\tau} \leq \frac{\alpha_\tau}{\eta} + \eta \leq \frac{\alpha_\tau}{\eta} + \beta)$

$$\leq \min_{x\in\Omega} \sum_{i=1}^{\tau} \alpha_\tau^i \left(\underline{f}_{-i}^s(x^s, y_{t_i}^s)\right) + \sum_{i=1}^{\tau} \alpha_\tau^i \left(\epsilon\phi(x_{t_i}^s) - \epsilon\phi(y_{t_i}^s)\right)$$

(by Lemma 23)

$$\leq \min_{x\in\Omega} \sum_{i=1}^{\tau} \alpha_\tau^i (x^s)^\top \left(G^s + \gamma\mathbb{E}_{s'\sim P^s}\left[\underline{V}_{t_i}^{s'}\right]\right)y_{t_i}^s + \mathcal{O}(\epsilon\ln(AT)). \quad (x_a \geq \frac{1}{AT} \text{ for any } x \in \Omega.)$$

Therefore, using a union bound over $s$ and $t$, we have with probability $1 - \delta$, for all $s$ and $t$,

$$\underline{V}_t^s = \max\{\underline{V}_t^s, 0\} \leq \min_x \sum_{i=1}^{\tau} \alpha_\tau^i (x^s)^\top \left(G^s + \gamma\mathbb{E}_{s'\sim P^s}\left[\underline{V}_{t_i}^{s'}\right]\right)y_{t_i}^s + C_4\epsilon\ln(AT) \qquad (22)$$

for some universal constant $C_4$. Next, we use induction to show the first inequality. Suppose that

$$\underline{V}_{t'}^s \leq V_\star^s + \frac{C_4\epsilon\ln(AT)}{1-\gamma}$$

for all $s$ and $t' < t$. Then by Eq. (22),

$$
\underline{V}_t^s \leq \min_x \sum_{i=1}^{\tau} \alpha_\tau^i \, (x^s)^\top \left( G^s + \gamma \mathbb{E}_{s' \sim P^s} \left[ V_\star^{s'} + \frac{C_4 \epsilon \ln(AT)}{1 - \gamma} \right] \right) y_{t_i}^s + C_4 \epsilon \ln(AT)
$$

$$
= \min_x \sum_{i=1}^{\tau} \alpha_\tau^i \, (x^s)^\top \left( G^s + \gamma \mathbb{E}_{s' \sim P^s} \left[ V_\star^{s'} \right] \right) y_{t_i}^s + \frac{C_4 \epsilon \ln(AT)}{1 - \gamma}
$$

$$
\leq \min_x \sum_{i=1}^{\tau} \max_y \alpha_\tau^i \, (x^s)^\top \left( G^s + \gamma \mathbb{E}_{s' \sim P^s} \left[ V_\star^{s'} \right] \right) y^s + \frac{C_4 \epsilon \ln(AT)}{1 - \gamma}
$$

$$
= \min_x \max_y (x^s)^\top \left( G^s + \gamma \mathbb{E}_{s' \sim P^s} \left[ V_\star^{s'} \right] \right) y^s + \frac{C_4 \epsilon \ln(AT)}{1 - \gamma}
$$

$$
= V_\star^s + \frac{C_4 \epsilon \ln(AT)}{1 - \gamma},
$$

which proves the first desired inequality. The other inequality can be proven in the same way. $\qquad\square$

### E.3 Part III. Policy Convergence to the Nash of the Regularized Game

**Lemma 27.** *Let $0 \leq p \leq 1$ be arbitrarily chosen, and define*

$$
f_\tau^s(x^s, y^s) \triangleq p \underline{f}_\tau^s(x^s, y^s) + (1 - p) \overline{f}_\tau^s(x^s, y^s)
$$

$$
= x^{s\top} \left( G^s + \mathbb{E}_{s' \sim P^s} \left[ p \underline{V}_{t_\tau(s)}^{s'} + (1 - p) \overline{V}_{t_\tau(s)}^{s'} \right] \right) y^s - \epsilon \phi(x^s) + \epsilon \phi(y^s).
$$

*Furthermore, let $\hat{z}_{\tau\star}^s = (\hat{x}_{\tau\star}^s, \hat{y}_{\tau\star}^s)$ be the equilibrium of $f_\tau^s(x, y)$, and define $z_{t\star}^s = \hat{z}_{\tau\star}^s$ where $\tau = n_t(s)$. Then with probability at least $1 - \mathcal{O}(\delta)$, the following holds for any $0 < \epsilon' \leq 1$:*

$$
\sum_s \sum_{i=1}^{n_{T+1}(s)} \mathbf{1}\left[ KL(\hat{z}_{i\star}^s, \hat{z}_i^s) \geq \epsilon' \right] \leq \mathcal{O}\left( \frac{S^2 A \ln^5(SAT/\delta)}{\eta \epsilon^2 \epsilon'(1 - \gamma)^3} \right)
$$

*if $\eta$ and $\beta$ satisfy the following*

$$
\beta \leq \frac{C_5(1 - \gamma)^3}{A \ln^3(AST/\delta)} \epsilon \epsilon' \tag{23}
$$

$$
\eta \leq \frac{C_6(1 - \gamma)}{A \ln^3(AST/\delta)} \beta \epsilon' \tag{24}
$$

*with sufficiently small universal constant $C_5, C_6 > 0$.*

*Proof.* In this proof, we write $\underline{\zeta}_i^s(\hat{x}_{i\star}^s)$ as $\underline{\zeta}_i$. By Lemma 22, we have

$$
KL(\hat{x}_{i\star}^s, \hat{x}_{i+1}^s) \leq (1 - \eta\epsilon) KL(\hat{x}_{i\star}^s, \hat{x}_i^s) + \eta \left( \underline{f}_i^s(\hat{x}_{i\star}^s, \hat{y}_i^s) - \underline{f}_i^s(\hat{x}_i^s, \hat{y}_i^s) \right)
$$

$$
+ \frac{10\eta^2 A \ln^2(AT)}{(1 - \gamma)^2} + \frac{2\eta^2 A}{(1 - \gamma)^2} \underline{\lambda}_i^s + \eta \underline{\xi}_i^s + \eta \underline{\zeta}_i^s.
$$

Similarly,

$$
KL(\hat{y}_{i\star}^s, \hat{y}_{i+1}^s) \leq (1 - \eta\epsilon) KL(\hat{y}_{i\star}^s, \hat{y}_i^s) + \eta \left( \overline{f}_i^s(\hat{x}_i^s, \hat{y}_i^s) - \overline{f}_i^s(\hat{x}_i^s, \hat{y}_{i\star}^s) \right)
$$

$$
+ \frac{10\eta^2 A \ln^2(AT)}{(1 - \gamma)^2} + \frac{2\eta^2 A}{(1 - \gamma)^2} \overline{\lambda}_i^s + \eta \overline{\xi}_i^s + \eta \overline{\zeta}_i^s.
$$

Adding the two inequalities up, we get

$$
KL(\hat{z}_{i+1\star}^s, \hat{z}_{i+1}^s)
$$

$$
\leq (1 - \eta\epsilon) KL(\hat{z}_{i\star}^s, \hat{z}_i^s) + \frac{20\eta^2 A \ln^2(AT)}{(1 - \gamma)^2} + \frac{2\eta^2 A}{(1 - \gamma)^2} \lambda_i^s + \eta \xi_i^s + \eta \zeta_i^s + v_i^s
$$

$$+ \eta \left( \overline{f}_i^s(\hat{x}_i^s, \hat{y}_i^s) - \underline{f}_i^s(\hat{x}_i^s, \hat{y}_i^s) + \underline{f}_i^s(\hat{x}_{i\star}^s, \hat{y}_i^s) - \overline{f}_i^s(\hat{x}_i^s, \hat{y}_{i\star}^s) \right) \tag{25}$$

where $v_i^s = \mathrm{KL}(\hat{z}_{i+1\star}^s, \hat{z}_{i+1}^s) - \mathrm{KL}(\hat{z}_{i\star}^s, \hat{z}_{i+1}^s)$ and $\square^s = \underline{\square}^s + \overline{\square}^s$. By Lemma 24, we have $\underline{f}_i^s(x,y) \leq \overline{f}_i^s(x,y)$ for all $x, y$, and thus $\underline{f}_i^s(\hat{x}_{i\star}^s, \hat{y}_i^s) - \overline{f}_i^s(\hat{x}_i^s, \hat{y}_{i\star}^s) \leq f_i^s(\hat{x}_{i\star}^s, \hat{y}_i^s) - f_i^s(\hat{x}_i^s, \hat{y}_{i\star}^s) \leq 0$. Therefore, Eq. (25) further implies

$\mathrm{KL}(\hat{z}_{i+1\star}^s, \hat{z}_{i+1}^s)$

$$\leq (1 - \eta\epsilon)\mathrm{KL}(\hat{z}_{i\star}^s, \hat{z}_i^s) + \frac{20\eta^2 A \ln^2(AT)}{(1-\gamma)^2} + \frac{2\eta^2 A}{(1-\gamma)^2}\lambda_i^s + \eta\xi_i^s + \eta\zeta_i^s + v_i^s + \eta\Delta_i^s$$

$$\leq (1 - \eta\epsilon)\mathrm{KL}(\hat{z}_{i\star}^s, \hat{z}_i^s) + \frac{20\eta^2 A \ln^2(AT)}{(1-\gamma)^2} + \frac{2\eta^2 A}{(1-\gamma)^2}\lambda_i^s + \eta\xi_i^s + \eta\zeta_i^s + v_i^s + \frac{1}{2}\eta\epsilon\epsilon' + \left[\eta\Delta_i^s - \frac{1}{2}\eta\epsilon\epsilon'\right]_+$$

where $\Delta_i^s = \overline{f}_i^s(\hat{x}_i^s, \hat{y}_i^s) - \underline{f}_i^s(\hat{x}_i^s, \hat{y}_i^s)$ and in the last step we use $a \leq [a - b]_+ + b$.

Unrolling the recursion, we get with probability at least $1 - \mathcal{O}(\delta)$, for all $s$ and $\tau$ (we show that the inequality holds for any fix $s$ and $\tau$ with probability $1 - \mathcal{O}(\frac{\delta}{ST})$ and then apply the union bound over $s$ and $\tau$),

$\mathrm{KL}(\hat{z}_{\tau+1\star}^s, \hat{z}_{\tau+1}^s)$

$$\leq (1 - \eta\epsilon)^\tau \mathrm{KL}(\hat{z}_{1\star}^s, \hat{z}_1^s) + \underbrace{\frac{20\eta^2 A \ln^2(AT)}{(1-\gamma)^2}\sum_{i=1}^\tau (1-\eta\epsilon)^{\tau-i}}_{\textbf{term}_1} + \underbrace{\frac{2\eta^2 A}{(1-\gamma)^2}\sum_{i=1}^\tau (1-\eta\epsilon)^{\tau-i}\lambda_i^s}_{\textbf{term}_2}$$

$$+ \underbrace{\eta\sum_{i=1}^\tau (1-\eta\epsilon)^{\tau-i}\xi_i^s}_{\textbf{term}_3} + \underbrace{\eta\sum_{i=1}^\tau (1-\eta\epsilon)^{\tau-i}\zeta_i^s}_{\textbf{term}_4}$$

$$+ \underbrace{\sum_{i=1}^\tau (1-\eta\epsilon)^{\tau-i}v_i^s + \frac{1}{2}\eta\epsilon\epsilon'\sum_{i=1}^\tau (1-\eta\epsilon)^{\tau-i}}_{\triangleq \textbf{term}_5(s,\tau)} + \underbrace{\eta\sum_{i=1}^\tau (1-\eta\epsilon)^{\tau-i}\left[\Delta_i^s - \frac{1}{2}\epsilon\epsilon'\right]_+}_{\triangleq \textbf{term}_6(s,\tau)}$$

$$\overset{(a)}{\leq} \mathcal{O}(e^{-\eta\epsilon\tau}\ln(AT)) + \ln^3(AST/\delta) \times \mathcal{O}\left(\frac{\eta A}{\epsilon(1-\gamma)^2} + \frac{\eta^2 A}{\beta(1-\gamma)^2} + \frac{\beta A}{\epsilon(1-\gamma)} + \frac{1}{1-\gamma}\sqrt{\frac{\eta}{\epsilon}} + \frac{\eta}{\beta(1-\gamma)}\right)$$

$$+ \textbf{term}_5(s,\tau) + \frac{1}{2}\epsilon' + \textbf{term}_6(s,\tau)$$

$$\overset{(b)}{\leq} \mathcal{O}\left(e^{-\eta\epsilon\tau}\ln(AT)\right) + \frac{3}{4}\epsilon' + \textbf{term}_5(s,\tau) + \textbf{term}_6(s,\tau) \tag{26}$$

where in $(a)$ we use the following calculation:

$$\textbf{term}_1 \leq \mathcal{O}\left(\frac{\eta^2 A \ln^2(AT)}{(1-\gamma)^2} \times \frac{1}{\eta\epsilon}\right) \leq \mathcal{O}\left(\frac{\eta A \ln^2(AT)}{\epsilon(1-\gamma)^2}\right).$$

$$\textbf{term}_2 \leq \mathcal{O}\left(\frac{\eta^2 A}{(1-\gamma)^2}\frac{\max_{i\leq\tau}(1-\eta\epsilon)^{\tau-i}\ln(AST/\delta)}{\beta}\right) = \mathcal{O}\left(\frac{\eta^2 A}{(1-\gamma)^2}\frac{\ln(AST/\delta)}{\beta}\right)$$
$$\text{(by Lemma 8)}$$

$$\textbf{term}_3 \leq \mathcal{O}\left(\frac{\eta A}{1-\gamma}\sum_{i=1}^\tau \beta(1-\eta\epsilon)^{\tau-i} + \eta\sqrt{\ln(AST/\delta)\sum_{i=1}^\tau (1-\eta\epsilon)^{\tau-i}}\right) \quad \text{(by Lemma 6)}$$

$$= \mathcal{O}\left(\frac{\beta A}{\epsilon(1-\gamma)} + \sqrt{\ln(AS/\delta)\frac{\eta}{\epsilon}}\right).$$

$$\textbf{term}_4 \leq \mathcal{O}\left(\frac{\eta}{1-\gamma} \times \frac{\max_{i\leq\tau}(1-\eta\epsilon)^{\tau-i}\ln(AST/\delta)}{\beta}\right) = \mathcal{O}\left(\frac{\eta\ln(AST/\delta)}{\beta(1-\gamma)}\right),$$
$$\text{(by Lemma 8)}$$

and in $(b)$ we use the conditions Eq. (23) and Eq. (24).

We continue to bound the sum of $\textbf{term}_5$ and $\textbf{term}_6$ over $t$. Note that

$$\sum_s \sum_{\tau=1}^{n_{T+1}(s)} \textbf{term}_5(s,\tau) \le \sum_s \sum_{\tau=1}^{n_{T+1}(s)} \sum_{i=1}^{\tau} (1-\eta\epsilon)^{\tau-i} v_i^s \le \frac{1}{\eta\epsilon} \sum_s \sum_{i=1}^{n_{T+1}(s)} v_i^s \le \mathcal{O}\left(\frac{S^2 \ln^3(AT)}{\eta\epsilon^2(1-\gamma)^2}\right),$$
(27)

where in the last inequality we use the following calculation:

$$\sum_{i=1}^{n_{T+1}(s)} |v_i^s| \le \mathcal{O}\left(\ln(A\tau)\right) \times \sum_{i=1}^{n_{T+1}(s)} \|\hat{z}_{i\star}^s - \hat{z}_{i+1\star}^s\|_1 \qquad \text{(by Lemma 14)}$$

$$= \mathcal{O}\left(\ln(AT)\right) \times \frac{\ln(AT)}{\epsilon} \times \sum_{i=1}^{n_{T+1}(s)} \sup_{s'}\left(p\left|\underline{V}_{t_i}^{s'} - \underline{V}_{t_{i+1}}^{s'}\right| + (1-p)\left|\overline{V}_{t_i}^{s'} - \overline{V}_{t_{i+1}}^{s'}\right|\right)$$
$$\text{(by the same calculation as Eq. (12))}$$

$$\le \mathcal{O}\left(\frac{\ln^2(AT)}{\epsilon}\right) \times \sum_{s'}\sum_{t=1}^{T}\left(\left|\underline{V}_t^{s'} - \underline{V}_{t+1}^{s'}\right| + \left|\overline{V}_t^{s'} - \overline{V}_{t+1}^{s'}\right|\right)$$

$$\le \mathcal{O}\left(\frac{\ln^2(AT)}{\epsilon} \times \frac{S\ln T}{(1-\gamma)^2}\right)$$
$$(|\underline{V}_t^s - \underline{V}_{t+1}^s| \le \tfrac{H+1}{H+\tau} \times \tfrac{1}{1-\gamma}\mathbf{1}[s_t = s] \text{ by the update rule})$$

$$= \mathcal{O}\left(\frac{S\ln^3(AT)}{\epsilon(1-\gamma)^2}\right),$$

and that

$$\sum_s \sum_{\tau=1}^{n_{T+1}(s)} \textbf{term}_6(s,\tau)$$

$$= \sum_s \sum_{\tau=1}^{n_{T+1}(s)} \eta \sum_{i=1}^{\tau} (1-\eta\epsilon)^{\tau-i}\left[\Delta_i^s - \frac{1}{2}\epsilon\epsilon'\right]_+$$

$$\le \sum_s \sum_{i=1}^{n_{T+1}(s)} \sum_{\tau=i}^{n_{T+1}(s)} \eta(1-\eta\epsilon)^{\tau-i}\left[\Delta_i^s - \frac{1}{2}\epsilon\epsilon'\right]_+$$

$$\le \frac{1}{\epsilon} \sum_s \sum_{i=1}^{n_{T+1}(s)} \left[\Delta_i^s - \frac{1}{2}\epsilon\epsilon'\right]_+$$

$$= \frac{1}{\epsilon} \sum_s \sum_{i=1}^{n_{T+1}(s)} \sum_{j=-1}^{j_{\max}} \mathbf{1}\left[\epsilon\epsilon'2^j \le \Delta_i^s \le \epsilon\epsilon'2^{j+1}\right]\epsilon\epsilon'2^{j+1} \qquad (\text{define } j_{\max} = \log_2\left(\tfrac{1}{(1-\gamma)\epsilon\epsilon'}\right))$$

$$\le \frac{1}{\epsilon} \sum_{j=-1}^{j_{\max}} \sum_{t=1}^{T} \mathbf{1}\left[\Delta_i^{s_t} \ge \epsilon\epsilon'2^j\right]\epsilon\epsilon'2^{j+1}$$

$$\le \frac{1}{\epsilon} \sum_{j=-1}^{j_{\max}} \mathcal{O}\left(\frac{AS\ln^4(AST/\delta)}{\eta\epsilon\epsilon'2^j(1-\gamma)^3}\right) \times \epsilon\epsilon'2^{j+1}$$
$$\text{(by Corollary 2 with } \tilde{\epsilon} = \epsilon\epsilon'2^j \text{ and the assumption that } \epsilon\epsilon' \gtrsim \tfrac{A\ln^3(AST/\delta)\beta}{(1-\gamma)^3})$$

$$= \mathcal{O}\left(\frac{AS\ln^5(AST/\delta)}{\eta\epsilon(1-\gamma)^3}\right) \qquad (\text{without loss of generality, assume } \log_2\left(\tfrac{1}{(1-\gamma)\epsilon\epsilon'}\right) \lesssim \log T)$$
(28)

From Eq. (26), we have

$$\sum_s \sum_{\tau=1}^{n_{T+1}(s)} \mathbf{1}\left[\text{KL}(\hat{z}_{\tau\star}^s, \hat{z}_\tau^s) \ge \epsilon'\right]$$

$$\leq \sum_s \sum_{\tau=1}^{n_{T+1}(s)} \mathbf{1}\left[\mathcal{O}(e^{-\eta\epsilon\tau}\ln(AT)) \geq \frac{1}{12}\epsilon'\right] + \sum_s \sum_{\tau=1}^{n_{T+1}(s)} \mathbf{1}\left[\mathbf{term}_5(s,\tau) > \frac{1}{12}\epsilon'\right]$$

$$+ \sum_s \sum_{\tau=1}^{n_{T+1}(s)} \mathbf{1}\left[\mathbf{term}_6(s,\tau) > \frac{1}{12}\epsilon'\right]$$

$$\leq S \times \mathcal{O}\left(\frac{\ln(AT)}{\eta\epsilon\epsilon'}\right) + \mathcal{O}\left(\frac{S^2\ln^3(AT)}{\eta\epsilon^2\epsilon'(1-\gamma)^2}\right) + \mathcal{O}\left(\frac{AS\ln^5(AST/\delta)}{\eta\epsilon\epsilon'(1-\gamma)^3}\right)$$

$$\leq \mathcal{O}\left(\frac{S^2 A\ln^5(SAT/\delta)}{\eta\epsilon^2\epsilon'(1-\gamma)^3}\right)$$

where in the second-to-last inequality we use Eq. (27) and Eq. (28). This finishes the proof.

$\square$

## E.4 Part IV. Combining

**Theorem 5.** *For any $u \in \left[0, \frac{1}{1-\gamma}\right]$, there exists a proper choice of parameters $\epsilon, \beta, \eta$ such that*

$$\sum_{t=1}^T \mathbf{1}\left[\max_{x,y}\left(x_t^{s_t^\top} Q_\star^{s_t} y^{s_t} - x^{s_t^\top} Q_\star^{s_t} y_t^{s_t}\right) > u\right] \leq \mathcal{O}\left(\frac{S^2 A^3 \ln^{17}(SAT/\delta)}{u^9(1-\gamma)^{13}}\right).$$

*with probability at least $1 - \mathcal{O}(\delta)$.*

*Proof.* We will choose $\epsilon$ such that $u \geq C_7 \frac{\epsilon\ln(AT)}{1-\gamma}$ with a sufficiently large universal constant $C_7$. By Lemma 26, we have

$$\max_{x,y}\left(x_t^{s_t^\top} Q_\star^{s_t} y^{s_t} - x^{s_t^\top} Q_\star^{s_t} y_t^{s_t}\right)$$

$$\leq \max_{x,y}\left(x_t^{s_t^\top}\left(G^{s_t} + \gamma\mathbb{E}_{s'\sim P^{s_t}}\left[\overline{V}_t^{s'}\right]\right)y^{s_t} - x^{s_t^\top}\left(G^s + \gamma\mathbb{E}_{s'\sim P^{s_t}}\left[\underline{V}_t^{s'}\right]\right)y_t^{s_t}\right) + \mathcal{O}\left(\frac{\epsilon\ln(AT)}{1-\gamma}\right)$$

$$\leq \max_{x,y}\left(x_t^{s_t^\top}\left(G^{s_t} + \gamma\mathbb{E}_{s'\sim P^{s_t}}\left[\overline{V}_t^{s'}\right]\right)y^{s_t} - x^{s_t^\top}\left(G^s + \gamma\mathbb{E}_{s'\sim P^{s_t}}\left[\underline{V}_t^{s'}\right]\right)y_t^{s_t}\right) + \frac{u}{4}.$$

Therefore, we can upper bound the left-hand side of the desired inequality by

$$\sum_{t=1}^T \mathbf{1}\left[\max_{x,y}\left(x_t^{s_t^\top}\left(G^{s_t} + \gamma\mathbb{E}_{s'\sim P^{s_t}}\left[\overline{V}_t^{s'}\right]\right)y^{s_t} - x^{s_t^\top}\left(G^s + \gamma\mathbb{E}_{s'\sim P^{s_t}}\left[\underline{V}_t^{s'}\right]\right)y_t^{s_t}\right) \geq \frac{3}{4}u\right]$$

$$\leq \sum_{t=1}^T \mathbf{1}\left[\max_y x_t^{s_t^\top}\left(G^{s_t} + \gamma\mathbb{E}_{s'\sim P^{s_t}}\left[\overline{V}_t^{s'}\right]\right)y^{s_t} - x_t^{s_t^\top}\left(G^{s_t} + \gamma\mathbb{E}_{s'\sim P^{s_t}}\left[\overline{V}_t^{s'}\right]\right)y_t^{s_t} \geq \frac{u}{4}\right]$$

$$+ \sum_{t=1}^T \mathbf{1}\left[x_t^{s_t^\top}\left(G^{s_t} + \gamma\mathbb{E}_{s'\sim P^{s_t}}\left[\overline{V}_t^{s'}\right]\right)y_t^{s_t} - x_t^{s_t^\top}\left(G^{s_t} + \gamma\mathbb{E}_{s'\sim P^{s_t}}\left[\underline{V}_t^{s'}\right]\right)y_t^{s_t} \geq \frac{u}{4}\right]$$

$$+ \sum_{t=1}^T \mathbf{1}\left[x_t^{s_t^\top}\left(G^{s_t} + \gamma\mathbb{E}_{s'\sim P^{s_t}}\left[\underline{V}_t^{s'}\right]\right)y^{s_t} - \min_x x^{s_t^\top}\left(G^{s_t} + \gamma\mathbb{E}_{s'\sim P^{s_t}}\left[\underline{V}_t^{s'}\right]\right)y_t^{s_t} \geq \frac{u}{4}\right].$$

$$(29)$$

For the first term in Eq. (29), we can bound it by

$$\sum_s \sum_{i=1}^{n_{T+1}(s)} \mathbf{1}\left[\max_y \overline{f}_i^s(\hat{x}_i^s, y^s) - \overline{f}_i^s(\hat{x}_i^s, \hat{y}_i^s) \geq \frac{u}{4} - \mathcal{O}\left(\epsilon\ln(AT)\right)\right]$$

$$\leq \sum_s \sum_{i=1}^{n_{T+1}(s)} \mathbf{1}\left[\max_y \overline{f}_i^s(\hat{x}_i^s, y^s) - \overline{f}_i^s(\hat{x}_i^s, \hat{y}_i^s) \geq \frac{u}{8}\right]$$

$$\leq \sum_s \sum_{i=1}^{n_{T+1}(s)} \mathbf{1}\left[\max_y \overline{f}_i^s(\hat{x}_{i\star}^s, y^s) - \overline{f}_i^s(\hat{x}_{i\star}^s, \hat{y}_{i\star}^s) + \mathcal{O}\left(\|\hat{z}_i^s - \hat{z}_{i\star}^s\|_1 \frac{\ln(AT)}{1-\gamma}\right) \geq \frac{u}{8}\right]$$

(because $\|\nabla \overline{f}_i^s(x,y)\|_\infty \leq \mathcal{O}\left(\frac{\ln(AT)}{1-\gamma}\right)$ — similar to the calculation in Eq. (9))

(here we choose $(\hat{x}_{i\star}^s, \hat{y}_{i\star}^s)$ to be the equilibrium under $\overline{f}_i^s(x,y)$)

$$\leq \sum_s \sum_{i=1}^{n_{T+1}(s)} \mathbf{1}\left[\mathcal{O}\left(\|\hat{z}_i^s - \hat{z}_{i\star}^s\|_1 \frac{\ln(AT)}{1-\gamma}\right) \geq \frac{u}{8}\right]$$

$$\leq \sum_s \sum_{i=1}^{n_{T+1}(s)} \mathbf{1}\left[\mathrm{KL}(\hat{z}_{i\star}^s, \hat{z}_i^s) \geq \Omega\left(\frac{u^2(1-\gamma)^2}{\ln^2(AT)}\right)\right]$$

$$\leq \mathcal{O}\left(\frac{S^2 A \ln^7(SAT/\delta)}{\eta \epsilon^2 u^2(1-\gamma)^5}\right) \qquad \text{(by Lemma 27 with } \epsilon' = \Theta\left(\frac{u^2(1-\gamma)^2}{\ln^2(AT)}\right))$$

The third term in Eq. (29) can be bounded in the same way. The second term in Eq. (29) can be bounded using Corollary 2 by

$$\mathcal{O}\left(\frac{SA \ln^4(SAT/\delta)}{\eta u(1-\gamma)^3}\right).$$

Overall, we have

$$\sum_{t=1}^T \mathbf{1}\left[\max_{x,y}\left(x_t^{s_t^\top} Q_\star^{s_t} y^{s_t} - x^{s_t^\top} Q_\star^{s_t} y_t^{s_t}\right) > u\right] \leq \mathcal{O}\left(\frac{S^2 A \ln^7(SAT/\delta)}{\eta \epsilon^2 u^2(1-\gamma)^5}\right). \tag{30}$$

Notice that the parameters $\epsilon, \beta, \eta$ needs to satisfy the conditions specified in this lemma and Lemma 27, with which we apply $\epsilon' = \Theta\left(\frac{u^2(1-\gamma)^2}{\ln^2(SAT/\delta)}\right)$. The constraints suggest the following parameter choice (under a fixed $u$):

$$\epsilon = \Theta\left(\frac{u(1-\gamma)}{\ln(SAT/\delta)}\right)$$

$$\beta = \Theta\left(\frac{(1-\gamma)^3}{A \ln^3(SAT/\delta)}\epsilon\epsilon'\right) = \Theta\left(\frac{u^3(1-\gamma)^6}{A \ln^6(SAT/\delta)}\right)$$

$$\eta = \Theta\left(\frac{(1-\gamma)}{A \ln^3(SAT/\delta)}\beta\epsilon'\right) = \Theta\left(\frac{u^5(1-\gamma)^9}{A^2 \ln^{11}(SAT/\delta)}\right)$$

Using these parameters in Eq. (30), we get

$$\sum_{t=1}^T \mathbf{1}\left[\max_{x,y}\left(x_t^{s_t^\top} Q_\star^{s_t} y^{s_t} - x^{s_t^\top} Q_\star^{s_t} y_t^{s_t}\right) > u\right] \leq \mathcal{O}\left(\frac{S^2 A^3 \ln^{20}(SAT/\delta)}{u^9(1-\gamma)^{16}}\right).$$

□

## F  Discussions on Convergence Notions for General Markov Games

In general Markov games, learning the equilibrium policy pair *on every state* is impossible because some state might have exponentially small visitation probability under all policies. Therefore, a reasonable definition of convergence is the convergence of the following quantity to zero:

$$\frac{1}{T}\sum_{t=1}^T \max_{x,y}\left(V_{x_t,y}^{s_t} - V_{x,y_t}^{s_t}\right), \tag{31}$$

which is similar to the best-iterate convergence defined in Section 3, but over the state sequence visited by the players instead of taking max over $s$. It is also a strict generalization of the sample complexity bound for single-player MDPs under the discounted criteria (see e.g., [LH14, WDCW20]).

The path convergence defined in our work is, on the other hand, that the following quantity converges to zero:

$$\frac{1}{T}\sum_{t=1}^{T}\max_{x,y}\left(x_t^{s_t^\top}Q_\star^{s_t}y^{s_t} - x_t^{s_t^\top}Q_\star^{s_t}y_t^{s_t}\right). \tag{32}$$

Since $\max_y(x^{s^\top}Q_\star^s y^s) \leq \max_y(x^{s^\top}Q_{x,y}^s y^s) = \max_y V_{x,y}^s$ for any $x$, the convergence of Eq. (31) is stronger than Eq. (32).

**Implications of Path Convergence**    Although Eq. (32) does not imply the more standard best-iterate guarantee Eq. (31), it still has meaningful implications. By definition, It implies that frequent visits to a state bring players' policies closer to equilibrium, leading to both players using near-equilibrium policies for all but $o(T)$ number of steps over time.

Path convergence also implies that both players have no regret compared to the game value $V_\star^s$, which has been considered and motivated in previous works such as [BT02, TWYS20]. To see this more clearly, we apply the results to the *episodic* setting, where in every step, with probability $1 - \gamma$, the state is redrawn from $s \sim \rho$ for some initial distribution $\rho$ (every time the state is redrawn from $\rho$, we call it a new episode). We can show that if Eq. (32) vanishes, then every player's long-term average payoff is at least the game value. First, notice that if Eq. (32) converges to zero, then

$$\sum_{t=1}^{T}(V_\star^{s_t} - x_t^{s_t^\top}Q_\star^{s_t}y_t^{s_t}) \leq \max_y \sum_{t=1}^{T}\left(x_t^{s_t^\top}Q_\star^{s_t}y^{s_t} - x_t^{s_t^\top}Q_\star^{s_t}y_t^{s_t}\right)$$

$$\leq \sum_{t=1}^{T}\left(\max_y x_t^{s_t^\top}Q_\star^{s_t}y^{s_t} - x_t^{s_t^\top}Q_\star^{s_t}y_t^{s_t}\right) = o(T). \tag{33}$$

Now fix an $i$ and let $t_i$ be time index at the beginning of episode $i$. Let $E_t = 1$ indicate the event that episode $i$ has not ended at time $t$. Then

$$\mathbb{E}\left[\sum_{t=t_i}^{t_{i+1}-1}\left(V_\star^{s_t} - x_t^{s_t^\top}Q_\star^{s_t}y_t^{s_t}\right)\right]$$

$$= \mathbb{E}\left[\sum_{t=t_i}^{\infty}\mathbf{1}[E_t = 1]\left(V_\star^{s_t} - x_t^{s_t^\top}G^{s_t}y_t^{s_t} - \gamma V_\star^{s_{t+1}}\right)\right]$$

$$= \mathbb{E}\left[\sum_{t=t_i}^{\infty}\mathbf{1}[E_t = 1]\left(V_\star^{s_t} - x_t^{s_t^\top}G^{s_t}y_t^{s_t} - \mathbf{1}[E_{t+1} = 1]V_\star^{s_{t+1}}\right)\right]$$

$$= \mathbb{E}\left[V_\star^{s_{t_i}}\right] - \mathbb{E}\left[\sum_{t=t_i}^{\infty}\mathbf{1}[E_t = 1]x_t^{s_t^\top}G^{s_t}y_t^{s_t}\right]$$

$$= \mathbb{E}_{s\sim\rho}\left[V_\star^s\right] - \mathbb{E}\left[\sum_{t=t_i}^{t_{i+1}-1}x_t^{s_t^\top}G^{s_t}y_t^{s_t}\right].$$

Combining this with Eq. (33), we get

$$\mathbb{E}\left[\sum_{t=1}^{T}x_t^{s_t^\top}G^{s_t}y_t^{s_t}\right] \geq (\#\text{episodes in } T \text{ steps})\mathbb{E}_{s\sim\rho}[V_\star^s] - o(T)$$

$$\geq (1 - \gamma)\mathbb{E}_{s\sim\rho}[V_\star^s]T - o(T).$$

Hence the one-step average reward is at least $(1 - \gamma)\mathbb{E}_{s\sim\rho}[V_\star^s]$. A symmetric analysis shows that it is also at most $(1 - \gamma)\mathbb{E}_{s\sim\rho}[V_\star^s]$. This shows that both players have no regret compared to the game value. Notice that this is only a loose implication of the path convergence guarantee because of the loose second inequality in Eq. (33).

**Remark on the notion of "last-iterate convergence" in general Markov games**    While Eq. (31) corresponds to best-iterate convergence for general Markov games, an even stronger notion one can

pursue after is "last-iterate convergence." As argued above, it is impossible to require that the policies on all states to converge to equilibrium. To address this issue, we propose to study this problem under the episodic setting described above, in which the state is reset after every trajectory whose expected length is $\frac{1}{1-\gamma}$. In this case, last-iterate convergence will be defined as the convergence of the following quantity to zero when $i \to \infty$:

$$\mathbb{E}_{s \sim \rho} \left[ \max_{x,y} \left( V^s_{x_{t_i},y} - V^s_{x,y_{t_i}} \right) \right]$$

where we recall that $i$ is the episode index and $(x_{t_i}, y_{t_i})$ are the policies used by the two players at the beginning of episode $i$. While last-iterate convergence seems reasonable and possibly achievable, we are unaware of such results even for the degenerated case of single-player MDPs — the standard regret bound corresponds to best-iterate convergence, while the techniques we are aware of to prove last-iterate convergence in MDPs require additional assumptions on the dynamics.

