# OpenReview forum: "Uncoupled and Convergent Learning in Two-Player Zero-Sum Markov Games with Bandit Feedback"
_NeurIPS.cc/2023/Conference — NeurIPS 2023 poster_

### Official Review · Reviewer_UUTi · 2023-06-28

**Soundness:** 3 good
**Presentation:** 3 good
**Contribution:** 2 fair
**Rating:** 6
**Confidence:** 3

**Summary:**

The paper studies the problem of designing uncoupled learning dynamics that provably converge to Nash equilibria in two-player zero-sum Markov games. As a preliminary result, the paper introduces the first dynamics that converge last iterate in self play in matrix games with bandit feedback. Then, the paper shows how to extend such last-iterate-convergent learning dynamics to the case of irreducible Markov games. Finally, the paper studies the case of general Markov games, proving that a variation of the previously-introduced learning dynamics achieve convergence according to a newly introduced definition of convergence (called path convergence).

**Strengths:**

The problem studied in the paper is interesting and it has received considerable attention over the last years. The paper is well written and easy to follow.

**Weaknesses:**

A major weakness that I see is that the results presented in the paper seem minor adaptations and different analysis of already known techniques/tools. The authors should focus more on discussing the novelty of their approach.

**Questions:**

Please resolve my concerns in the weaknesses part.

**Limitations:**

None.

---

> ### Author Rebuttal · Authors · 2023-08-09
>
> Thank you for your comments! We address your concerns below.
>
> *Q: A major weakness that I see is that the results presented in the paper seem minor adaptations and different analysis of already known techniques/tools. The authors should focus more on discussing the novelty of their approach.*
>
> A : Although our algorithms share similarity with previous works that also use entropy regularization, we believe that both the design and the analysis of the algorithms are novel and non-trivial. To the best of our knowledge, all previous entropy regularized two-player zero-sum Markov game algorithms are coupled (e.g., [1,2,3]), while ours is the first that achieves uncoupledness under entropy regularization. We will further discuss this by comparing our algorithms to those in [3], highlighting the new technical challenges we encounter.
>
> The entropy-regularized OMWU algorithm in [3] is tailored to the full-information setting and their value function updates require both players' entropy information: $ V^{t+1}(s)=(1 - \alpha_{t+1})V^t(s) +\alpha_{t+1} ( x^{t+1}(s)^\top Q^{t+1}(s)y^{t+1}(s) + \tau  \phi(x^{t+1}(s)) - \tau \phi(y^{t+1}(s))$. This necessitates both players to know the entropy value of the other player's policy, which is unnatural. Indeed, the authors explicitly present the removal of this information sharing as an open question: *[the introduction of entropy regularization requires each agent to reveal the entropy of their current policy to each other, which prevents the proposed method from being fully decentralized. Can we bypass this by dropping the entropy information in value learning? We leave the answers to future work.]*
>
> We answer this open question affirmatively by giving a fully decentralized algorithm for zero-sum Markov games with provable last-iterate convergence rates. In Algorithm 2, the update of the value function $V$ is simple and does not require any entropy information: $V_{t+1}^{s_t} \leftarrow (1-\alpha_\tau)V_t^{s_t} + \alpha_\tau \left(\sigma_t + \gamma V_t^{s_{t+1}}\right)$. This modification results in a discrepancy between the policy update and the value update. While the policy now incorporates a regularization term, the value function does not. Such a mismatch is unprecedented in earlier studies and necessitates a non-trivial approach to resolve.
> Additionally, Algorithm 2 operates on bandit feedback instead of full-information feedback, presenting further technical challenges
>
> [1] Cen, Shicong, Yuting Wei, and Yuejie Chi. "Fast policy extragradient methods for competitive games with entropy regularization." NeurIPS 2021.
>
> [2] Ziyi Chen, Shaocong Ma, and Yi Zhou. Sample efficient stochastic policy extragradient algorithm for zero-sum
> markov game. ICLR 2021.
>
> [3] Cen, Shicong, Yuejie Chi, Simon S. Du, and Lin Xiao. "Faster last-iterate convergence of policy optimization in zero-sum Markov games." ICLR 2023.

---

> > ### Comment · Reviewer_UUTi · 2023-08-20
> >
> > I would like to thank the authors for their response. They convinced me that there is indeed several novel components in their analysis. Thus, after having a look back to the paper and to the other reviews, I decided to raise my score accordingly.

---

### Official Review · Reviewer_EDuK · 2023-07-03

**Soundness:** 3 good
**Presentation:** 3 good
**Contribution:** 3 good
**Rating:** 6
**Confidence:** 3

**Summary:**

The paper introduces a new algorithm for learning in two-player zero-sum Markov games based on prior work that  is uncoupled (agents only need their own reward as feedback), convergent (to NE) and rational. The result is an algorithm that is similar in concept to a single agent RL algorithm, but guarantees convergence if both players use the same algorithm. For their initial algorithm, there is an additional assumption of irreducibility on the Markov games, and it achieves last iterate convergence to the Nash with rate O(t^{-1/9+eps}). Meanwhile, for general 2 player zero sum Markov games without additional assumptions, a modification of the algorithm that applies optimism achieves O(t^{-1/10}) path convergence rate.

**Strengths:**

The algorithm proposed takes inspiration from prior works and makes useful modifications that give practical and theoretical advantages. The structure of the results and explanation of analysis also flows well and is easy to read despite the technical complexity. The authors are also very clear in their motivation and I find the literature review to be quite comprehensive.

**Weaknesses:**

The paper does a good job of setting up motivation and prior ideas, but in my opinion the key takeaway from this work is that there is now a truly uncoupled and convergent algorithm for general Markov games, which is in my opinion the paper's most significant contribution. However, there is too much space dedicated to matrix and irreducible Markov games, when in my view, the section on general Markov games should be greatly expanded. In addition to the overview of the analysis, I am interested to know how optimism can be leveraged in this type of algorithm, and the thought process behind the modifications to Algorithm 3 compared to Algorithm 2. As a suggestion, perhaps part of the analysis overview for Thm 3 can be repurposed and moved forward to add to the explanation about Alg 3, which would improve the flow of the section in my opinion. Finally, there is a lack of experimental results which would make help frame the results better in comparison to prior work.

**Questions:**

Have any experiments been run using Algorithms 1, 2 and 3 to compare them to current SOTA methods? If so, how does the empirical performance compare? It seems that your algorithms would scale well with larger games since players only use local information, how does this compare to existing methods?

**Limitations:**

The authors have adequately addressed the paper's limitations.

---

> ### Author Rebuttal · Authors · 2023-08-09
>
> Thank you for your positive and constructive feedback! We will add more discussions on the optimism technique in Algorithm 3 in the revised version. We address your question below.
>
> *Q: Have any experiments been run using Algorithms 1, 2 and 3 to compare them to current SOTA methods? If so, how does the empirical performance compare? It seems that your algorithms would scale well with larger games since players only use local information, how does this compare to existing methods?*
>
> A: To our knowledge, our work presents the first provable algorithms that are uncouple, convergent, and have finite rate under bandit feedback. Therefore, there is no comparable previous state of the art in our setting. However, if we only care about computing Nash equilibrium under bandit feedback (without caring about uncoupledness and convergence), there are a few algorithms that have faster rates than our work. For example, the V-Learning algorithm [1] has a faster $O(t^{-1/2})$ *average-iterate* convergence rate but has no last-iterate/path convergence guarantee.  We do think that it is a very interesting future direction to evaluate the empirical performance of our algorithms in applications such as Game AI [2].
>
> [1] Jin C, Liu Q, Wang Y, et al. V-Learning--A Simple, Efficient, Decentralized Algorithm for Multiagent RL[J]. arXiv preprint arXiv:2110.14555, 2021
>
> [2] Perolat J, De Vylder B, Hennes D, et al. Mastering the game of Stratego with model-free multiagent reinforcement learning[J]. Science, 2022, 378(6623): 990-996.

---

> > ### Comment · Reviewer_EDuK · 2023-08-16
> > **Response to Author Rebuttal**
> >
> > Thanks for the answer to my question. If the changes suggested by the authors is implemented I think this is a good paper worthy of acceptance.
> >
> > Best regards, Reviewer EDuK

---

### Official Review · Reviewer_dxZp · 2023-07-04

**Soundness:** 3 good
**Presentation:** 4 excellent
**Contribution:** 3 good
**Rating:** 7
**Confidence:** 3

**Summary:**

The paper studies the last iteration convergence of uncoupled learning in two player zero-sum markov game, with bandit feedback. The paper provide the first finite last-iterate convergence guarantee under the bandit feedback.

The paper studies the problem of two-player zero-sum Markov game and designs a new uncoupled learning algorithm that enjoys last iteration convergence, under bandit feedback. Along the way, the paper derive a couple of results, summarized below:

(1) Even for the standard matrix game (where there is no Markov transition), the paper provides an algorithm with $O(T^{-1/8})$ convergence rate

(2) For irreducible Markov game, the paper provides an algorithm with $O(T^{-1/9})$ convergence rate

(3) For general irreducible Markov game, the paper provides an algorithm with $O(T^{-1/10})$ convergence rate, but only under the notion of path convergence.

It is worth noting that there is a large body of literature, but previous work [WLZL21] are either not uncoupled, or [BJY'20 ] not last iterate convergent.

The technical part seems novel, from a high level, the results are obtained by adding an entropy regularizer, but there are many subtle details to make it really work and the analysis is involved.

------------------
I have read the author response and want to keep my positive evaluation.



**Strengths:**

The paper studies a fairly popular topic and provide the first finite last-iterate convergence guarantee for Markov game (two-player, zero-sum). The technical contribution seems to be novel. The paper is also well-written and the technical part is well explained.

**Weaknesses:**

No.

**Questions:**

I have a few minor questions:

(1) Can you provide some intuition on the number of $T^{-1/8}$ or $T^{-1/9}$, ideally, it would be nice if one can written down a short explanation on how these magic exponent comes from.

(2) The following paper seems very relevant?

[1] Regret Minimization and Convergence to Equilibria in General-sum Markov Games

**Limitations:**

no.

---

> ### Author Rebuttal · Authors · 2023-08-09
>
> Thank you for your very positive comments! We address your questions below.
>
> *Q1: Can you provide some intuition on the number of $T^{-1/8}$ or $T^{-1/9}$, ideally, it would be nice if one can written down a short explanation on how these magic exponent comes from.*
>
> A: These exponents are a result of the parameters $k_\eta, k_\beta, k_\epsilon$ (with $\eta_t = t^{-k_\eta}$, $\beta_t = t^{-k_\beta}$, $\epsilon_t = t^{-k_\epsilon}$) we choose to optimize the convergence rate.
> For example, the analysis of Algorithm 1 (see Appendix B) shows that the last-iterate convergence rate  (ignoring log factors and other dependence) is $O(t^{\frac{-k_\eta + k_\epsilon}{4}} + t^{\frac{k_\beta}{2} - k_\eta} + t^{\frac{-k_\beta+k_\epsilon}{2}} + t^{\frac{-k_\eta +k_\beta}{2}} + t^{\frac{-1+k_\eta +k_\epsilon}{2}} + t^{-k_\epsilon})$. The choice of $k_\eta = \frac{5}{8}, k_\beta = \frac{3}{8}, k_\epsilon = \frac{1}{8}$ gives us the optimized $O(t^{-\frac{1}{8}})$ rate.
>
> *Q2: The following paper seems very relevant? [1] Regret Minimization and Convergence to Equilibria in General-sum Markov Games*
>
> A: Thank you for pointing this out! We will add discussion about this paper in the revised version of our paper. Erez et al. (2022) [1] studies regret minimization in general-sum Markov games and provide algorithm with sublinear regret under self-play and *average-iterate* convergence to equibria, while our work focuses on *last-iterate* convergence rates to Nash equilibrium.

---

### Official Review · Reviewer_Zo84 · 2023-07-09

**Soundness:** 3 good
**Presentation:** 2 fair
**Contribution:** 3 good
**Rating:** 6
**Confidence:** 2

**Summary:**

This paper studies the problem of learning a Nash equilibrium in two-player zero-sum Markov games with bandit feedback.
The proposed algorithm introduces the entropy regularization technique into online mirror descent.
First, the author proves that the proposed algorithm converges to an equilibrium in two-player normal-form games.
Then, the last-iterate convergence rate for irreducible Markov games is provided.
Finally, the paper presents a path convergence rate for Markov games without the irreducible assumption.

**Strengths:**

* The problem is well-motivated. In many scenarios, the last-iterate convergence property under bandit feedback is more suitable than the average-iterate convergence property.
* It seems novel to derive the last-iterate convergence rates under bandit feedback.
* The proof sketch of Theorem 1 is intuitive and easy to follow.

**Weaknesses:**

* Since action probabilities of strategies in $\Omega_t$ are lower bounded by $\frac{1}{At^2}$, line 6 in Algorithm 1 seems to have no closed-form solution. I wonder how much computational cost it will take to update the strategy.
* The term of $\ln^{20}(SAT/\delta)$ in Theorem 3 depends heavily on $T$. This term would be dominant practically and would not be able to be ignored.
* I could not completely follow the sketch of Theorem 3. I would have appreciated more detail.

**Questions:**

* How much computational cost will it take to update the strategy (e.g., line 6 in Algorithm 1)?
* Does the path convergence imply the average-iterate convergence?
* What prior knowledge of the game is required for setting of $\epsilon,\beta$, and $\eta$ in Theorem 3?

**Limitations:**

The authors adequately addressed the limitations.

---

> ### Author Rebuttal · Authors · 2023-08-09
>
> Thank you for your positive and constructive comments. We will include a more detailed proof sketch of Theorem 3 in the revised version. Your questions are addressed below.
>
> *Q1:How much computational cost will it take to update the strategy (e.g., line 6 in Algorithm 1)?*
>
> A: To update the strategy, the main computation happens in line 6 where we need to solve the following optimization problem over $\Omega_t=\{x \in \Delta^{|A|}: x[i] \ge \frac{1}{|A|t^2}\}$: $x_{t+1} = \arg \min_{x \in \Omega_t} (g_t^\top x + \frac{1}{\eta} D(x, x_t))$. By the KKT condition, we have $$x_{t+1}[i] =  \frac{x_t[i] \cdot \exp(-\eta_t(g_t[i] + \lambda_i))}{\sum_{i'\in[|A|]} x_t[i'] \cdot \exp(-\eta_t(g_t[i'] + \lambda_{i'}))},$$ where $\lambda_i = 0$ if $x_{t+1}[i] > \frac{1}{|A|t^2}$ and $\lambda_i\le 0$ if $x_{t+1}[i] = \frac{1}{|A|t^2}$. The computation of feasible $(x_{t+1}[i], \lambda_i)$ can be done in $O(|A|\log |A|)$ time as explained below.
>
> For simplicity, let us denote $h[i] :=  x_t[i]\cdot\exp(-\eta_t g_t[i])$. The algorithm works as follows:
> 1. sort $i$'s based on the value of $h[i]$ in increasing order and let us assume w.o.l.g. that $h[1] \le h[2] \ldots \le h[|A|]$;
> 2. find $j \in [|A|]$ such that $\frac{j\cdot h[j]}{j\cdot h[j] + \sum_{i=j+1}^{|A|} h[i]} \le \frac{j}{|A|t^2}$ and $\frac{(j+1)\cdot h[j+1]}{(j+1)\cdot h[j+1] + \sum_{i=j+2}^{|A|} h[i]} > \frac{j+1}{|A|t^2}$; if such $j$ does not exist, then we can set $\lambda_i = 0$ for all $i$ and compute $x_{t+1}$.
> 3. compute $\theta$ such that $\frac{j\cdot \theta}{j\cdot \theta + \sum_{i=j+1}^{|A|} h[i]} = \frac{j}{|A|t^2}$ and set $x_{t+1}[i] = \frac{1}{|A|t^2}$ for $1 \le i \le j$ and $x_{t+1}[i] = \frac{h[i]}{j\cdot \theta + \sum_{i=j+1}^{|A|} h[i]}$ for $j+1 \le i \le |A|$.
>
> Sorting in step 1 can be done in $O(|A| \log |A|)$ time and the computation in other steps can be done in time $O(|A|)$.
>
> *Q2: Does the path convergence imply the average-iterate convergence?*
>
> A: Path convergence does not imply averge-iterate convergence. Path convergence concerns visited states $\{s_t\}$ only and the policies on states that are not reached in learning can be arbitrary, so it does not imply average-iterate convergence. Similarly, path convergence does not imply best-iterate convergence. However, we would like to remark that path convergence excludes the possibility of cycling and has many interesting game-theoretical implications (see our discussion in section 6.1 and Appendix F).
>
>
> *Q3: What prior knowledge of the game is required for setting of $\eta$, $\epsilon$, and $\beta$ in Theorem 3?*
>
> A: The prior knowledge of the game needed for setting $\eta$, $\epsilon$, and $\beta$ is the number of states $S$, the number of actions $A$, and the discounting factor $\gamma$. See Appendix G (page 39) for the concrete choice of these hyper parameters, and we plan to include it in the main body in the revised version.

---

### Official Review · Reviewer_qUKM · 2023-07-30

**Soundness:** 3 good
**Presentation:** 3 good
**Contribution:** 3 good
**Rating:** 6
**Confidence:** 3

**Summary:**

This paper studies algorithms for two-player zero-sum Markov games that are uncoupled, convergent, and rational. Previous attempts at designing such algorithms failed short in one aspect or the other. This work uses recent advances in entropy-based regularization to design new algorithms that overcome the inherent challenges. The other main contribution of this work is deriving the convergent rates of the proposed algorithms in Matrix, Markov, and general Markov games.


**Strengths:**

1. The proposed algorithms do not require the two players to exchange information (including that related to entropy)
2. The proposed ideas extend to general Markov games without any assumptions on the dynamics.
3. The work seems technically sound. The algorithms follow easy-to-implement modifications of existing methods.
4. The problem setup, the algorithms, and the proofs are written in an easy-to-digest format.





**Weaknesses:**

See Questions.

**Questions:**

1. How tight are the obtained convergence rates?
2. My understanding of the proposed algorithm is that at time step $t,$ player 1 plays action $a _t$ and player 2 action $b_t.$ Thereafter, the local update steps are carried out. If this is correct, then why do the authors claim that their proposed algorithm does not require the two players to have synchronized policy updates?




**Limitations:**

The authors do address the limitations in their work adequately.

---

> ### Author Rebuttal · Authors · 2023-08-09
>
> Thank you for your positive comments. We address your questions below.
>
> *Q1:How tight are the obtained convergence rates?*
>
> A: The obtained convergence rates in this paper may not be tight. For example, Algorithm 1 achieves a high-probability $O(t^{-1/8})$ last-iterate convergence rate under bandit feedback for matrix games, while the known best lower bound of convergence rate in this setting is $\Omega(t^{-1/2})$. However, we remark that no  last-iterate/path convergence *rates* are known in the considered setting (i.e., uncoupled algorithm with bandit feedback) prior to our work. Improving the convergence rates and establishing matching lower bounds are very interesting future directions.
>
> *Q2: My understanding of the proposed algorithm is that at time step $t$ player 1 plays action $a_t$ and player 2 action $b_t$.
> Thereafter, the local update steps are carried out. If this is correct, then why do the authors claim that their proposed algorithm does not require the two players to have synchronized policy updates?*
>
> A: Thank you for pointing this out. Our interaction follows the standard model in the literature, where both players simultaneously choose an action in each round. This is not what we meant by ``synchronized policy updates.'' By "synchronized policy updates," we refer to the method in [1] where each player is directed to use a fixed policy when interacting with others for an extended period. Afterward, all players update their policies simultaneously. We find such coordination between the players unnatural, and the subsequent algorithm no longer maintains the no-regret guarantee against adversaries. This drawback is also pointed out by [2] (see their second bullet point on Page 3). In contrast, the algorithm in our paper uses a very simple policy update rule and is robust (no-regret) even against an adversary. We will add the above explanation to the revised paper.
>
> **References：**
>
> [1] Wei, C. Y., Lee, C. W., Zhang, M., & Luo, H. Last-iterate convergence of decentralized optimistic gradient descent/ascent in infinite-horizon competitive Markov games. COLT, 2021
>
> [2]  Sayin, M., Zhang, K., Leslie, D., Basar, T., & Ozdaglar, A. Decentralized Q-learning in zero-sum Markov games. NeurIPS, 2021

---

> > ### Comment · Reviewer_qUKM · 2023-08-13
> >
> > Thank you for your response. The gap between the lower bound and the obtained upper bound iss extremely high. It may be useful to have a discussion, highlighting the key reasons for the gap. Secondly, the use of the terminology `synchronous' seems incorrect, and you may want to consider an alternative description.

---

> > > ### Author Response · Authors · 2023-08-18
> > >
> > > Thank you for your feedback!
> > >
> > > 1. We will include a discussion on the gap between upper and lower bounds. We first remark that if we only care about convergence in expectation rather than high-probability bound, then we can get $t^{-\frac{1}{6}}$ last-iterate convergence rate in matrix games (see Appendix C). Towards closing the gap between upper and lower bounds, the following directions are promising. Firstly, [1] proves an impossibility result that specific algorithms with $\sqrt{T}$ regret does not converge in last-iterate, indicating that the current $\frac{1}{\sqrt{T}}$ lower bound on convergence rate may not be tight. Secondly, our results provide insights and a useful template for further improvement on upper bounds. For instance, instead of EXP3 update, adaptation of optimistic policy update or accelerated first-order methods to the bandit feedback setting might give faster rates.
> > >
> > > 2. Thank you for pointing out! We will change the term "synchronous policy update" to "coordinated policy update" in the revised version.
> > >
> > > [1] Muthukumar, Vidya, Soham Phade, and Anant Sahai. "On the Impossibility of Convergence of Mixed Strategies with No Regret Learning." arXiv preprint arXiv:2012.02125 (2020).

---

### Decision · Program_Chairs · 2023-09-21

**Decision:**

Accept (poster)

**Comment:**

As its core contribution, the paper provides uncoupled learning algorithms for  zero-sum matrix games and irreducible Markov games with provable last-iterate convergence rates under bandit feedback. All reviewers are positive about the theoretical contributions. Notably though, there remains a wide gap between the last-iterate upper bound and best known lower bound in these settings. It is suggested that the authors could add discussions the limitation of their results and give some insights on why such a gap exists and possibly how to bridge them.